# TFPa/HADHA is required for fatty acid beta-oxidation and cardiolipin re-modeling in human cardiomyocytes

Jason W. Miklas[1,2], Elisa Clark [1,2], Shiri Levy[1,3], Damien Detraux [1,3], Andrea Leonard[1,4,5], Kevin Beussman[1,4,5], Megan R. Showalter[6], Alec T. Smith[2], Peter Hofsteen[1,5,7], Xiulan Yang[1,5,7], Jesse Macadangdang[1,2], Tuula Manninen[8,9], Daniel Raftery [10], Anup Madan[11], Anu Suomalainen [8,9,12], Deok-Ho Kim [1,2], Charles E. Murry [1,2,5,7,13], Oliver Fiehn [6,14], Nathan J. Sniadecki [1,2,4,7], Yuliang Wang[1,15] & Hannele Ruohola-Baker[1,2,3]*

Mitochondrial trifunctional protein deficiency, due to mutations in hydratase subunit A (HADHA), results in sudden infant death syndrome with no cure. To reveal the disease etiology, we generated stem cell-derived cardiomyocytes from HADHA-deficient hiPSCs and accelerated their maturation via an engineered microRNA maturation cocktail that upregulated the epigenetic regulator, *HOPX*. Here we report, matured HADHA mutant cardiomyocytes treated with an endogenous mixture of fatty acids manifest the disease phenotype: defective calcium dynamics and repolarization kinetics which results in a pro-arrhythmic state. Single cell RNA-seq reveals a cardiomyocyte developmental intermediate, based on metabolic gene expression. This intermediate gives rise to mature-like cardiomyocytes in control cells but, mutant cells transition to a pathological state with reduced fatty acid beta-oxidation, reduced mitochondrial proton gradient, disrupted cristae structure and defective cardiolipin remodeling. This study reveals that HADHA (tri-functional protein alpha), a monolysocardiolipin acyltransferase-like enzyme, is required for fatty acid beta-oxidation and cardiolipin remodeling, essential for functional mitochondria in human cardiomyocytes.

[1] Institute for Stem Cell and Regenerative Medicine, University of Washington, School of Medicine, Seattle, WA 98109, USA. [2] Department of Bioengineering, University of Washington, Seattle, WA 98195, USA. [3] Department of Biochemistry, University of Washington, School of Medicine, Seattle, WA 98195, USA. [4] Department of Mechanical Engineering, University of Washington, Seattle, WA 98195, USA. [5] Center for Cardiovascular Biology, University of Washington, Seattle, WA 98109, USA. [6] NIH West Coast Metabolomics Center, University of California Davis, Davis, CA 95616, USA. [7] Department of Pathology, University of Washington, Seattle, WA 98109, USA. [8] Helsinki University Hospital, 00290 Helsinki, Finland. [9] Research Programs Unit, Stem Cells and Metabolism, University of Helsinki, 00290 Helsinki, Finland. [10] Department of Anesthesiology and Pain Medicine, Mitochondria and Metabolism Center, University of Washington, Seattle, WA 98109, USA. [11] Covance Genomics Laboratory, Redmond, WA 98052, USA. [12] Neuroscience Center, University of Helsinki, 00290 Helsinki, Finland. [13] Department of Medicine/Cardiology, University of Washington, Seattle, WA 98109, USA. [14] Biochemistry Department, Faculty of Science, King Abdulaziz University, Jeddah, Saudi Arabia. [15] Paul G. Allen School of Computer Science & Engineering, University of Washington, Seattle, WA 98195, USA. *email: hannele@u.washington.edu

Mitochondrial trifunctional protein (MTP/TFP) deficiency is thought to be a result of impaired fatty acid oxidation (FAO) due to mutations in hydroxyacyl-CoA dehydrogenase/3-ketoacyl-CoA thiolase/enoyl-CoA hydratase subunit A (HADHA HADHA/LCHAD) or subunit B (HADHB)[1]. A major phenotype of MTP-deficient newborns is sudden infant death syndrome (SIDS), which manifests after birth once the child begins nursing on lipid-rich breast-milk. Defects in FAO have a role in promoting a pro-arrhythmic cardiac environment; however, the exact mechanism of action is not understood, and there are no current therapies[2,3].

Pluripotent stem cell derived cardiomyocytes (hPSC-CM) provide a means to study human disease in vitro but have limitations due to their immaturity as they are representative of fetal cardiomyocytes (FCM) instead of adult cardiomyocytes (ACM)[4,5]. Due to the lack of knowledge in how committed cardiomyocytes transition from an immature FCM to a mature ACM, many cardiac diseases with postnatal onset are poorly characterized[6–11] During cardiogenesis, FCMs go through developmental states and once past cardiomyocyte commitment exhibit: exit of cell cycle, utilization of lactate, cessation of spontaneous beating, and then at the postnatal stage utilization of fatty acids and cardiolipin maturation[12–17]. Since immature hPSC-CMs are unable to utilize fatty acids through FAO as an energy source, they are limited in their use to model FAO disorders.

Current approaches to mature hPSC-CMs toward ACM focus on prolonged culture[18], physically stimulating the cells with either electrical[19] or mechanical stimulation[20] or by 2D surface pattern cues to direct cell orientation[21]. An emerging area of hPSC-CM maturation is in manipulating microRNAs (miRs)[22–24]. Overexpressing just one miR, Let-7, can accelerate human embryonic stem cell derived cardiomyocyte (hESC-CM) maturation towards an ACM-like state[24]. However, no maturation regimen has been able to mature hPSC-CMs to an adult state.

In this study, we analyze mitochondrial trifunctional protein deficiency by generating stem cell derived cardiomyocytes from HADHA-deficient human induced pluripotent stem cells (hiPSCs) and accelerate their maturation by our engineered microRNA maturation cocktail. The data reveal the essential dual role of HADHA in fatty acid beta-oxidation and as an acyltransferase in cardiolipin remodeling for cardiac homeostasis.

## Results

### Generation of MTP deficient CMs.
To recapitulate the cardiac pathology of mitochondrial trifunctional protein deficiency on the cellular level in vitro (Fig. 1a), we used the CRISPR/Cas9 system to generate mutations in the gene HADHA of human iPSCs. From our wild-type (WT) hiPSC line, that serves as our isogenic control, we generated HADHA mutant hiPSC lines using two different guides targeting exon 1 of HADHA (Supplemental Fig. 1A, B). To identify the phenotypes specifically caused by mutations in HADHA and to control for potential background effects, we chose to study a knockout (KO) HADHA (HADHA^KO) and compound heterozygote (HADHA^Mut) hiPSC lines that were generated using gRNA1. We also utilized the hiPSC line HEL87.1, which was derived from a patient carrying the founder point mutation most common in mitochondrial trifunctional protein disorder, HADHA c.1528G>C (Supplemental Fig. 1C, D)[25].

Examining the DNA sequence of the HADHA^KO line showed a homozygous 22 bp deletion, which resulted in an early stop codon in exon 1 (Fig. 1b and Supplemental Fig. 1E). The HADHA^Mut line had a 2 bp deletion and 9 bp insertion on the first allele and a 2 bp insertion on the second allele (Fig. 1c and Supplemental Fig. 1F). Both lines showed no off-target mutations on the top

three predicted sites (Supplemental Fig. 1G). The mutations found in the HADHA^Mut line resulted in a predicted early stop codon on both alleles (Supplemental Fig. 1H). However, when we examined the protein in each line we found that HADHA was expressed in the WT hiPSC line, not expressed in the HADHA^KO line and was still expressed, to a lower degree, in the HADHA^Mut line (Fig. 1d). We then examined the transcript of HADHA expressed in WT and HADHA^Mut lines. We found the WT line expressed the full length HADHA transcript from exon 1–20 while the HADHA^Mut line skipped exons 1–3 and expressed HADHA exons 4–20 (Fig. 1c). It is possible that the mutations generated at the intron-exon junction induced an alternative splicing event and a new transcript since there is no known transcript of HADHA from exon 4–20 (Supplemental Fig. 1I). The observed reduction in the HADHA mutant molecular weight (Fig. 1d) supports this hypothesis. The expressed HADHA^Mut protein skips the expression of exons 1–3, 60 amino acids, generating a truncated ClpP/crotonase domain, which likely compromises the mitochondrial localization and protein folding of the enzyme pocket resulting in the inability to stabilize enolate anion intermediates during FAO (Supplemental Fig. 1J).

Using a monolayer directed differentiation protocols[26,27] we generated human induced pluripotent stem cell derived cardiomyocytes (hiPSC-CMs) from the WT and hiPSC lines with HADHA mutations. We found that the reduction or loss of HADHA did not hinder the ability to generate cardiomyocytes (Fig. 1e). However, we found that all CMs, even the control CMs, were unable to utilize long-chain FAs (Fig. 1f) and needed to be matured[14,24,28].

### Screening microRNAs for hPSC-CM maturation.
MicroRNAs have recently been shown to regulate the key, opposing processes of cardiomyocyte regeneration, maturation and dedifferentiation[24,29,30]. We cross-referenced in vivo miR-sequencing data of human fetal ventricular to adult ventricular myocardium[24,31,32] and combined multiple miRs together with Let-7 to rapidly mature hPSC-CMs by promoting a more complete adult like transcriptome. Analyzing each miR's predicted targets affecting glucose and/or fatty acid metabolism, cell growth and hypertrophy and cell cycle, six miRs were chosen to assess for their CM maturation potency: three upregulated miRs (miR-452, −208b[33] and −378e[34]) and three downregulated miRs (miR-122, −200a, and −205) (Fig. 2a).

### Functional analysis of candidate microRNAs.
These six miRs were assessed using four functional tests to determine hPSC-CM maturation: cell area, force of contraction, metabolic capacity, and electrophysiology. WT D15 hiPSC-CMs were transduced with a lentivirus to either OE a miR or KO a miR using CRISPR/Cas9. Cells were then lactate selected to enrich for the cardiomyocyte population and puromycin selected to enrich for the population containing the viral vector. Functional assessment was performed after two weeks of miR perturbation on D30 (Fig. 2b).

An important feature of cardiomyocyte maturation is an increase in cell size. We found only miR-208b OE brought a significant increase in cell area (Fig. 2c and Supplemental Fig. 2A). Immature hPSC-CMs spontaneously beat at a high rate and have a short field potential duration when studied by extracellular micro-electrodes. Using micro-electrode arrays, we found only miR-452 OE increased the corrected field potential duration (cFPD) to a more adult like duration (Fig. 2d). One of the hallmarks of cardiomyocyte maturation is the increase in contractile force generated by the cell. We performed single cell force of contraction analysis using a micropost platform[24,35,36] and found only the KO of miR-200a brought about a significant increase in force of contraction (Fig. 2e and Supplemental

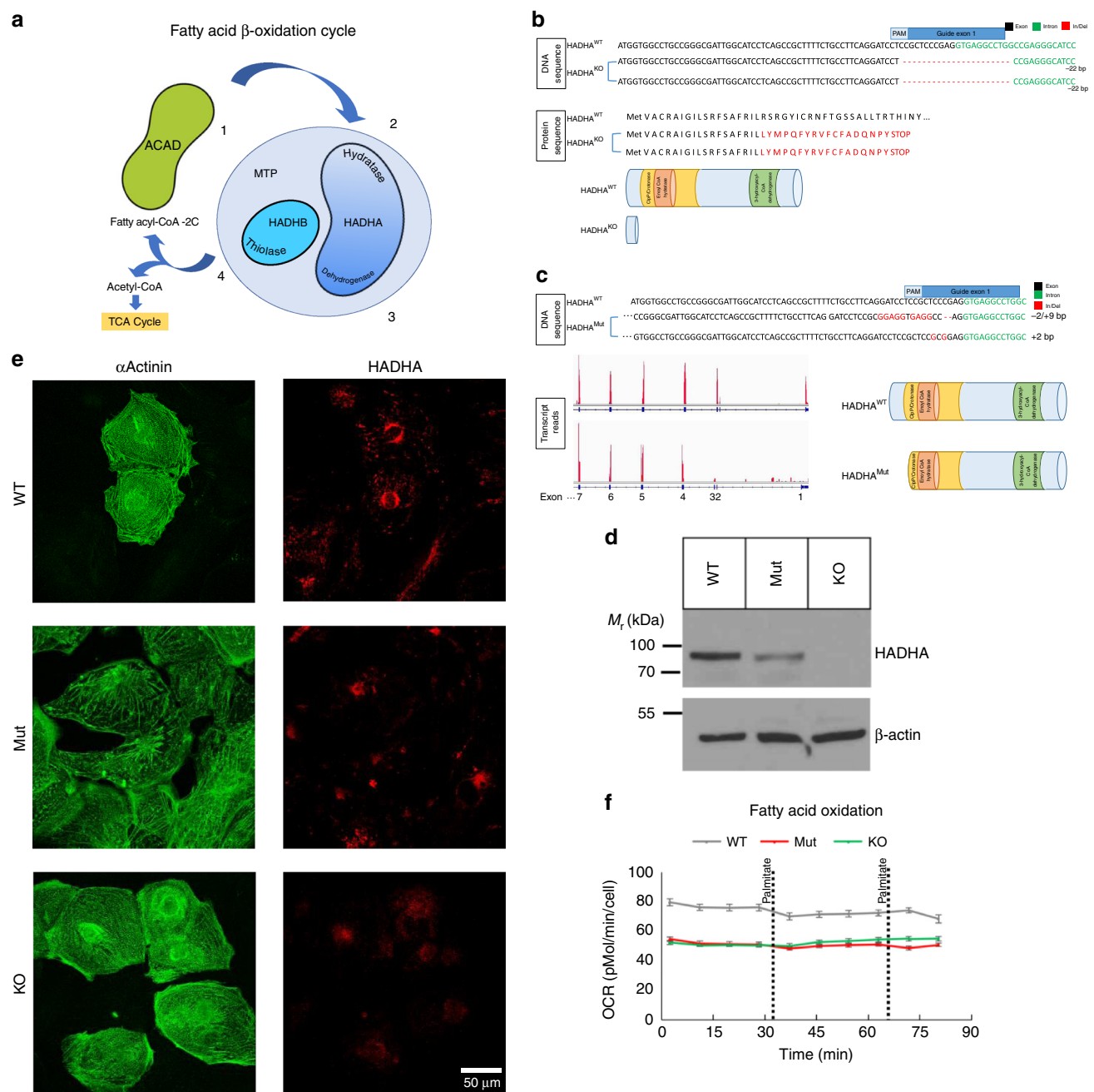

**Fig. 1** Generation of HADHA Mutant and Knockout stem cell derived cardiomyocytes. **a** Schematic of fatty acid beta-oxidation detailing the four enzymatic steps. **b** Schematic of HADHA KO DNA and protein sequence from WTC iPSC line showing a 22 bp deletion, which resulted in an early stop codon. **c** Schematic of HADHA Mut DNA and protein sequence from WTC iPSC line showing a 2 bp deletion and 9 bp insertion on the first allele and a 2 bp deletion on the second allele. RNA-Sequencing read counts show that the HADHA Mut expresses exons 4–20 resulting in a truncated protein. **d** Western analysis of HADHA expression and housekeeping protein β-Actin in WTC iPSCs. **e** Confocal microscopy of WT, HADHA Mut and HADHA KO hiPSC-CMs for the cardiac marker αActinin (green) and HADHA (red). **f** Seahorse analysis trace of fatty acid oxidation capacity of WT, HADHA Mut and HADHA KO hiPSC-CMs. n = 6–7 biological replicates. Source data are provided as a Source Data file

Fig. 2B). Finally, we assessed the metabolic capacity and found only the KO of miR-122 brought about a significant increase in maximum oxygen consumption rate (OCR) indicating more active mitochondria (Fig. 2f and Supplemental Fig. 2C).

**Bioinformatic analysis of candidate microRNAs**. RNA-Sequencing was performed after alterations of some of the miRs (miR-378e OE, −208b OE, −452 OE, −122 KO, or −205 KO) to assess their global transcriptional impact in hPSC-CMs. In each

sample, ~11,000 protein-coding genes were expressed with an aggregated expression of at least three FPKM (fragments per kilobase of transcript per million mapped reads) across all samples were used for principle component analysis (PCA). PCA showed that each miR was able to bring a significant change from their respective controls (Supplemental Fig. 2D). Furthermore, since none of the miRs clustered with one another, each miR was capable of inducing a unique expression signature.

Each miR's function was then analyzed by specifically examining pathways that are essential for cardiac maturation. A

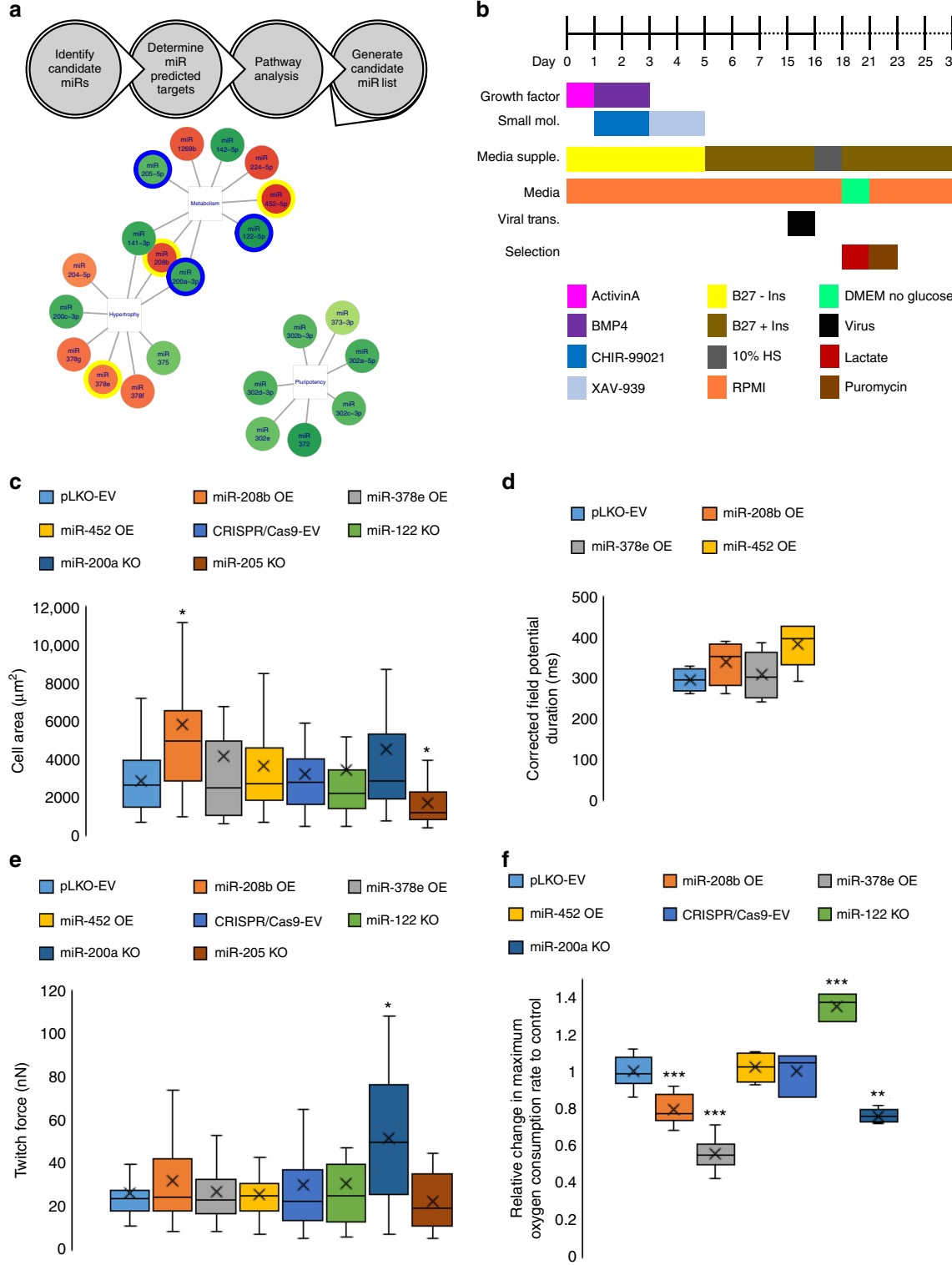

pathway enrichment heatmap was generated showing how each miR influenced seven different pathways chosen as hallmarks of cardiomyocyte maturation (Supplemental Fig. 2E).

From these data, we generated a MicroRNA Maturation Cocktail we termed MiMaC, consisting of: Let7i OE, miR-452 OE, miR-122 KO, and miR-200a KO. Let7i was chosen due to our previous study showing the potency of this miR to bring about cardiomyocyte maturation[24]. From each of the functional assays, we chose a miR that brought a significant increase in maturation

to generate a cocktail that consisted of the smallest number of miRs.

**Functional assessment of MiMaC.** To assess MiMaC treated hPSC-CM maturation we performed force of contraction, cell area, and metabolic assays (Fig. 3a). MiMaC treated hiPSC-CMs had a statistically significant increase in twitch force and power as compared to control cells (Fig. 3b–d). MiMaC treated hiPSC-CMs and hESC-CMs had a statistically significant increase in cell

**Fig. 2** Cardiomyocyte maturation microRNA screen. **a** Schematic of the workflow performed to determine candidate microRNAs to screen for cardiomyocyte maturation. **b** Schematic of the workflow performed to generate microRNA transduced stem cell derived cardiomyocytes. **c** Cell area analysis of microRNA treated hiPSC-CMs. MicroRNA-208b OE lead to a significant increase in cell area while miR-205 KO led to a significant decrease. EV: 2891 μm$^2$, 208b: 5802 μm$^2$. *$p < 0.05$, one-way ANOVA on Ranks performed. $n = 16$–$51$ cells measured. **d** Micro-electrode array analysis of microRNA treated hiPSC-CMs corrected field potential duration (cFPD). MiR-452 OE led to a longer cFPD. EV: 296 ms, 452: 380 ms. $n = 3$–$6$ biological replicates. **e** Single cell twitch force analysis using a micropost assay. MiR-200a KO led to a significant increase in twitch force of hiPSC-CMs. EV: 30.8nN, miR-200a: 51.7nN. *$p < 0.05$, one-way ANOVA on Ranks performed. $n = 12$–$41$ cells measured. **f** Seahorse analysis of the maximum change in oxygen consumption rate (OCR) due to FCCP after oligomycin treatment of microRNA treated hiPSC-CMs. MiR-122 KO led to a significant increase in maximum OCR while miR-208b OE, −378e OE and −200a KO led to significant decreases in maximum OCR. miR-122 KO: 1.35-fold change compared to EV. **$p < 0.01$, ***$p < 0.001$, one-way ANOVA performed. $n = 3$–$17$ biological replicates. Box plot middle line represents the median, x represents mean, bottom line of the box represents the median of the bottom half (1st quartile) and the top line of the box represents the median of the top half (3rd quartile). The whiskers extend from the ends of the box to the non-outlier minimum and maximum value. Source data are provided as a Source Data file

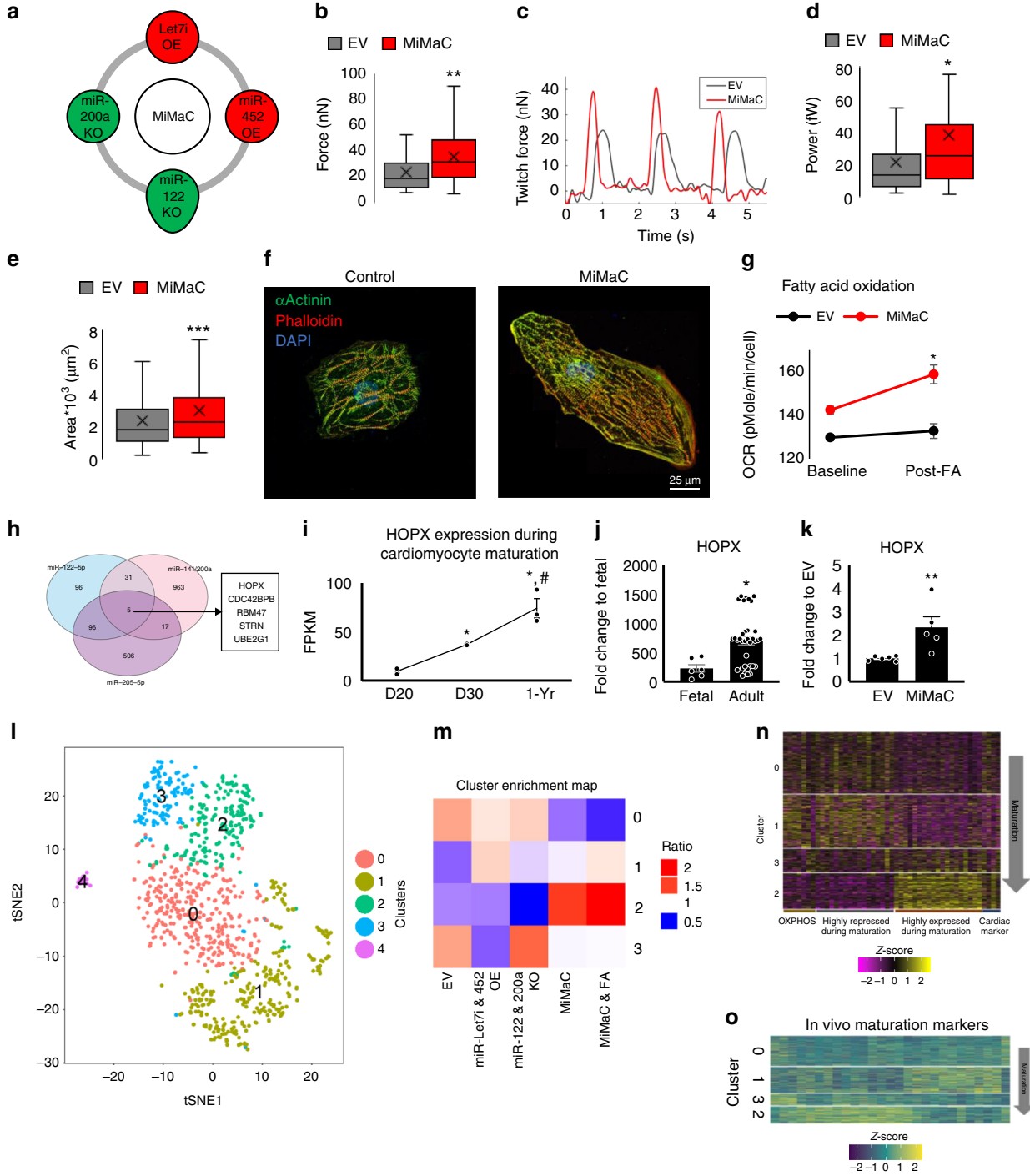

**Fig. 3** MiMaC accelerates hiPSC-CM maturation. **a** Schematic of the four microRNAs combined to generate MiMaC. **b** Single cell force of contraction assay on micro-posts showed that MiMaC treated hiPSC-CMs led to a significant increase in twitch force. EV: 24nN, MiMaC: 36nN. **\*\***$p < 0.01$, $t$-test followed by a Mann-Whitney rank sum test. $n = 40$–54 cells measured. **c** Representative trace of an EV (control) and a MiMaC treated hiPSC-CM. **d** Single cell force of contraction assay on micro-posts showed that MiMaC treated hiPSC-CMs led to a significant increase in power. EV: 22fW, MiMaC: 38fW. **\***$p < 0.05$, $t$-test followed by a Mann-Whitney rank sum test. $n = 40$–54 cells measured. **e** Cell size analysis showed that MiMaC treated hiPSC-CMs led to a significant increase in area. EV: 2389 $\mu m^2$, MiMaC: 3022 $\mu m^2$. **\*\*\***$p < 0.001$, $t$-test followed by a Mann-Whitney rank sum test. $n = 220$–298 cells measured. **f** Representative confocal microscopy images of EV and MiMaC treated hiPSC-CMs. αActinin (green), phalloidin (red) and DAPI are shown. **g** Seahorse analysis of fatty acid oxidation capacity showed that MiMaC treated hiPSC-CMs matured to a point where they could oxidize palmitate for ATP generation while controls cells were not able to utilize palmitate. MiMaC hiPSC-CMs had a significant increase in OCR due to palmitate addition. **\***$p < 0.05$, $t$-test was performed. $n = 8$–9 biological replicates. Error bars are standard error. **h** Venn diagram of KO microRNA predicted targets and the identification of HOPX as a common predicted targeted between all KO miRs screened for cardiomyocyte maturation. **i** Plot of HOPX expression from RNA-Sequence data during cardiomyocyte maturation. HOPX expression is significantly higher in D30 and 1-year hESC-CMs and 1-year hESC-CMs have significantly higher HOPX as compared to D30 hESC-CMs. \* denotes significance vs D20. # denotes significance vs D30. **\*\***$p < 0.01$ and **\***$p < 0.05$ are vs D20, #$p < 0.05$ is vs D30, one-way ANOVA was performed. $n = 2$–4 biological replicates. Error bars are standard error. **j** HOPX expression in adult human ventricle tissue is significantly higher than fetal human ventricular tissue. Plotted using RNA-sequencing data. **\***$p < 0.05$, a negative binomial test was used, $n = 6$ for fetal samples, $n = 35$ for adult samples. **k** RT-qPCR of HOPX expression showed that MiMaC treated hiPSC-CMs at D30 had a statistically significant higher level of HOPX as compared to EV control D30 hiPSC-CMs. **\*\***$p < 0.01$, $t$-test followed by a Mann-Whitney rank sum test. $n = 5$–6 biological replicates. **l** Single cell RNA-Seq tSNE plot of unbiased clustering of microRNA treated hPSC-CMs. **m** Cluster plot detailing which treatment groups are enriched in each cluster. **n** Heatmap of maturation categories based on MiMaC cluster. **o** Heatmap of in vivo human maturation markers that are upregulated with maturation (yellow). Box plot middle line represents the median, x represents mean, bottom line of the box represents the median of the bottom half (1st quartile) and the top line of the box represents the median of the top half (3rd quartile). The whiskers extend from the ends of the box to the non-outlier minimum and maximum value. Bar graphs show mean with standard error. Source data are provided as a Source Data file

area (Fig. 3e, f and Supplemental Fig. 3A). Furthermore, both MiMaC-treated hESC-CMs and miMaC-treated hiPSC-CMs were able to utilize palmitate significantly greater than control CMs (Fig. 3g and Supplemental Fig. 3B).

**Transcriptional assessment of MiMaC.** To gain a better understanding of how MiMaC was affecting the transcriptome of hiPSC-CMs we performed RNA-Sequencing comparing D30 EV control CMs to D30 MiMaC treated CMs. Pathway enrichment analysis using a hallmark gene set (Supplemental Table S1) showed that many cell maturation and muscle processes were upregulated such as: myogenesis and epithelial mesenchymal transition[37]. The top downregulated pathways were associated with cell cycle, a key feature of cardiomyocyte maturation. Using STRING Analysis, we determined the network of significantly upregulated and interconnected genes associated with two pathways: myogenesis and epithelial mesenchymal transition (Supplemental Fig. 3C). STRING analysis was also used to show that the significantly downregulated and interconnected genes were associated with repression of cell cycle, specifically, the mitotic spindle and G2M checkpoint (Supplemental Fig. 3D). These findings show that the MiMaC tool promotes a more mature transcriptome in hiPSC-CMs.

**HOPX is a regulator of CM maturation.** To better understand the molecular mechanisms that are critical for cardiac maturation, the overlapping predicted targets of the screened six miRs were determined. We had previously studied the predicted targets of Let7, insulin receptor-pathway and polycomb repressive complex 2 function, and their role in cardiomyocyte maturation[24]. We now found that all four downregulated miRs during maturation had five common predicted targets. One of these predicted targets, HOPX (Fig. 3h), is important for cardiomyoblast specification[38]. Furthermore, we have recently shown that HOPX is involved in cardiomyocyte maturation[39]. However, the regulation of HOPX expression and mechanism of HOPX action during maturation are not understood. We found HOPX expression was upregulated in vitro, in vivo and in MiMaC treated hiPSC-CMs (Fig. 3i-k). We also found HOPX was upregulated 6.8-fold in D30 miR-122 KO hiPSC-CMs while Let7i OE

matured hiPSC-CMs had no effect on HOPX expression (Supplemental Fig. 3E). These data suggest that Let7i OE maturation does not govern HOPX cardiac maturation pathways. This highlights the necessity of combining multiple miRs together to generate a robust maturation effect in hPSC-CMs.

We previously assessed the role of HOPX OE in cardiomyocyte maturation and found that HOPX OE led to an increase in CM size[39]. Using STRING analysis, we found the differentially expressed genes associated with cell division in the HOPX OE group generated a highly-interconnected network with key cell cycle genes highly downregulated (Supplemental Fig. 4A). This recapitulated the cell cycle repression we found during the in vitro CM maturation process (MiMaC treated hiPSC-CMs). We then generated four clusters using Kmeans clustering: regulation of mitotic cell cycle, cell division, inhibition of cilia and ubiquitin protein. Representative cell cycle genes, BUB1, MKI67, and CENPE, were downregulated while the inhibitor of many G1 cyclin/cdk complexes, CDKN1C, was significantly upregulated in the HOPX OE condition (Supplemental Fig. 4B). These data suggest that HOPX OE mechanistically increases cell size by driving the exit from cell cycle and inducing cardiomyocyte hypertrophy.

**HOPX regulates cell cycle via SRF genes.** HOPX is a homeodomain protein that does not bind DNA but rather is recruited to locations in the genome by serum response factor (SRF)[40]. HOPX in turn recruits histone deacetylase (HDAC) and removes acetylation marks resulting in the silencing of genes (Supplemental Fig. 4C). HOPX OE led to a significant down-regulation of 294 SRF targets (hypergeometric test p-value is $1.31 \times 10^{-5}$) (Supplemental Fig. 4D). We validated using qPCR a known SRF target gene that should be repressed during cardiomyocyte maturation, natriuretic peptide precursor A (NPPA). After 2 weeks of HOPX OE, NPPA was significantly repressed, while cardiac troponin C, a non-SRF cardiac gene was unaffected by HOPX OE. The ventricular isoform of myosin light chain, MYL2, which is upregulated as ventricular cardiomyocytes mature, increased 1.87-fold in expression after two weeks of HOPX OE (Supplemental Fig. 4E).

We determined the SRF target genes in common between HOPX OE vs. the negative control (NC) hiPSC-CMs and the

human adult vs. fetal myocardium (ventricular myocardium) transitions. 76 SRF targets were common between the two groups and formed a significant group of genes (hypergeometric test $p$-value is $5.44 \times 10^{-24}$) (Supplemental Fig. 4F) with a strong association for repression of cell cycle (Supplemental Fig. 4G). Using STRING analysis, we determined the network of connected genes out of the 76 genes in common and ran Kmeans clustering to generate four clusters (Supplemental Fig. 4H); two cell cycle clusters, DNA repair and muscle development gene clusters. The network of SRF regulated cell cycle genes that was in common between the *HOPX* OE line and adult cardiomyocytes (Supplemental Fig. 4I) showed genes associated with cell cycle with 7 of the 10 genes associated with the spindle machinery. These data indicate that MiMaC acts through HOPX to repress SRF cell cycle targets.

**scRNA-sequencing analysis of miR treated CM maturation.** Using single cell RNA-sequencing (scRNA-Seq), we utilized the MiMaC tool to provide further insight into the underlying mechanisms of cardiomyocyte maturation. We performed scRNA-Seq and unbiased clustering on five groups of miR treated CMs: EV, Let7i & miR-452 OE, miR-122 & −200a KO, MiMaC and MiMaC + FA. The enrichment of the miR perturbation was analyzed in the five identified clusters (Fig. 3l, m) using a Chi-square test. The EV group was enriched in clusters 0 and 3, Let7i and miR-452 OE group was enriched in clusters 0 and 1, miR-122 and −200a KO group was enriched in clusters 0 and 3 and MiMaC and MiMaC + FA were enriched in clusters 1 and 2. Cluster 4 mainly consisted of cells with poor read counts and was not analyzed further. Characterizing the cell fate in each subgroup showed the majority of cells were cardiomyocytes with a very small subset of cells in cluster 1 displaying fibroblast (*ENC1*, *DCN*, and *THY1*) and epicardial markers (*WT1*, *TBX18*) (Supplemental Fig. 5A).

To rank which clusters had a higher degree of cardiomyocyte maturation we assessed the genes highly up- and downregulated in the MiMaC enriched cluster, cluster 2, compared to cardiac markers, oxidative phosphorylation genes (Fig. 3n and Supplemental Table S2)[41], and in vivo human cardiac maturation markers (Fig. 3o and Supplemental Table S3). We found that cluster 2 had high upregulation of genes associated with myofibril structural proteins and in vivo maturation markers (Fig. 3n, o; $P < 2 \times 10^{-16}$, using linear mixed effects model). We also found the MiMaC-treated cells were the most mature (Supplemental Fig. 5B-E and Supplemental Table S4). Based on these findings, we ranked each cluster from least mature to most mature as cluster: 0 < 1 < 3 < 2. Cluster 2, the most mature CM cluster enriched for the MiMaC treated CMs, showed the highest expression of HOPX, a gene that is upregulated in maturation and is the predicted target of the downregulated miRs in MiMaC (Supplemental Fig. 5F and Supplemental Table S3). Importantly, these data indicate that the observed transcriptional maturation mirrors normal in vivo cardiomyocyte maturation (Fig. 3o).

Finally, we assessed the addition of fatty acids with MiMaC to increase cardiomyocyte maturation. Three long-chain fatty acids, palmitate, linoleic and oleic acid were added to the basal cardiac media used[42]. We found the MiMaC + FA cells were enriched in cluster 2. Importantly, while some studies have shown lipotoxicity with some FAs, the analysis of our carefully optimized FA-treatment procedure showed no increase in transcripts indicative of apoptosis, revealing minimal lipotoxicity in this assay (Supplemental Table S5)[43]. These data indicate that MiMaC was essential for a robust transcriptional maturation of hiPSC-CMs.

**scRNA-Seq reveals an intermediate CM maturation stage.** After unbiased analysis of the miR treated CMs it was clear each miR combination resulted in enrichment of different states of CM maturation. Interestingly, cardiomyocyte cluster 1, enriched for Let7i and miR-452 OE, showed a robust upregulation of OXPHOS and Myc target genes but was not yet significantly increased in most cardiomyocyte maturation markers (Fig. 3n, o and Supplemental Fig. 5G,H; Supplemental Table S6). Hence, treatment of Let7i and miR-452 OE created an intermediate maturity CM in which metabolic maturation was the leading force. These data suggest a possible intermediate stage is a necessary transition stage between a fetal like CM to a more mature CM, which requires transient up-regulation of OXPHOS genes.

**HADHA-Deficient CMs display reduced mitochondrial function.** The generation of the MiMaC tool allowed us to study HADHA CM disease etiology. First, we assessed the maximum OCR of WT, HADHA Mut, and KO CMs. MiMaC treated WT CMs had a statistically significant increase in maximum OCR as compared to control cells (Fig. 4a, b). Interestingly, control and MiMaC-treated HADHA Mut CMs had maximum OCR similar to control WT-CMs while the HADHA KO CMs had depressed maximum OCR. These data suggest defective mitochondrial activity of HADHA Mut and KO CMs.

We showed that only MiMaC treated WT CMs showed a statistically significant increase in oxygen consumption due to palmitate addition (Fig. 4c). WT control CMs along with control and MiMaC treated HADHA Mut and KO CMs were unable to utilize FAs. These data show that MiMaC treated CMs have the capacity to utilize long-chain FAs; however, MiMaC-treated HADHA Mut and KO CMs are unable to do so.

**Abnormal calcium handling of HADHA Mut CMs.** MTP-deficient infants can present with sudden, initially unexplained death after birth[44]. It is possible that the stress of lipids, the main substrate for ATP production found in a mother's breast-milk, is what precipitates the early infant death due to MTP deficiency. We chose to utilize a combination of three long-chain fatty acids supplemented to our base cardiac media which contains glucose (Glc + FA media): palmitate, oleic, and linoleic acid, since these FAs are the most abundant in the serum of breastfed human infants[42,45]. Palmitate, as a fatty acid substrate, is one of the most abundant fatty acids circulating during the neonatal period, representing 36% of all long-chain free fatty acids[46]. While challenging CMs with FAs can lead to lipotoxicity, we have carefully developed a concentration and combination of three fatty acids that do not result in lipotoxicity (Fig. 3l and Supplemental Fig. 6A)[42,47,48]. Moreover, other groups in the field of hPSC-CM maturation have also found that carefully chosen and fully conjugated FAs stimulate aspects of CM maturation[49].

To better understand the way in which MTP-deficient CMs may be precipitating an arrhythmic state leading to SIDS, we measured calcium transients in our WT and HADHA Mut CMs (Fig. 4d and Supplemental Fig. 6B). We found, the fold change in calcium being cycled was significantly higher in WT CMs as compared to HADHA Mut CMs (Fig. 4e). This suggested calcium was being cycled from the cytosol and stored in an aberrant manner in HADHA Mut CMs. When examining the tau-decay constant, we found HADHA Mut CMs had a higher average value (Fig. 4f). This suggested the rate at which calcium was being pumped back into the sarco/endoplasmic reticulum was slower in the HADHA Mut CMs.

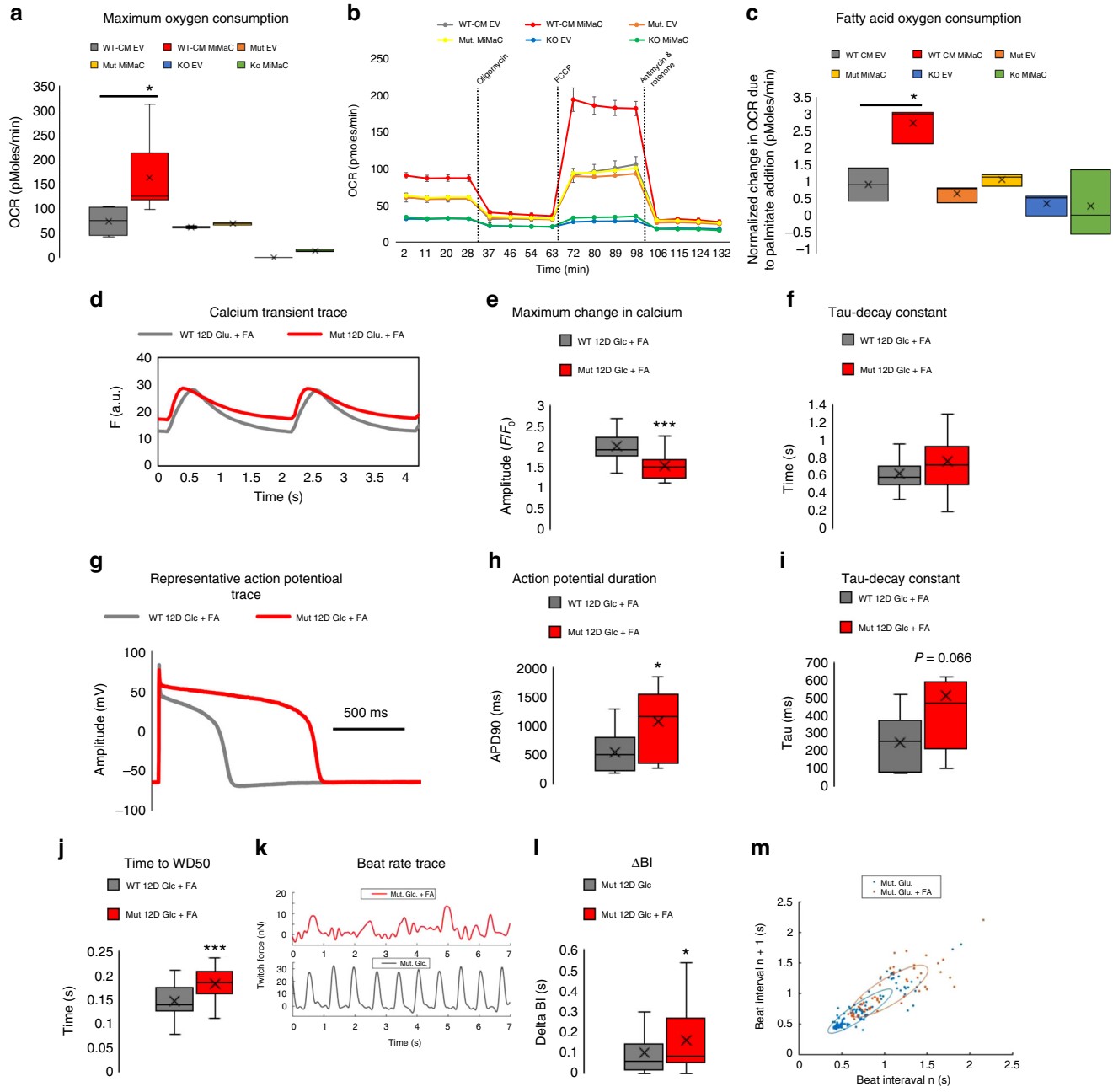

**Beat rate abnormalities in HADHA Mut CMs**. Since HADHA Mut CMs cultured in Glc + FA media exhibited abnormal calcium cycling, we assessed whether or not these CMs also exhibited abnormal electrophysiology. Using single cell whole-cell patch clamp, we found there was a significant increase in action potential duration in the HADHA Mut CMs as compared to WT CMs (Fig. 4g, h). This elongation period is seen during the plateau phase of the action potential where calcium ions are opposing the change in voltage due to potassium ions. This extent of elongation is indicative of a pathological state and suggests calcium handling as a potential source of this abnormal action potential (Fig. 4d–f). Furthermore, as in the calcium handling data, the tau-decay constant was higher in HADHA Mut CMs as compared to WT CMs (Fig. 4i). There was no difference in resting membrane potential (Supplemental Fig. 6C). Since, elongations of action potentials have been shown to result in arrhythmic heart conditions, phase 2 re-entry[50], our whole-cell patch clamp data suggest the HADHA Mut CMs may be in a pro-

arrhythmic state suggesting that indeed, HADHA Mut CMs when challenged with FAs could result in SIDS due to abnormal calcium handling.

To assess the electrophysiology of a monolayer of cardiomyocytes, as a syncytium of cardiomyocytes is more representative of "tissue level" myocardium, we determined membrane potential changes using a voltage-sensitive fluorescent dye, Fluovolt. We found that while HADHA Mut CMs had no change in the maximum change in voltage amplitude or the rate of depolarization (Fig. 4j and Supplemental Fig. 6D–H), significant differences were observed when examining repolarization rates. We found that the time to wave duration (WD) 50% (WD50) and 90% (WD90) were significantly longer in the HADHA Mut CMs as compared to WT CMs (Fig. 4j and Supplemental Fig. 6H). These data suggest that the HADHA Mut CMs, at a monolayer level, had impaired repolarization resulting in slightly longer action potentials. Both the single cell and monolayer electrophysiology data showed abnormal action potential durations, both of which

**Fig. 4** Fatty acid challenged HADHA Mut CMs displayed elevated cytosolic calcium levels leading to increased beat rate irregularities. **a** Seahorse mitostress assay to analyze maximum oxygen consumption rate after oligomycin and FCCP addition. MiMaC treated CMs showed a significant increase (2.2-fold change) in maximum OCR compared to control EV CMs. $*p < 0.05$, one-way ANOVA on ranks vs WT-CM EV was performed. $n = 1$–14 biological replicates. **b** Representative trace of the mitostress assay. **c** Seahorse analysis of fatty acid oxidation capacity showed that MiMaC treated hiPSC-CMs matured to a point where they could oxidize palmitate for ATP generation while controls cells were not able to utilize palmitate. MiMaC hiPSC-CMs had a significant increase in OCR due to palmitate addition. Both MiMaC treated Mut and KO hiPSC-CMs were unable to oxidize palmitate. $*p < 0.05$, one-way ANOVA on ranks vs WT-CM EV was performed. $n = 1$–14 biological replicates. **d** Representative trace of the change in fluorescence during calcium transient analysis. **e** Quantification of the maximum change in fluorescence during calcium transients. Mut CMs as compared to WT CMs after 12D of Glc + FA media treatment had a statistically significantly lower change in calcium. WT CM: 2.03, Mut CM: 1.55. $***p < 0.001$, t-test was performed. $n = 28$–30 cells measured. **f** Quantification of the tau-decay constant. Mut CMs as compared to WT CMs after 12D of Glc + FA media treatment had a higher tau-decay constant. WT CM: 0.63 s, Mut CM: 0.76 s. $n = 28$–30 cells measured. **g** Representative trace of the change in cell membrane potential during whole-cell patch clamp analysis. **h** Quantification of the APD90 in WT and HADHA Mut CMs after 12D of Glc + FA media. WT 541 ms, Mut 1068 ms. $*p < 0.05$, t-test was performed. $n = 9$–10 cells measured. **i** Quantification of the tau-decay constant in WT and HADHA Mut CMs after 12D of Glc + FA media. $p = 0.066$, t-test followed by a Mann-Whitney rank sum test. $n = 9$–10 cells measured. **j** Time to wave duration 50% is significantly longer in Mut CMs as compared to WT CMs after 12D of Glc + FA media treatment. $***p < 0.001$, t-test was performed. $n = 18$–36 cells measured. **k** Representative beat rate trace of Mut CM in Glc or Glc + FA media. **l** Quantification of the change in beat interval ($\Delta$BI). Mut CMs in Glc + FA media as compared to Mut CMs in Glc media had a statistically significant higher $\Delta$BI. $*p < 0.05$, t-test followed by a Mann-Whitney rank sum test. $n = 13$–16 cells measured. **m** Poincaré plot showing ellipses with a 95% confidence interval for each group. The more rounded ellipse of the Mut Glc + FA condition shows that these cells had a greater beat to beat instability as compared to Mut Glc CMs. Box plot middle line represents the median, x represents mean, bottom line of the box represents the median of the bottom half (1st quartile) and the top line of the box represents the median of the top half (3rd quartile). The whiskers extend from the ends of the box to the non-outlier minimum and maximum value. Source data are provided as a Source Data file

could be caused by abnormal calcium handling found in HADHA CMs.

We tracked the spontaneous beating of HADHA Mut CMs in the presence of FAs and found that the HADHA Mut CMs had a significantly higher beat interval (Fig. 4k; Supplemental Fig. 6I) and a significantly higher change in beat-to-beat interval ($\Delta$BI) than controls (Fig. 4l). These data suggest that HADHA Mut CMs beat on average slower and the time between beats was more variable. Furthermore, we quantified the percentage of $\Delta$BI that were greater than 250 ms and on average the HADHA Mut CMs had a higher percentage of potentially arrhythmic $\Delta$BIs (Supplemental Fig. 6J)[51]. Finally, we generated a Poincaré plot with fitted ellipses (95% confidence interval) around each group's beat interval data (Fig. 4m). A narrow and elongated ellipse suggested uniform beat intervals while a more rounded ellipse suggested beat rate abnormalities. Taking the ratio of the major to minor axis of each ellipse we found that the HADHA Mut Glc condition had a ratio of 4.36 while the HADHA Mut Glc + FA condition had a ratio of 3.12 indicating that the HADHA Mut Glc + FA condition had a more rounded ellipse, indicating a higher beat-to-beat variability. These data combined suggest that the FA treated HADHA Mut CMs enter a pro-arrhythmic state potentially due to abnormal calcium handling, which results in elongated action potentials and abnormal repolarization.

**scRNA-sequencing identifies HADHA Mut CM subpopulations.** Single-cell RNA-Sequencing and unbiased clustering were performed to better understand how the HADHA Mut CM population was behaving when challenged with FAs. A tSNE plot detailing each of the sequenced cell groups showed a clear distinction between WT and HADHA Mut CMs, with a small but significant overlap (Fig. 5a). When performing unbiased clustering, six clusters were found: 0 HADHA Mut CMs non-replicating, 1 an intermediate maturation population of WT and Mut CMs, 2 HADHA Mut CMs replicating, 3 healthy CMs, 4 fibroblast like population and, 5 epicardial like population (Fig. 5b, c and Supplemental Fig. 7A).

To assess the degree of maturation and disease state we categorized each cluster based on the key categories described above (Fig. 3n). Upregulated genes in cluster three were associated with myofibril assembly and striated muscle cell development. Interestingly, a subset of both WT and HADHA

Mut CMs were identified in an intermediate CM maturation cluster, cluster 1, as described above (Figs. 3l, 5d). This cardiac population had a high up-regulation of OXPHOS and Myc target genes (Supplemental Fig. 7B). WT cells that further developed from this intermediate state were identified in the more mature CM state, cluster 3. HADHA Mut cells, however, entered two sequential pathological states of disease. We postulate that first, HADHA Mut cells lose many highly expressed and repressed cardiac markers along with cell cycle inhibitor *CDKN1A*, as seen in cluster 0 (Supplemental Fig. 7C). Finally, very diseased HADHA Mut CMs in cluster 2 up-regulated genes that should be highly repressed in mature CMs, and activated cell cycle genes (Fig. 5d, Supplemental Fig. 7C, D and Supplemental Table S7). We benchmarked these stages of maturation and disease progression against in vivo mouse and human maturation markers and found a similar trend for maturation, disease progression and loss of cardiac identity (Fig. 5e and Supplemental Fig. 7E–G).

Examining significantly changed hallmark pathways between HADHA Mut CM clusters and the WT CM cluster we found OXHPOS, cardiac processes and myogenesis, being depressed in the mutant cells (Supplemental Table S8, S9). Furthermore, while WT CMs showed strong expression of cell cycle repressor *CDKN1A*, both HADHA Mut CM populations lost this expression. Cluster 2, the replicating HADHA Mut CMs, had an upregulation of DNA replication, G2M checkpoint and mitotic spindle genes (Supplemental Table S9). Moreover, genes that are expressed in replicating and/or endocycling cells such as *MKI67* and *RRM2* were expressed only in cluster 2 HADHA Mut CMs (Supplemental Fig. 7C). To address potential pathological outcomes of the abnormal cell cycle marker increase, we analyzed the number of nuclei per cell in HADHA Mut CMs. Importantly, we observed a significant increase of the nuclei per cell in HADHA Mut CMs as compared to WT CMs (Chi-square test $p < 0.001$) (Fig. 5f, g). The majority of WT CMs were mono- or bi-nucleated, which is the healthy state found in vivo for nuclei number in CMs[52]. However, the number of mono-nucleated HADHA Mut CMs was significantly reduced while the number of bi- and multi-nucleated HADHA Mut CMs were increased suggesting a pathological state in the HADHA Mut CMs[53]. These data support the surprisingly high cell cycle transcript expression we found in a subpopulation of HADHA Mut CMs (cluster 2),

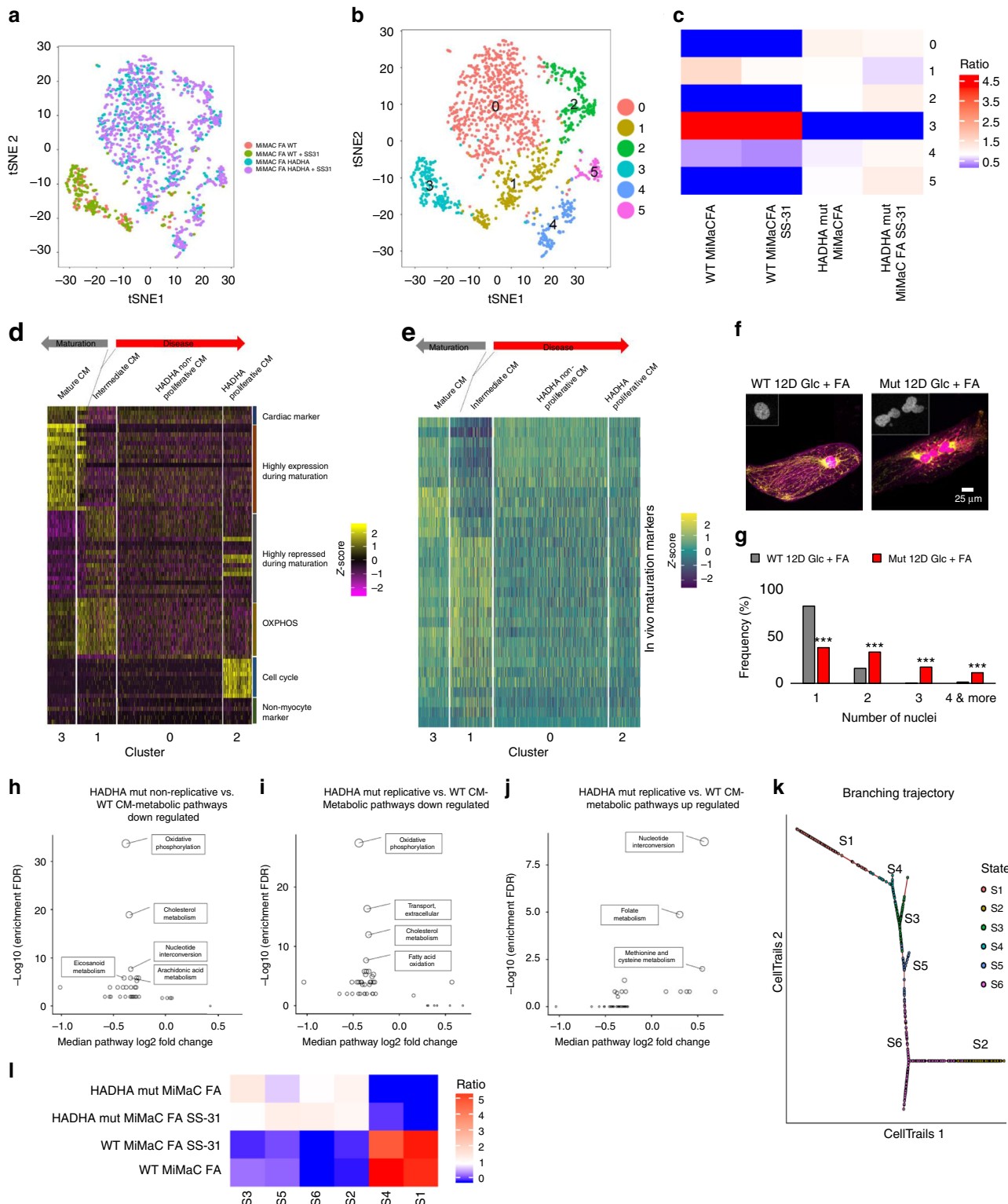

and suggest multiple stages of disease state in the HADHA Mut CMs.

To ensure cell cycle was not the underlying difference between all clusters, we examined cell cycle genes in each cluster (Supplemental Fig. 7H). Unlike previous studies which found that the bias imposed on cluster differences was dictated by which state of the cell cycle the cells were in[54], we found that only cluster 2 (Fig. 5b and Supplemental Fig. 7H) showed upregulated cell cycle genes. We also re-processed the clustering data with the removal of cell cycle genes and all clusters remained, except for original cluster 2, high cell cycle HADHA Mut CMs (Supplemental Fig. 7I). These findings suggest that cell cycle is the underlying reason for cluster 2 phenotype, but not for the rest of the cell populations (Fig. 5a, b).

Finally, we found two genes (*BAX* and *MFN2*) that were highly expressed in cluster 3, MiMaC cluster, but downregulated in cells defective for HADHA, cluster 0 (Supplemental Fig. 7J). These findings support a recent study showing that HADHA mutants

**Fig. 5** scRNA-Seq reveals multiple disease states of fatty acid challenged HADHA Mut CMs. **a** Single cell RNA-sequencing tSNE plot of WT compared to HADHA Mut CMs shows a clear distinction between these two groups. Four conditions of D30 CMs: 6 days of FA treated MiMaC WT CM, 6 days of FA and SS-31 MiMaC WT CMs, 6 days of FA treated MiMaC HADHA Mut CMs and 6 days of FA and SS-31 treated MiMaC HADHA Mut CMs. **b** Unbiased clustering revealed six unique groups. **c** Heatmap detailing the enrichment of conditions in each cluster. **d** Heatmap of maturation categories based on MiMaC cluster. **e** Heatmap of in vivo mouse maturation markers that are upregulated with maturation. **f** Confocal microscopy showing that HADHA Mut CMs have more nuclei than WT CMs. Blue—DAPI, green—ATP synthase beta subunit, and pink—Titin. Inset is of the nuclei shown in grey scale. **g** Histogram of the frequency of cells with either 1, 2, 3-, or 4 or more nuclei. HADHA mutant CMs have a significant number of cells with three or more nuclei. ***$p < 0.001$, Chi-square test with three degrees of freedom performed. $n = 150–225$ cells measured. **h** Downregulated metabolic pathways in cluster 0 (non-replicating HADHA CMs) as compared to cluster 3 (WT CMs). **i** Downregulated metabolic pathways in cluster 2 (endoreplicating HADHA CMs) as compared to cluster 3 (WT CMs). **j** Upregulated metabolic pathways in cluster 2 (endoreplicating HADHA CMs) as compared to cluster 3 (WT CMs). Metabolic bubble plot circle size is proportional to the statistical significance. The smaller the $p$-value, the larger circle. Adjusted $p$-value 0.01 used as cutoff. **k** Branching trajectory kinetics plot based on CellTrails clustering. **l** Heatmap detailing the enrichment of conditions in each state. Source data are provided as a Source Data file

have defects in mitochondrial fission and fusion machinery, specifically, they also found the gene *MFN2* to be mis-regulated leading to punctate malfunctioning mitochondria[55]. We postulate three different states of pathology in HADHA Mut CMs challenged with FAs: intermediate state(cluster1)::non-replicating CM state(cluster 0)::replicating CM state(cluster 2[56]). Cluster 1 showed an intermediate state of CM maturity, characterized by elevated OXPHOS and Myc target genes (Supplemental Table S10). Importantly, both WT and HADHA CMs are found in cluster 1, suggesting that the HADHA CMs only manifest pathological phenotypes that separate them from wild-type cells later in development, during the maturation process, similar to that seen in human development.

We performed unbiased metabolic pathway analysis, screening 68 metabolic pathways and found HADHA Mut CM clusters, 0 and 2, displayed reduced metabolic pathway gene expression in comparison to WT CM, cluster 3 (Fig. 5h, i). Specifically, OXPHOS was one of the most downregulated pathways followed by cholesterol metabolism and fatty acid oxidation. Interestingly, in cluster 2, there were two highly upregulated metabolic pathways: nucleotide interconversion and folate metabolism, two key metabolic processes involved in DNA synthesis (Fig. 5j). Since HADHA Mut CMs displayed a down-regulation of many metabolic pathways including fatty acid and OXPHOS genes, we next wanted to examine the mitochondria and myofibrils of these cells.

**Predicted cell trajectory from healthy to diseased state.** Since we had identified two different disease states, replicating/endoreplicating or intermediate state, we wondered if we could find a trajectory upon which a healthy CM follows to a disease state in an unbiased manner. To do this, we used the CellTrails[57] method as it allows for branching. Based on CellTrails clustering, we found six different states with a clear distinction between mutant and WT CMs (Supplemental Figure 8A). Clusters 1 and 4 were identified as WT cells, while clusters 2, 3, 5, and 6 were identified as HADHA Mut CMs. To identify the intermediate states, we utilized two hallmark maturation genes, one that goes up with ventricular maturation (*MYH7*) and one that goes down with ventricular maturation (*MYL7*) (Fig. 5k, l; Supplemental Figure 8B). S4, enriched for WT CMs, and S3, enriched for HADHA Mut CMs, both had low expression of the maturation gene *MYH7* and a high expression of the immature gene *MYL7*. S5, also enriched for HADHA Mut CMs, had a scattered expression of both genes, *MYL7* and *MYH7* suggesting further progression to a pathological state. These data suggest that S4 and S3 are less mature CMs, representing the previously identified intermediate CM that can transition into a normal mature CM (S1) or fall into a diseased state (S5).

Finally, our kinetics model identified two branches as end states of the pathological HADHA Mut CMs, S6 and S2 (Supplemental Fig. 8C). These two branches identified the two different HADHA Mut states from our scRNA-Seq analysis, non-replicating (S2) and replicating/endocycling (S6) HADHA Mut CMs. Together, these kinetics data in an unbiased manner, identified WT CMs as one potential starting point that led, through an intermediate CM state that had not been able to mature to a matured WT CM, to the pathological non-replicating and replicating/endocycling HADHA Mut CMs.

**Structural and mitochondrial deficiencies of HADHA Mut and KO CMs.** When HADHA Mut and KO CMs were cultured in glucose-media alone, no obvious defects were observed in HADHA Mut and KO compared to the WT CMs (Supplemental Figure 9A,B). However, when cultured 6–12 days in FA media, sarcomere and mitochondrial defects manifested in the HADHA Mut and KO CMs, while the WT CMs appeared normal (Fig. 6a, Supplemental Fig. 9C). After 12D of Glc + FA media treatment, WT CMs had healthy myofibrils while the HADHA Mut CMs showed sarcomere dissolution, as α-actinin staining became punctate and actin filaments were difficult to detect (Fig. 6a). We assessed mitochondrial health since the HADHA Mut and KO CMs were unable to process long-chain FAs by first staining for mitochondrial ATP synthase beta subunit to examine the presence of a mitochondrial network. We found that both the WT and HADHA Mut CMs had many connected mitochondria while the KO CMs, at 6D FA, had lost their mitochondrial network to small, more circular mitochondria. To assess the functionality of these mitochondria, we analyzed the mitochondrial proton gradient via mitotracker orange staining. After 12-days of Glc + FA rich media, HADHA Mut CMs had highly depressed mitochondrial membrane proton gradient (Fig. 6a, b). Using a mitochondrial calcium sequestering dye, Rhod2, we examined the relative fluorescence intensity in the mitochondria of WT and HADHA Mut CMs and found a significant decrease in colocalization and intensity of the calcium indicator dye in the HADHA Mut CMs treated for 12-days of glucose and fatty acid medium as compared to the WT CMs (Fig. 6c and Supplemental Fig. 9D).

To better assess the sarcomere and mitochondrial disease phenotype we performed transmission electron microscopy (TEM) on WT and HADHA Mut CMs after 12D of Glc + FA exposure (Fig. 6d). WT CMs showed abundant myofibrils, clear Z bands but indistinct A-bands and I-bands, and no M-lines, indicating an intermediate, normal stage of CM myofibrillogenesis. Furthermore, WT CMs showed healthy mitochondria with good cristae formation. In contrast, HADHA Mut CMs showed poor myofibrils with a disruption of Z-disk structure replaced by punctate Z-bodies[58] and disassembled myofilaments in the cytoplasm. Interestingly, HADHA Mut CM mitochondria were

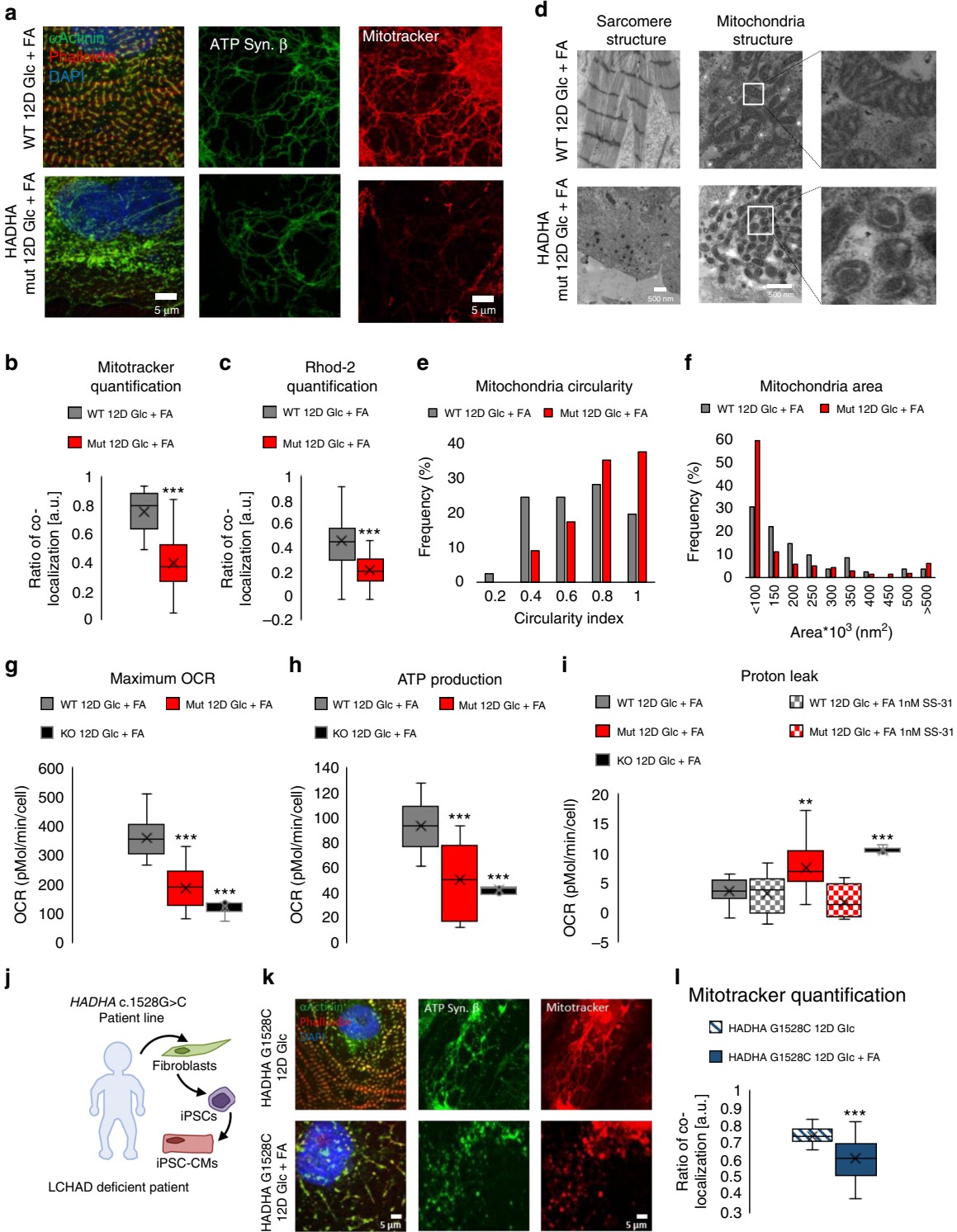

small and swollen with very rudimentary cristae morphology (Fig. 6d)[59]. Quantifying the WT and HADHA Mut CM mitochondria revealed HADHA Mut mitochondria were smaller in area and more rounded as compared to WT mitochondria (Fig. 6e, f and Supplemental Fig. 9E, F). Finally, examining complex I–V proteins showed that HADHA Mut CMs had depressed complex I–IV protein expression in Glc + FA conditions (Supplemental Figure 9G). These data show HADHA, but not control CMs lose sarcomere structure, mitochondrial membrane potential and calcium homeostasis and morphology when exposed to FAs.

**SS-31 rescues aberrant proton leak in HADHA Mut CMs.** To better understand the pathological state of HADHA Mut and KO CMs exposed to FAs we functionally assessed their mitochondria. We found that the maximum OCR of Glc + FA treated HADHA Mut and KO CMs were significantly depressed as compared to WT cells (Fig. 6g). Furthermore, HADHA Mut CMs displayed reduced oxygen-dependent ATP production (Fig. 6h) and reduced glycolytic capacity (Supplemental Fig. 6H). Since exposure to FAs led to a reduction in mitochondrial membrane potential and reduced ATP production, we postulated that this may be due in part to an increased proton leak[60,61]. We

**Fig. 6** Fatty acid challenged HADHA Mut CMs displayed swollen mitochondria with severe mitochondrial dysfunction. **a** Representative confocal images of WT and Mut CMs in 12D of Glc + FA media. **b** Quantification of mitotracker and ATP synthase β colocalization and intensity. ***p < 0.001, t-test followed by a Mann-Whitney rank sum test. n = 40 cells per group measured. **c** Quantification of Rhod-2 and ATP synthase β colocalization and intensity. ***p < 0.001, t-test followed by a Mann-Whitney rank sum test. n = 53–66 cells measured. **d** Transmission electron microscopy images of WT and Mut CMs after 12D of Glc + FA media showing sarcomere and mitochondria structure. **e** Histogram of mitochondria circularity index for WT and HADHA Mut CMs after 12 days of Glc + FA media showed HADHA Mut CMs mitochondria are rounder. n = 81–251 mitochondria measured. **f** Histogram of mitochondria area for WT and HADHA Mut CMs after 12 days of Glc + FA media showed HADHA Mut CMs mitochondria are smaller. **g** Quantification of maximum OCR from mitostress assay. Mut and KO CMs as compared to WT CMs after 12D of Glc + FA media had a significantly lower max OCR. WT CM: 359 pmoles/min/cell, Mut CM: 190 pmoles/min/cell, KO CM: 125 pmoles/min/cell. ***p < 0.001, one-way ANOVA was performed vs WT 12D Glc + FA. n = 6–19 biological replicates. **h** Quantification of ATP production from mitostress assay, calculated as the difference between baseline OCR and OCR after oligomycin. Mut and KO CMs as compared to WT CMs after 12D of Glc + FA media had significantly lower ATP production. WT CM: 93 pmoles/min/cell, Mut CM: 51 pmoles/min/cell, KO CM: 43 pmoles/min/cell. ***p < 0.001, one-way ANOVA was performed vs WT 12D Glc + FA. n = 6–18 biological replicates. **i** Quantification of proton leak from mitostress assay, calculated as the difference between OCR after oligomycin and OCR after antimycin & rotenone. Mut and KO CMs as compared to WT CMs after 12D of Glc + FA media had significantly higher proton leak. SS-31 treated Mut CMs after 12D of Glc + FA had a significantly lower proton leak and non-treated Mut CMs. WT CM: 3.64 pmoles/min/cell, Mut CM: 7.66 pmoles/min/cell, KO CM: 10.52 pmoles/min/cell. **p < 0.01, ***p < 0.001, one-way ANOVA was performed vs WT 12D Glc + FA. n = 4–19 biological replicates. **j** Schematic of patient harboring the point mutation, c.1528G > C, in the gene HADHA and the process of obtaining patient skin cells, reprograming them into iPSCs and then differentiating them into iPSC-CMs. **k** Representative confocal images of HADHA c.1528G > C CMs treated for 12-days of Glc or Glc + FA media. **l** Quantification of mitotracker and ATP synthase β colocalization and intensity. ***p < 0.001, t-test followed by a Mann-Whitney rank sum test. n = 31–35 cells measured. Box plot middle line represents the median, x represents mean, bottom line of the box represent the median of the bottom half (1st quartile) and the top line of the box represents the median of the top half (3rd quartile). The whiskers extend from the ends of the box to the non-outlier minimum and maximum value. Source data are provided as a Source Data file

found HADHA Mut and KO CMs had a significantly higher proton leak than WT CMs (Fig. 6i). Previous studies have revealed that elamipretide (SS-31), a mitochondrial-targeted peptide can prevent mitochondrial depolarization, the proton leak[62]. Interestingly, a 1 nM treatment of HADHA Mut cardiomyocytes with elamipretide (SS-31) rescued the increased proton leak in Glc + FA challenged Mut CMs (Fig. 6i). These data suggest that HADHA Mut and KO CMs exposed to FAs resulted in reduced mitochondrial capacity due in part to increased proton leak.

**HADHA c.1528G > C CMs display structural and mitochondrial abnormalities.** To test if observed HADHA Mut and KO CM disease pathology resembled the human clinical disease and to ensure the phenotype was not due to off-target gRNA mutations, we analyzed a hiPSC line that was derived from a patient that has the founder point mutation most common in mitochondrial trifunctional protein disorder, HADHA c.1528G > C (Fig. 6j and Supplemental Fig. 1C, D). We differentiated the HADHA c.1528G > C hiPSCs to cardiomyocytes to assess their sarcomere structure and mitochondria in the presence and absence of fatty acids. HADHA c.1528G > C CMs cultured in glucose have well-formed myofibrils and sarcomeres as seen by phalloidin and α-actinin staining (Fig. 6k). When HADHA c.1528G > C CMs were cultured in Glc + FAs for 12-days, we found the same sarcomere dissolution and loss of mitochondrial potential gradient, as shown via mitotracker, as we previously had shown with our HADHA Mut CM line (Fig. 6k, l and Fig. 6a, b). These data show a cardiac disease phenotype in human patient cells with the founder mutation, HADHA c.1528G > C, in a cell culture model. Furthermore, this third, independent HADHA mutant line, with a different background from our CRISPR/Cas9 modified hiPSC lines with only a single point mutation in the gene HADHA at c.1528G > C, recapitulated the disease phenotype found in our HADHA Mut and KO CMs. Consequently, the data show CRISPR/Cas9 disease phenotype phenocopies that of the human clinical phenotype.

**Loss of HADHA leads to long-chain fatty acid accumulation.** To assess the disruption of long-chain fatty acid oxidation in

HADHA Mut and KO CMs, we performed untargeted lipidomic analysis to characterize global lipidomic changes. We found an increase in long-chain acyl-carnitines in HADHA Mut and KO CMs as compared to WT CMs with no significant change in medium-chain acyl-carnitine levels (Fig. 7a, b and Supplemental Figure 10A). These data suggest that a mutation in HADHA led to an accumulation of long-chain FAs in the mitochondria. During the first step of long-chain FAO, saturated FAs will be processed into FAs with a single double bond, for instance: 14:0→14:1, 16:0→16:1 and 18:0→18:1, while unsaturated FAs, on the carboxyl end, will go through the first step of FAO and gain another double bond, for instance: 18:1→18:2 and 18:2→18:3. Accordingly, we found minimal variation in the levels of the saturated FAs: 14:0, 16:0 and 18:0 (Supplemental Fig. 10B–D) but, large increases in the abundance of 14:1, 16:1, 18:1 in the HADHA Mut and KO CMs along with slight increases in 18:2 and 18:3 in the HADHA KO CMs (Fig. 7c–e and Supplemental Fig. 7E,F). We also examined the total abundance of triglycerides, neutral lipids, in the HADHA Mut and KO CMs and found those to be significantly higher than WT CMs suggesting FAs are not being catabolized and are instead being packaged for storage due to their accumulation (Fig. 7f). These data recapitulate the in vivo findings from short lived HADHA KO mice serum levels of elevated FA species[44].

One of the hallmarks for clinically testing whether a patient has MTP deficiency is to examine if there are increased levels of hydroxylated long-chain FAs in patient blood serum or cultured fibroblasts[3,63–65]. We found a significant increase in the sum of all hydroxylated long-chain FAs in the HADHA KO CMs as compared to the WT CMs (Fig. 7g) while little increase in medium-chain hydroxylated FAs (Supplemental Fig. 10G). We next identified all hydroxylated fatty acid species found within our WT, HADHA Mut, and HADHA KO CM samples. We identified 14 hydroxylated fatty acid compounds, many more than what has been previously published for either human or mouse studies. We found the fold increase of many long-chain hydroxylated fatty acids to be significantly elevated as compared to the WT CMs (Table 1). These data re-exemplify the specificity of mutations in HADHA resulting in long-chain fatty acid disruption. Furthermore, these data match well with in vivo mouse data from short lived HADHA KO mice along with data

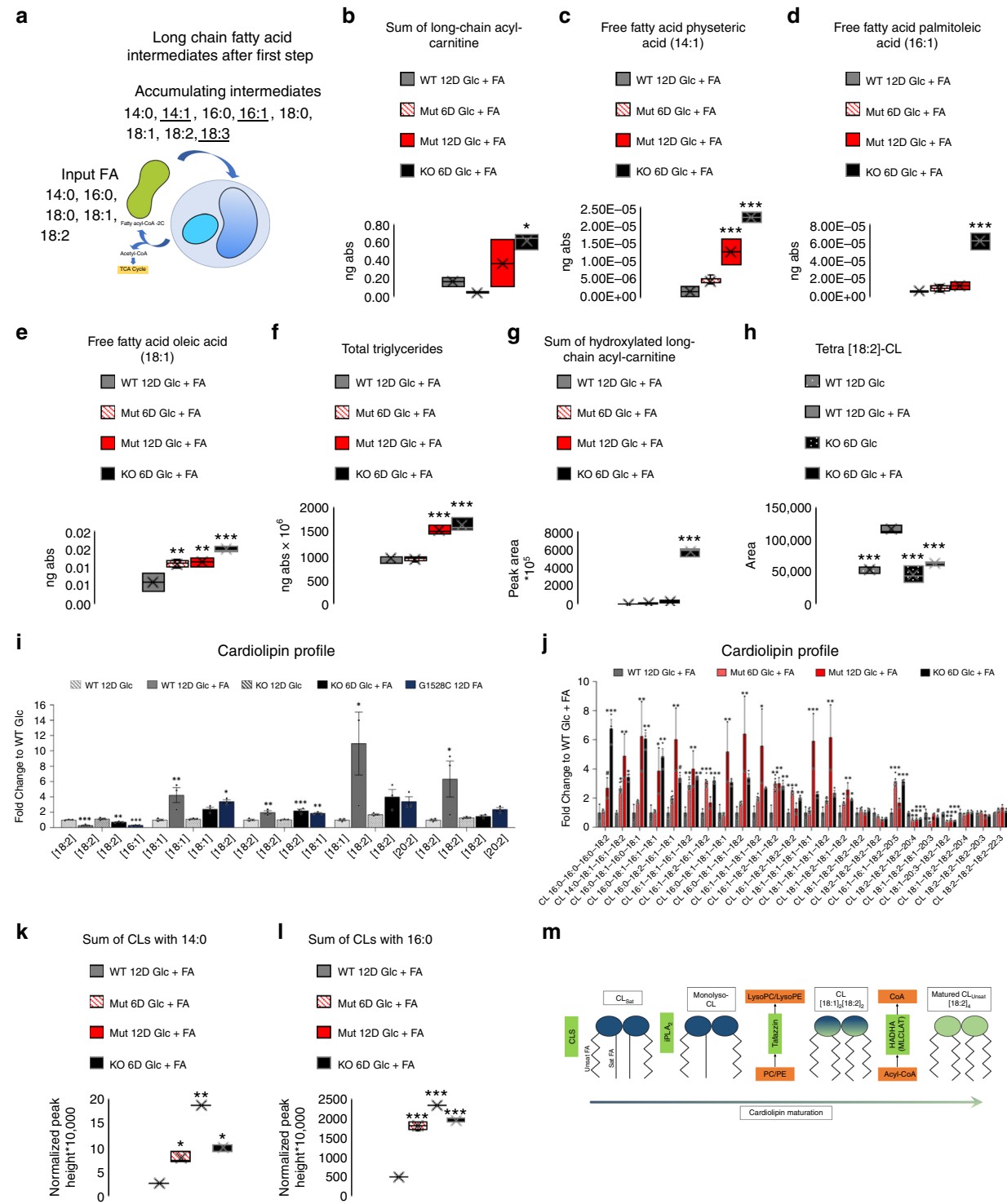

from MTP-deficient and/or LCHAD deficient patient serum and cultured fibroblasts[3,63–65].

These data show that disruption and KO of HADHA leads to a specific long-chain FA intermediate accumulation. Yet, one of the striking phenotypes we observed were rounded and collapsed mitochondria and not bursting mitochondria due to potential FA overload[66]. We therefore decided to examine another phospholipid category that regulates mitochondrial structure, cardiolipins.

**HADHA and TAZ act in series to mature cardiolipin.** Cardiolipin (CL) is a phospholipid essential for optimal mitochondrial

function and homeostasis as it maintains electron transport chain function along with other mitochondrial functions[67,68]. CL is the major phospholipid of the mitochondrial inner membrane that is synthesized in the mitochondria and is dynamically remodeled during postnatal development and disease[16,17]. The most abundant species of CL in the human heart is tetralinoleoyl-CL (tetra [18:2]-CL)[69]. In cardiac diseases such as diabetes, ischemia/reperfusion and heart failure, or due to a specific mutation in a cardiolipin remodeling enzyme tafazzin (TAZ), which leads to Barth syndrome, tetra[18:2]-CL levels are abnormal[70–73]. Using targeted lipidomics, we analyzed WT CMs supplemented with

**Fig. 7** Fatty acid challenged HADHA KO and Mut CMs have elevated fatty acids and abnormal cardiolipin profiles. **a** Model of long-chain FA intermediate accumulation after the first step of long-chain FAO due to the loss of HADHA. **b** The sum of all long-chain acyl-carnitines in WT, Mut and KO FA treated hPSC-CMs. *$p < 0.05$, one-way ANOVA was performed vs WT 12D Glc + FA. $n = 2$–6 biological replicates. **c** Amount of physeteric acid in the free fatty acid state in WT, Mut and KO FA treated hPSC-CMs. ***$p < 0.001$, one-way ANOVA was performed vs WT 12D Glc + FA. $n = 2$–6 biological replicates. **d** Amount of palmitoleic acid in the free fatty acid state in WT, Mut and KO FA treated hPSC-CMs. ***$p < 0.001$, one-way ANOVA was performed vs WT 12D Glc + FA. $n = 2$–6 biological replicates. **e** Amount of oleic acid in the free fatty acid state in WT, Mut and KO Glc + FA treated hPSC-CMs. **$p < 0.01$, ***$p < 0.001$, one-way ANOVA was performed vs WT 12D Glc + FA. $n = 2$–6 biological replicates. **f** The sum of all triglycerides in WT, Mut and KO FA treated hPSC-CMs. ***$p < 0.001$, one-way ANOVA was performed vs WT 12D Glc + FA. $n = 2$–6 biological replicates. **g** The sum of all hydroxylated long-chain acyl-carnitine species found in WT, Mut and KO Glc + FA treated hPSC-CMs. ***$p < 0.001$, one-way ANOVA was performed vs WT 12D Glc + FA. $n = 2$–6 biological replicates. **h** Relative amount of tetra[18:2]-CL in WT and HADHA KO CMs treated with either Glc or Glc + FA. ***$p < 0.001$, one-way ANOVA was performed vs WT 12D Glc + FA. $n = 3$ biological replicates. **i** Cardiolipin profile generated from targeted lipidomics for WT, HADHA KO and *HADHA* c.1528G > C CMs treated with either Glc or Glc + FA. *$p < 0.05$, **$p < 0.01$, ***$p < 0.001$, one-way ANOVA was performed vs WT 12D Glc. $n = 3$ biological replicates. **j** Cardiolipin profile generated from global lipidomics for WT CMs 12D Glc + FA, HADHA Muts CM 6D and 12D Glc + FA and HADHA KO CMs 12D Glc + FA. *$p < 0.05$, **$p < 0.01$, ***$p < 0.001$, one-way ANOVA was performed vs WT 12D Glc + FA. $n = 2$–6 biological replicates. **k** The sum of all CLs that have myristic acid (14:0) in their side chain in WT, HADHA Mut and HADHA KO CM FA treated hPSC-CMs. **$p < 0.01$, ***$p < 0.001$, one-way ANOVA was performed vs WT 12D Glc + FA. $n = 2$–6 biological replicates. **l** The sum of all CLs that have palmitic acid (16:0) in their side chain in WT, HADHA Mut and HADHA KO CM FA treated hPSC-CMs. ***$p < 0.001$, one-way ANOVA was performed vs WT 12D Glc + FA. $n = 2$–6 biological replicates. **m** Schematic diagram of how HADHA works in series with TAZ to remodel CL. Box plot middle line represents the median, x represents mean, bottom line of the box represents the median of the bottom half (1st quartile) and the top line of the box represents the median of the top half (3rd quartile). The whiskers extend from the ends of the box to the non-outlier minimum and maximum value. Bar graphs show mean with standard error. Source data are provided as a Source Data file

| Table 1 Hydroxylated FA Species Fold-Change Abundance Compared to WT 12D Glc + FA | | | | |
|---|---|---|---|---|
| Hydroxylated species | WT 12D Glc + FA | Mut 6D Glc + FA | Mut 12D Glc + FA | KO 6D Glc + FA |
| C6:0 | 1 | 1.79 | 5.82* | 2.85 |
| C8:0 | 1 | 3.61** | 9.28** | – |
| C10:0 | 1 | 2.36 | 4.49 | 1.8 |
| C12:0 | 1 | 2.70** | 5.17*** | 6.08*** |
| C14:1 | 1 | 4.84* | 16.52*** | 10.52*** |
| C14:0 | 1 | 1.98 | 6.82 | 24.15*** |
| C16:1 | 1 | 3.58 | 14.07 | 102.47*** |
| C16:0 | 1 | 3.05 | 9.97 | 189.69*** |
| C18:3 | 1 | 2.33 | 11.08* | 55.36*** |
| C18:2 | 1 | 3.72 | 10.14 | 446.42*** |
| C18:1 | 1 | 5.37 | 14.32 | 449.16*** |
| C18:0 | 1 | 3.01 | 7.63 | 209.22*** |
| C20:1 | 1 | 1.62 | 3.98 | 96.82*** |
| C20:0 | 1 | 1.37 | 6.35 | 149.29*** |
| One-way ANOVA, $n = 2$–6 biological samples. *$p < 0.05$, **$p < 0.01$ and ***$p < 0.001$ | | | | |

and without FAs. We found that FA treated WT CMs resulted in a significant increase in tetra[18:2]-CL (Fig. 7h), similar to previously observed findings during in vivo cardiomyocyte postnatal maturation[16,17]. These data show that CL maturation in cardiomyocytes can be induced in vitro. However, the HADHA KO CMs, after FA treatment, were unable to increase the amount of tetra[18:2]-CL as compared to WT FA treated CMs. Furthermore, as shown in postnatal in vivo development, WT CMs shift their CL profile to a more mature CL profile showing a significant decrease in CLs with [16:1] and increased CLs with carbons greater than 18, including the intermediate [18:1][18:2][18:2] [20:2][16]. However, HADHA KO CMs and patient derived *HADHA* c.1528G > C CMs were unable to remodel their CL profiles as efficiently as WT CMs (Fig. 7i). These data show that, surprisingly, HADHA, in addition to its role in long-chain FAO, is also required for the cardiomyocyte CL remodeling process.

Since HADHA KO CMs showed a CL remodeling defect, we next analyzed the cardiolipin species in more detail in WT, HADHA Mut and KO CMs using full lipidomics. Reinforcing our targeted lipidomics results, we found that HADHA Mut and KO CMs challenged with FAs showed an increased abundance of lighter chain CLs and a depletion of heavier chain CLs (Fig. 7j and Supplemental Figure 10H). Three CL species, tetra[18:1], [18:1][18:1][18:1][18:2] and [18:1][18:1][18:2][18:2] were significantly enriched in the HADHA Mut and KO CMs (Fig. 7j). Interestingly, [18:1][18:1][18:2][18:2] CL is specifically depleted in Barth syndrome patients who have a mutation in TAZ[74,75].

It has been previously shown that the HADHA protein has a similar enzymatic function to monolysocardiolipin acyltransferase (MLCL AT)[76,77]. MLCL AT transfers mainly unsaturated fatty acyl-chains to lyso-CL. It therefore seems plausible that HADHA has a direct role in remodeling cardiolipin to produce mature tetra[18:2]-CL species in cardiomyocytes. If TAZ and HADHA are acting in parallel to produce remodeled CL, they should both be equally depleting the MLCL pool. When TAZ is KO'd, there is a dramatic increase in MLCL, showing the direct usage of MLCL by TAZ to generate mature CL[78,79]. However, when HADHA is KO'd, there is no change in the MLCL pool (Supplemental Fig. 10I). This would suggest that HADHA does not remodel MLCL but rather CL. If TAZ and HADHA are acting in parallel, the KO of each should not result in the inverse accumulation relationship to specific CL intermediates. For instance, TAZ KO results in the decrease of [18:1][18:1][18:2] [18:2] CL[74,75]. Yet in our HADHA KO we see an accumulation of the same species. We propose that TAZ first remodels MLCL to an intermediate of CL such as [18:1][18:1][18:2][18:2] and then HADHA continues to remodel the CL species to tetra[18:2]-CL.

**Loss of HADHA function does not augment ALCAT1 function.** To garner a better understanding of how the cardiolipin profile was changing due to the lack of HADHA, we examined which new CL species became enriched in the HADHA Mut and KO CMs. CL species that had fatty acid acyl-chains of saturated fatty acids, such as 14:0 and 16:0, were enriched in the HADHA Mut and KO CMs (Fig. 7k, l). We did not identify any CL acyl-chains that had 18:0. Typically, nascent CL with multiple saturated fatty acid acyl-chains ($CL_{Sat}$), have been synthesized from cardiolipin synthase (CLS) (Fig. 7m)[80]. During the remodeling process of $CL_{Sat}$, the saturated fatty acid acyl-chains are replaced

by unsaturated fatty acid acyl-chains. Our data suggest a nascent $CL_{Sat}$ accumulation in HADHA mutants.

We next examined ALCAT1 (acyl-CoA:lysocardiolipin acyl-transferase-1) as a means for the HADHA Mut and KO CMs to utilize for CL remodeling. Since ALCAT1 has no preference for fatty-acyl substrate, it should utilize whichever fatty-acyl-CoA substrate is present[81,82]. Hallmarks of ALCAT1 activity are an increase in polyunsaturated fatty acid acyl-chains being incorporated to CL[83]. However, when we examined the CL species that had acyl-chains with fatty acids with a carbon length 20 or greater, the majority of the HADHA Mut and KO CMs actually had less species as compared to WT CMs (Fig. 7j). Furthermore, there was no increase in CL species that had multiple acyl-chains with fatty acids with a carbon length 20 or greater in any of the groups. Consequently, these data suggest that ALCAT1 is not being engaged in the HADHA Mut and KO CMs to compensate for the loss of HADHA.

## Discussion

We have generated the first human MTP-deficient cardiac model in vitro utilizing MiMaC matured hiPSC-CMs and discovered that a TFPα/HADHA defect in long-chain FAO and CL remodeling results in disease like erratic beating suggesting a pro-arrhythmic state. We further showed a mechanism of action; mutations in HADHA resulted in abnormal composition of the prominent phospholipid, CL due to HADHA's acyl-CoA transferase activity[76].

CL is important for mitochondrial architecture, it has been shown to function in organizing the cristae and electron transport chain (ETC) higher order structure, important for ETC activity, and acts as a proton trap on the outer leaflet of the inner mitochondrial membrane[84]. CMs with defective HADHA are unable to generate large amounts of tetra[18:2]-CL, due to their inability to remodel nascent CL during CM maturation. Hence the mitochondrial morphology becomes rounded in HADHA Mut, KO and c.1528G > C CMs. These data suggest that FAO phenotypes alone might not explain the defects observed in HADHA Mut, KO and *HADHA* c.1528G > C CMs and that CL remodeling is particularly important during the CM maturation process. Hence, we propose that a mutation in the HADHA enzyme during the CM maturation process results in an over accumulation of immature CL-saturated species, that may be causal for the observed mitochondrial defects and pathology (Fig. 7m).

Here we have shown that the MTP-deficient pathology in CMs leads to an abnormal cardiolipin pattern that results in severe mitochondrial defects and calcium abnormalities that pre-dispose CMs to erratic beating in HADHA Mut CMs. We identified SS-31 as a therapy to rescue the proton leak phenotype of FA challenged HADHA Mut CMs. This suggests that SS-31, or other cardiolipin affecting compounds, may serve as a potential treatment to help mitigate aspects of mitochondrial dysfunction in MTP deficiency.

## Methods

**hESC and hiPSC and cardiac differentiation**. The hESC line RUES2 (NIHhESC-09–0013, WiCell, RUESe002-A) and hiPSC line WTC #11 (Coriell Institute, GM25256), previously derived in the Conklin laboratory[85], were cultured on Matrigel growth factor-reduced basement membrane matrix (Corning) in mTeSR media (StemCell Technologies). A monolayer-based directed differentiation protocol was followed to generate hESC-CMs and hiPSC-CMs, as done previously[26]. hiPSC-CM cardiolipin assay was done with a small molecule monolayer-based directed differentiation protocol, as done previously[27]. Fifteen days after differentiation hPSC-CMs were enriched for the cardiomyocyte population using a lactate selection process[86]. We generated cardiomyocyte populations ranging from 40–60% that were then enriched to 75–80% cardiomyocytes after 4 days of lactate enrichment.

**HADHA line creation**. Using LentiCrisprV2 plasmid[87] (lentiCRISPRV2 was a gift from Feng Zhang (Addgene plasmid # 52961) two different gRNAs targeted to

Exon 1 of HADHA were designed using CRISPRscan[88]. Sequences for the gRNAs can be found in Supplemental Table S12. The gRNA and Cas9 expressing plasmids were transiently transfected into the WTC line using GeneJuice (EMD Millipore). Twenty-four hour after transfection, WTCs were puromycin selected for 2 days and then clonally expanded. DNA of the clones was isolated, the region around the targeting guides was PCR amplified (see guides in Supplemental Table S12) and sequenced to determine the insertion and deletion errors generated by CRISPR-Cas9 system in exon 1 of HADHA. Western analysis was performed to determine the levels of HADHA protein in HADHA mutants. Thirty-one clones were sent for sequencing from gRNA1 experiment, six clones (19%) had no mutations while 25 clones (81%) were found to have mutations. Twenty-four clones were sent for sequencing from gRNA2, one clone had no mutations (4%) while 23 clones (96%) were found to have mutations. Two of the mutant lines were analyzed further in this study.

**CRISPR off-target**. The potential off-targets of the HADHA gRNA were identified using Crispr-RGEN's Cas-OFFinder tool[89]. The top predicted off-targets were then amplified by GoTaq PCR and sequenced. Off-target primers can be found in Supplemental Table S13. Sequencing for primer pair #1 can be found in Supplemental Fig. 1G.

**HADHA c.1528G > C patient and the cell line**. The patient manifested the disease during the first months after birth with hypoketotic hypoglycemia and failure to thrive, with metabolic acidosis, cardiomyopathy, and hepatomegaly. The skin sampling to obtain fibroblast cultures was performed with informed consent of the parents, as approved by the ethical review board of Helsinki University Hospital, and according to Helsinki Declaration. The HEL87.1 LCHAD patient cell line was isolated from skin fibroblasts and reprogrammed, by Sendai vector-based Cyto-Tune-TM method as previously described[25]. Their stemness characteristics were confirmed by immunohistochemistry (expression of pluripotent markers OCT4, SSEA4, Tra-1–60). Quantitative PCR was used to assess the expression levels of endogenous stem cell markers and to confirm removal of transgene vectors. The karyotype was confirmed to be diploid, 46XY. The HEL87.1 iPSCs were cultured and differentiated as described above for hiPSCs and hESCs. Point mutation was confirmed via Sanger sequencing by amplifying a region around the c.1528G > C mutation using primers found in Supplemental Table 16.

**RNA extraction and qPCR analysis**. RNA was extracted from cells using Trizol and analyzed with SYBR green qPCR using the 7300 real-time PCR system (Applied Biosystems). Primers used are listed in Supplemental Table S14. Linear expression values for all qPCR experiments were calculated using the delta-delta Ct method.

**Protein extraction and western blot analysis**. Cells were lysed directly on the plate with a lysis buffer containing 20 mM Tris-HCl pH 7.5, 150 mM NaCl, 15% Glycerol, 1% Triton X-100, 1 M β-Glycerolphosphate, 0.5 M NaF, 0.1 M Sodium Pyrophosphate, Orthovanadate, PMSF and 2% SDS[90]. 25U of Benzonase Nuclease (EMD Chemicals, Gibbstown, NJ) was added to the lysis buffer right before use. Proteins were quantified by Bradford assay (Bio-rad), using BSA (Bovine Serum Albumin) as Standard using the EnWallac Vision. The protein samples were combined with the 4x Laemmli sample buffer, heated (95 °C, 5 min), and run on SDS-PAGE (protean TGX pre-casted 4%-20% gradient gel, Bio-rad) and transferred to the Nitro-Cellulose membrane (Bio-Rad) by semi-dry transfer (Bio-Rad). Membranes were blocked for 1 h with 5% milk and incubated in the primary antibodies overnight at 4 °C. The membranes were then incubated with secondary antibodies (1:10,000, goat anti-rabbit or goat anti-mouse IgG HRP conjugate (Bio-Rad) for 1 h and the detection was performed using the immobilon-luminol reagent assay (EMD Millipore). Full blots can be found in the source data file. Primary antibodies are as follows: Alpha tubulin antibody Cell Signalling Technologies (2144) 1:2000, Beta tubulin Promega (G7121) anti-mouse 1:4000, Beta Actin Cell Signalling Technologies (4970) 1:4000, HADHA Abcam (ab54477 anti-rabbit 1:1000, UCP3 Abcam (ab3477) anti-rabbit 1:200, SLC25A4 (ANT1) Sigma (SAB2105530) anti-rabbit 1:1000, OXPHOS MitoSciences (MS604/G2830) anti-mouse 1:1000, anti-GFP Invitrogen (A-11122) anti-rabbit 1:1000.

**microRNA overexpression and Knockout**. We used LentiCrisprV2 plasmid (Addgene 52961) to KO microRNAs-141, −200a, −205, and −122. gRNAs for each miR that had either the protospacer adjacent motif (PAM) NGG cut site adjacent or in the seed region of the mature microRNA were chosen to test. gRNAs can be found in Supplement Table S12. The global reduction of each miR was assessed via TaqMan RT-qPCR with probes specific against the mature form of each respective miR.

We used the pLKO.1 TRC vector (pLKO.1—TRC cloning vector was a gift from David Root (Addgene plasmid # 10878) to OE a microRNA[91]. The genomic sequence 200 bp up- and downstream of the mature microRNA was amplified and purified. Primers for each microRNA can be found in Supplemental Table S15. The amplicons were cloned between AgeI and EcoRI sites of pLKO.1 TRC vector under the human U6 promoter.

**Viral production**. HEK 293FT cells were plated one day before transfection. On the day of transfection, the OE or KO plasmid of choice was combined with packaging vectors psPAX2 (psPAX2 was a gift from Didier Trono Addgene plasmid # 12260) and pMD2.G (pMD2.G was a gift from Didier Trono Addgene plasmid # 12259) in the presence of 1 μg/μL of polyethylenimine (PEI) per 1 μg of DNA. Medium was changed 24 h later and the lentiviruses were harvested 48 and 72 h after transfection. Viral particles were concentrated using PEG-it (System Biosciences, Inc).

**hiPSC-CM transduction and selection**. hiPSC-CMs were transduced on day 14 post-induction in the presence of hexadimethrine bromide (Polybrene, 6 μg/ml). Lentivirus was applied for 17–24 h and then removed. Cells were cultured for an additional two weeks. Lactate selection was employed to obtain an enriched population of cardiomyocytes[86]. Puromycin selection was used to select for cells that have positively incorporated the vector. After 2 weeks of culture, cells were harvested for end point analysis. For the MiMaC group, hiPSC-CMs were transduced with a lower dose of the four different lentiviruses concurrently while controls were transduced with both control vectors: pLKO.1 and the Lenti-CRISPRv2 empty vector.

**Immunocytochemistry and morphological analysis**. Cells were fixed in 4%(vol/vol) paraformaldehyde, blocked for an hour with 5% (vol/vol) normal goat serum (NGS), and incubated overnight with primary antibody in 1% NGS, followed by secondary antibody staining in NGS. Measurements of CM area were performed using Image J software. Quantification of mitotracker intensity were performed using Image J software and following previously published methods on colocalization quantification[92]. Analysis was done on a Leica TCS-SPE Confocal microscope using a ×40 or ×63 objective and Leica Software. Primary antibodies used were: αActinin 1:250 Sigma A7811 anti-mouse, HADHA 1:250 abcam ab54477 anti-rabbit, ATP Synthase β 1:250 abcam ab14730 anti-mouse, Titin 1:300 Myomedix TTN-9 (cTerm) anti-rabbit, GFP 1:300 Invitrogen A-11122 anti-rabbit. Secondary antibodies and other reagents used were: DAPI at a concentration of 0.02 μg/mL, phalloidin alexa fluor 568 1:250, alexa fluor 488 or 647-conjugated goat anti-mouse and anti-rabbit secondary antibodies 1:500 (Molecular Probes). MitotrackerCMTMRos Life technologies (M7510) used at a final concentration of 300 nM in RPMI with B27 plus insulin supplement, incubated with cells for 45 min prior to fixation.

**Mitochondrial calcium**. HADHA Mut and WT CMs were plated following lactate enrichment at 20,000 cells per Matrigel-coated well in a 24 well, glass bottom plate (Cellvis) and treated with Glc + FA medium for 12 days. Cells were stained using 4.5 mM Rhod-2 (Thermofisher R1244) in DMSO and 2 nM Mitotracker green (Thermofisher M7514) in DMSO for 30 min[93]. Cells were rinsed with PBS and returned to culture medium for imaging on the heated, 5%CO2 stage of an inverted Nikon eclipse Ti equipped a Yokogawa W1 spinning disk. Colocalization analysis was performed as previously described[92].

**Micro-electrode array**. Electrophysiological recording of spontaneously beating cardiomyocytes was collected for 2 min using the AxIS software (Axion Biosystems). After raw data collection, the signal was filtered using a Butterworth band-pass filter and a 90μV spike detection threshold. Field potential duration was automatically determined using a polynomial fit T-wave detected algorithm.

**Microposts (force of contraction and beat rate)**. Arrays of polydimethylsiloxane (PDMS) microposts were fabricated as previously described[36]. The tips of the microposts were coated with mouse laminin (Life Technologies), and cells were seeded onto the microposts in Attoflour® viewing chambers (Life Technologies) at a density of approximately 75,000 per cm[2] in RPMI medium with B27 supplement and 10% fetal bovine serum. The following day, the media was removed and replaced with serum-free RPMI medium, which was exchanged every other day. Once the cells resumed beating (typically 3–5 days after seeding), contractions of individual cells were imaged (at a minimum of 70 FPS) using a Hamamatsu ORCA-Flash2.8 Scientific CMOS camera fitted on a Nikon Eclipse Ti upright microscope using a ×60 water-immersion objective. Prior to imaging, the cell culture media was replaced with a Tyrode buffer containing 1.8 mM Ca2+, and a live cell chamber was used to maintain the cells at 37 °C throughout the imaging process. A custom-written matlab code was used to track the deflection, $\Delta_i$, of each post i underneath an individual cell, and to calculate the total twitch force,

$$F_{twitch} = \sum_{i=1}^{\#posts} k_{post} \times \Delta_i^{36}, \text{ where } k_{post} = 56.5 nN/\mu m \text{ and the spacing between}$$

posts was 6 μm.

**Seahorse assay**. The Seahorse XF96 extracellular flux analyzer was used to assess mitochondrial function as previously described[24]. The plates were pre-treated with 1:60 diluted Matrigel reduced growth factor (Corning). At around 28 days after differentiation, cardiomyocytes were seeded onto the plates with a density of 50,000 cells per XF96 well. The seahorse assays were carried out 3 days after the seeding onto the XF96 well plate. One hour before the assay, culture media was exchanged for base media (unbuffered DMEM (Seahorse XF Assay Media) supplemented with

sodium pyruvate (Gibco/Invitrogen, 1 mM) and with 25 mM glucose (for MitoStress assay), 25 mM glucose with 0.5 mM Carnitine for Palmitate assay. Injection of substrates and inhibitors were applied during the measurements to achieve final concentrations of 4-(trifluoromethoxy) phenylhydrazone at 1 μM (FCCP; Seahorse Biosciences), oligomycin (2.5 μM), antimycin (2.5 μM) and rotenone (2.5 μM) for MitoStress assay; 200 mM palmitate or 33 μM BSA, and 50 μM Etomoxir (ETO) for palmitate assay. The OCR values were further normalized to the number of cells present in each well, quantified by the Hoechst staining (Hoechst 33342; Sigma–Aldrich) as measured using fluorescence at 355 nm excitation and 460 nm emission. Maximal OCR is defined as the change in OCR in response to FCCP compared to OCR after the addition of oligomycin. ATP production was calculated as the difference between the basal respiration and respiration after oligomycin. Proton leak was calculated as the difference between respiration after oligomycin and after antimycin and rotenone. Cellular capacity to utilize palmitate as an energy source was calculated as the difference between the average OCR after second palmitate addition and the final respiration value before the second addition of palmitate. The reagents were from Sigma, unless otherwise indicated.

**Whole-cell patch clamp analysis**. WT and Mut CMs were plated as single cells onto Matrigel-coated glass coverslips and treated with Glc + FA medium for 12 days. Whole-cell patch clamp recordings were performed on an inverted DIC microscope (Nikon) connected to an EPC10 patch clamp amplifier and computer running Patchmaster software (HEKA). Coverslips were loaded onto the stage and bathed in a Tyrode's solution containing 140 mM NaCl, 5.4 mM KCl, 1.8 mM CaCl2, 1 mM MgCl2, 10 mM glucose, and 10 mM HEPES. The intracellular recording solution (120 mM L-aspartic acid, 20 mM KCl, 5 mM NaCl, 1 mM MgCl2, 3 mM Mg2+-ATP, 5 mM EGTA, and 10 mM HEPES) was loaded into borosilicate glass patch pipettes (World Precision Instruments). Patch pipettes with a resistance in the range of 2–6 MΩ were used for all recordings. Offset potentials were nulled before formation of a gigaΩ seal and fast and slow capacitance was compensated for in all recordings. Membrane potentials were corrected by subtraction of a 20 mV tip potential, calculated using the HEKA software. Voltage-clamp and current-clamp experiments were then performed. To generate a single action potential, a 5 ms depolarizing current pulse of sufficient intensity was applied in current-clamp mode. Inward and outward currents were evoked by a series of 500 ms depolarizing steps from −120 to +70 mV with +10 mV increments in voltage-clamp mode. Gap-free recordings of spontaneous activity of patched cardiomyocytes were performed for 30 s with 0 pA current injection to provide a measure of the maximum diastolic potential (resting membrane potential) held by the cell without current input. All recordings and analyses were performed using the HEKA Patchmaster software suite.

**RNA-sequencing**. Day-30 hiPSC-CMs were harvested for RNA preparation and genome wide RNA-seq (>20 million reads). RNA-seq samples were aligned to hg19 using Tophat, version 2.0.13[94]. Gene-level read counts were quantified using htseq-count[95] using Ensembl GRCh37 gene annotations. Genes with total expression above 1 normalized read count across RNA-seq samples in each binary comparison were kept for differential analysis using DESeq[96]. Princomp function from R was used for Principal Component Analysis. TopGO R package[97] was used for Gene Ontology enrichment analysis. To assess the effects of miR perturbation on cardiac maturation pathways, each condition was compared against their empty vector (EV), and upregulated genes (>1.5-fold change) and downregulated genes were identified (<−1.5-fold change). A hypergeometric test was performed on up- and downregulated genes separately for enrichment against a curated set of pathways that are beneficial for cardiac maturation, resulting in a m by n matrix, where m is the number of pathways (m = 7) and n is the number of conditions (n = 6, including EV). The negative $\log_{10}$ of the ratio between enrichment p-value for up- and downregulated genes were calculated to represent the overall net "benefit" of a treatment: large positive value (>0) means the treatment results in more upregulation of genes in cardiac maturation pathways than downregulation of these genes, and more negative values means the treatment results in more downregulation of genes in cardiac maturation pathways. Human fetal and adult ventricular data was obtained from the Roadmap Epigenomics Consortium[98].

**Single-cell RNA-sequencing**. Raw single-cell RNA-seq data are processed through the CellRanger pipeline from 10X Genomics. Output of the CellRanger pipeline is further analyzed using Seurat R package[99]. Cells with more than 40% of reads mapped to mitochondrial genes, less than 200 detected genes or less than 2000 Unique Molecular Identifiers (UMIs) are removed. Remaining cells are scaled by number of UMIs and % mapped to mitochondrial genes. Parameters for tSNE analysis of maturation single cell RNA-seq data were 2905 top variable genes, top 10 principal components, and resolution 0.5. Parameters for tSNE analysis of HADHA mutant single cell RNA-seq data were 3375 top variable genes, top 10 principal components, and resolution 0.4. Cell cycle genes from Kowalczyk et al[54]. and the CellCycleScoring function in the Seurat package were used to assess the effects of cell cycle on clustering. Genes detected in at least 25% of cells in either cluster and have false discovery rate <0.1 are defined as differentially expressed. Expression values are normalized for each gene across all cells plotted in the heat maps (i.e., Z-scores). Human in vivo maturation markers are based on genes

upregulated in adult heart compared to fetal heart in the Roadmap Epigenomics Project[98]. Mouse in vivo maturation markers are based on genes upregulated in the in vivo cardiomyocyte single cell RNA-seq data from Delaughter et al.[41]. We also used PCA projection to assess the maturity of our MiMac single cell RNA-seq data to single cell RNA-seq of human fetal heart development[100]. We selected genes significantly higher in adult compared to fetal using DESeq (2 fold higher in adult, FDR < 0.05). We then intersected these genes with the top 30 most highly expressed genes in each scRNA-seq cluster to get the final gene list for heatmap in Fig. 3o. Gene Ontology enrichment is performed using the TopGO package[97]. We used the CellTrails method[57] to further dissect the dynamics of HADHA perturbation. CellTrails is a new algorithm that uses lower-dimensional manifold learning for de novo chronological ordering. The parameter for gene selection were keeping genes with fano factor >1.2; parameters for clustering were min_size = 0.02, min_feat = 10, max_pval = 1e-2, min_fc = 1.5 (each cluster should include at least 2% of all cells, and contain at least 10 genes that are expressed 1.5 fold higher in that cluster compared to others). The yEd graph editor (https://www.yworks.com/products/yed) was used to generate visualization of single cell dynamic trails, as recommended by the CellTrails algorithm.

**Calcium transient analysis method**. Cardiomyocytes were plated on Matrigel-coated round glass coverslips. The cardiomyocytes were incubated for 25 min at 37 °C with 1 mM Fluo-4 AM (Life Technologies, F14201) in Tyrode's buffer (1.8 mM $CaCl_2$, 1 mM $MgCl_2$, 5.4 mM KCl, 140 mM NaCl, 0.33 mM $NaH_2PO_4$, 10 mM HEPES, 5 mM glucose, pH to 7.4). The substrate was then transferred to a 60 mm Petri dish fresh with pre-warmed Tyrode's buffer for imaging. Samples were imaged using a Hamamatsu ORCA-Flash2.8 Scientific CMOS camera fitted on a Nikon Eclipse Ti upright microscope. Videos were taken with a ×40 water-immersion objective at a framerate of at least 20 frames per second. The fluorescence power was adjusted to ensure adequate capture of fluorescence change during depolarization without bleaching, and the same fluorescence power was used for all experiments. The cardiomyocytes were biphasically stimulated at 5 V/cm with carbon electrodes (Ladd Research, 30250) at either 0.5 Hz or 1 Hz, and at least five beats were captured during each video for analysis.

Videos were analyzed with a custom MATLAB code; calcium transients were obtained finding the cell boundary and averaging the fluorescence within the boundary for each video frame. The background fluorescence was determined automatically for each video frame and subtracted from the calcium transients. The calcium transients were then analyzed to find the peak fluorescence (F), baseline fluorescence ($F_0$), time to peak ($T_{peak}$), and time to 50 and 90% relaxation ($T_{50R}$, $T_{90R}$). The rates to peak, 50%, and 90% relaxation ($R_{peak}$, $R_{50R}$, $R_{90R}$) were calculated by dividing the respective fluorescence change by the respective time. An exponential decay function ($e^{-t/\tau}$) was fit to the relaxation between 10 and 90% relaxation to determine the relaxation coefficient, $\tau$. All of these measurements were obtained for at least four beats in each video and averaged for comparison.

**TEM**. Cells were fixed in 4% glutaraldehyde in sodium cacodylate buffer, post fixed in osmium tetroxide, en bloc stained in 1% uranyl acetate, dehydrated through a series of ethanol, and embedded in Epon Araldite. 70 nm sections were cut on a Leica EM UC7 ulta microtome and viewed on a JEOL 1230 TEM.

**Glucose and fatty acid media**. The base media, which we are calling Glucose Media, is RPMI supplemented with B27 with insulin. The fatty acid media is the glucose media with oleic acid conjugated to BSA (Sigma O3008): 12.7 µg/mL, linoleic acid conjugated to BSA (Sigma L9530): 7.05 µg/mL, sodium palmitate (Sigma P9767) conjugated to BSA (Sigma A8806): 52.5 µM and L-carnitine: 125 µM. Fatty acid (FA) experiments used this B27 + insulin + the three FAs (oleic acid, linoleic acid and palmitic acid), in RPMI media. This media was changed every other day during the 6-days or 12-days of treatment.

**Elamipretide (SS-31)**. SS-31 came from Stealth BioTherapeutics and was dissolved in PBS. A final concentration of 1 nM was used in experiments.

**Box plots**. The 'x' in each box plot denotes the average value while the horizontal bar denotes the median value, no outlier values are shown. * denotes $P < 0.05$.

**Bar graphs**. Bar graphs show the mean with error bars showing standard error.

**STRING analysis**. Protein association maps were generated using STRING version 10.5. In each diagram, genes connected to one another have an association with one another. There are three action effects: arrow – > positive,–| — negative and line with a circle on the end—unspecified. There are also eight different action types that are denoted by line color: green—activation, blue—binding, cyan—phenotype, black—reaction, red—inhibition, purple—catalysis, pink—post-translational modification, and yellow—transcriptional regulation. Kmeans clustering was used to identify the significantly changed genes due to MiMaC for: muscle structure development and extracellular matrix organization. Markov Clustering Algorithm (MCL) was used to identify genes MiMaC had downregulated to control cell division.

**Statistical ānalysis**. *P*-values were calculated using student t-test or one-way ANOVA. For one-way ANOVA analysis that failed the normality test, ANOVA a Kruskal-Wallis one-way ANOVA of Variance on Ranks was performed. All statistical tests used an α = 0.05.

**Targeted cardiolipin analysis Using LC-MS/MS**. We used WT hiPSC-CMs treated for 12D Glc + FA media and HADHA Mut hiPSC-CMs treated for 6D and 12D Glc + FA media. Immediately before extraction, each cell pellet was dissolved in 40 µL DMSO and the membranes were disrupted by sonication. Cells were subjected to sonication using 3 cycles consisting of 20 s on, 10 s off. Care was taken to keep the cells on ice during sonication. After shaking, the suspension was transferred into a 2 mL glass LC vial.

For cardiolipin extraction, an extraction mixture consisting of 20 mL chloroform/methanol mix (2:1 v/v) and 30 µL internal standard solution (5 mg PC (18:0/18:1(9Z)) (Avanti Polar Lipids, Inc., Alabaster, AL) was prepared. Next, 600 µL of the extraction mixture was added to the samples, followed by vortexing and incubation at −20 °C for 20 min. The samples were then sonicated in an ice bath for 15 min. Purified water (100 µL) was added, and the samples were shaken for 30 min at room temperature. After centrifugation at 12,000 × g for 10 min at 4 °C. The bottom phase was transferred to a new glass LC vial and dried under vacuum. The residue was then reconstituted by adding 150 µL acetonitrile/isopropanol/$H_2O$ (65:30:5, v/v/v), and centrifuged at 20,000 × g for 10 min at 4 °C. The supernatant was transferred to individual glass vials for MS analysis.

For targeted cardiolipin measurements, 2 µL of each prepared sample was injected into a 6410 Agilent Triple Quad LC-MS/MS system for analysis using an electrospray ionization source and negative ionization mode. Chromatographic separation was achieved on an Agilent 300 SB-C8 RRHD column (1.8 µm, 2.1 × 50 mm). The mobile phase A was 10 mM ammonium acetate in acetonitrile/$H_2O$ (6:4, v/v), and mobile phase B was 10 mM ammonium acetate in isopropyl alcohol/acetonitrile/$H_2O$ (90:10:1, v/v/v). The mobile phase composition changed from 60% A to 1% A over the 12 min separation, followed by a rapid increase to 60% A and equilibration to prepare for the next injection. The total experimental time for each injection was 20 min. The flow rate was 0.26 mL/min, the auto-sampler temperature was 4 °C, and the column compartment temperature was set to 55 °C. Targeted MS/MS data were acquired using multiple-reaction-monitoring (MRM) mode. MassHunter Workstation Software Quantitative Analysis for QQQ B.07.00 (Agilent) was used to integrate extracted MRM peaks.

**Untargeted lipidomic analysis**. One million cells were extracted with 225 µl of methanol at −20 °C containing an internal standard mixture of PE(17:0/17:0), PG (17:0/17:0), PC(17:0/0:0), C17 sphingosine, ceramide (d18:1/17:0), SM (d18:0/17:0), palmitic acid-$d_3$, PC (12:0/13:0), cholesterol-$d_7$, TG (17:0/17:1/17:0)-$d_5$, DG (12:0/12:0/0:0), DG (18:1/2:0/0:0), MG (17:0/0:0/0:0), PE (17:1/0:0), LPC (17:0), LPE (17:1), and 750 µL of MTBE (methyl tertiary butyl ether) (Sigma–Aldrich) at −20 °C containing the internal standard cholesteryl ester 22:1. Cells were vortexed for 20 sec, sonicated for 5 min and shaken for 6 min at 4 °C with an Orbital Mixing Chilling/Heating Plate (Torrey Pines Scientific Instruments). Then 188 µl of LC-MS grade water (Fisher) was added. Samples were vortexed, centrifuged at 14,000 rcf (Eppendorf 5415D). The upper (non-polar, organic) phase was collected in two 350 µL aliquots and evaporated to dryness. One organic phase aliquot was re-suspended in 100 µL of methanol:toluene (9:1, v/v) mixture containing 50 ng/mL CUDA ((12-[[(cyclohexylamino)carbonyl]amino]-dodecanoic acid) (Cayman Chemical). Samples were then vortexed, sonicated for 5 min and centrifuged at 16,000 rcf and prepared for lipidomic analysis. Method blanks and pooled human plasma (BioreclamationIVT) were included as quality control samples. WT FA CM and HADHA Mut 12D FA were $N = 2$ with each N being a pool of 1–3 samples. Mut 12DFA #1, #3 and #4 were averaged as 1 sample and WT 12DFA #1, #2 and #4 were averaged as 1 sample.

HADHA KO CM were $N = 3$ and HADHA Mut 6D FA was $n = 6$ where $N = 2$ with 3 technical replicates per N.

**Chromatographic and mass spectrometric conditions for lipidomic RPLC-QTOF analysis**. Re-suspended samples were injected at 3 µL and 5 µL for ESI positive and negative modes respectively, onto a Waters Acquity UPLC CSH C18 (100 mm length × 2.1 mm id; 1.7 µm particle size) with an additional Waters Acquity VanGuard CSH C18 pre-column (5 × 2.1 mm id; 1.7 µm particle size) maintained at 65 °C which was coupled to a Vanquish UHPLC System. To improve lipid coverage, different mobile phase modifiers were used for positive and negative mode analysis[101]. For positive mode 10 mM ammonium formate and 0.1% formic acid were used and 10 mM ammonium acetate (Sigma–Aldrich) was used for negative mode. Both positive and negative modes used the same mobile phase composition of (A) 60:40 v/v acetonitrile:water (LC-MS grade) and (B) 90:10 v/v isopropanol:acetonitrile. The gradient started at 0 min with 15% (B), 0–2 min 30% (B), 2–2.5 min 48% (B), 2.5–11 min 82% (B), 11–11.5 min 99% (B), 11.5–12 min 99% (B), 12–12.1 min 15% (B), and 12.1–15 min 15% (B). A flow rate of 0.6 mL/min was used. For data acquisition a Q-Exactive HF Hybrid Quadrupole-Orbitrap Mass Spectrometer was used with the following parameters: mass range, $m/z$ 100–1200; $MS^1$ resolution 60,000: data-dependent $MS^2$ resolution 15,000; NCE 20, 30, 40;

4 targets/MS[1] scan; gas temperature 369 °C, sheath gas flow (nitrogen), 60 units, aux gas flow 25 units, sweep gas flow 2 units; spray voltage 3.59 kV.

**LC-MS data processing using MS-DIAL and statistics**. Untargeted lipidomic data processing was performed using MS-DIAL[102] for deconvolution, peak picking, alignment, and identification. In house m/z and retention time libraries were used in addition to MS/MS spectra databases in msp format[103]. Features were reported when present in at least 50% of samples in each group. Statistical analysis was done by first normalizing data using the sum of the knowns, or mTIC normalization, to scale each sample. Normalized peak heights were then submitted to R for statistical analysis. ANOVA analysis was performed with FDR correction and post-hoc testing.

**Reporting summary**. Further information on research design is available in the Nature Research Reporting Summary linked to this article.

## Data availability

All of our RNA-Seq (bulk and single cell) can be found at the following GEO Submission number: GSE135595. Full lipidomics data can be found at the NIH Metabolomics Workbench Study ID: ST001246.

The source data underlying Figs. 1D, 1F, 2C-F, 3B, D, E, G, I-K, 4A, C, E, F, H, I, J, L, 5G, 6B, C, E-I, L, 7B-L and Supplementary Figs 1A, B, 3A, B, E, 4B, E, 6C, E-J, 9E-H, 10A-I.

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

## Acknowledgements

We thank members of the Ruohola-Baker laboratory for helpful discussions throughout this work. This work is supported by grants from the National Institutes of Health R01GM097372, and R01GM083867 for HRB, 1P01GM081619 for C.M. and H.R.B., R01HL146436 and R01HL135143 for H.R.B. and D.H.K. and the NHLBI Progenitor Cell Biology Consortium (U01HL099997; UO1HL099993) for C.M. and H.R.B. National

Science Foundation grants CBET-1509106 and CMMI-1661730 for N.S.J. A.S. was supported by the Academy of Finland and Finnish Foundation for Cardiovascular Research. We would like to thank: Dr. David Marcinek for the SS-31, BGI for their sequencing services, Bruce Conklin (UCSF, Gladstone Institute) for the WTC CRISPRi hiPSCs and pQM plasmid backbone, the Vision Core for their TEM services (P30 EY001730) and Professor Vockley and Dr. El-gharbawy for helpful discussions and advice. Scholarship support from the Wellstone Muscular Dystrophy Cooperative Research Center: U54AR065139 and the NSERC Alexander Graham Bell Graduate Scholarship for J.W.M. A.L. was supported by National Institute of Health grants F32 HL126332. S.L. was supported by the WRF Posdoctoral Fellowship Program and E.C. was supported by an NSF Graduate Research Fellowship.

## Author contributions

Conceptualization, J.W.M. and H.R.B.; methodology, H.R.B., J.W.M., A.L., N.J.S., K.B., Y.W., and J.M.; validation H.R.B. and J.W.M.; formal analysis, J.W.M., A.L., K.B., Y.W., and T.M.; investigation, J.W.M, E.C, M.R.S., D.R., S.L., D.D., A.T.S, A.L., K.B., and Y.W., resources, J.W.M., H.R.B., P.H., X.Y., A.M., N.J.S., C.E.M., and A.S.; writing – original draft, J.W.M. and H.R.B.; writing—review and editing, J.W.M., S.L., D.D., A.L., K.B., P.H., X.Y., J.M., M.R.S., O.F., A.M., D.H.K., C.E.M., N.J.S., Y.W., and H.R.B.; visualization, J.W.M. and H.R.B.; supervision, H.R.B.; project administration, J.W.M. and H.R.B.; funding acquisition, H.R.B.

## Competing interests

N.J.S. is a co-founder, board member, and has equity in Stasys Medical Corporation. He is also a scientific advisor and has equity in Nanosurface Biomedical Inc. C.E.M. is a scientific founder and equity holder in Cytocardia. D.H.K. is a scientific founder and equity holder in NanoSurface Biomedical Inc. J.W.M was a paid consultant of Nanosurface Biomedical Inc. H.R.B. is a member of the Nanosurface Biomedical Inc. scientific advisory board.
