## [Peer Review File · Nature Communications]

Reviewers' comments:

Reviewer #1 (Remarks to the Author):

General concerns

In this study Miklas et al. reported a miRNA cocktail (MiMaC) promoting maturation of human iPSC-derived cardiomyocytes (CM), and defined its target HOPX essential for CM maturation. They generated 3 different hiPSC lines including wild type, HADHAMut, and HADHAKO using CRISPR/Cas9 system, and induced CM differentiation and maturation to explore the mechanism of a human disease sudden infant death syndrome with mitochondrial trifunctional protein deficiency. The authors showed the mutant iPSC-CM has defect in fatty acid beta-oxidation, reduced mitochondrial proton gradient, disrupted cristae structure and defective cardiolipin remodeling. Although several high-throughput screenings were used to determine the microRNA candidates and their targets for CM maturation, similar works have been reported by other groups and thus the novelty of the current study may be a concern. Furthermore, the quality of data presented should be improved, there lacks decisive evidence for concluding HOPX as a key regulator of iPSC-CM maturation, and there is no data to support peptide SS-31 as a treatment for stabilizing cardiolipin.

Specific comments:

1. The maturity of iPSC-CM should be further characterized especially the electrophysiology, given that infant sudden death is typically caused by arrhythmia. At least, whole cell patch clamp for calcium channels should be performed in the HADHA iPSC-CMs.
2. In screening candidate miRNAs for CM maturation (which has been reported by several groups), the authors stated that some candidates targeting the same seed region thus choosing miR-378e from the miR-378 family and miR-200a instead of miR-141. Why is that? A random selection? How did the authors determine which one from the microRNA families to be essential for CM maturation?
3. The authors stated that different miRNAs regulate different stages of CM maturation. Is it possible that treatment of different miRNAs at different time points may lead to different degrees of maturity? Also, why miR-208b was excluded from the MiMaC, given that miR-208b OE enhances iPSC-CM cell size and twitch force as shown in Figure 2? How about Let-7i?
4. The authors should carefully define the status of so-called "intermediate maturity of iPSC-CM" including the morphological, electrophysiological and functional changes. Also, how does the treatment of Let-7i and miR-452 OE lead to the intermediate status?

5. The authors showed changes of mitochondrial structure and function in HADHA mutant cells. Did the authors check mitochondrial fusion, fission and mitophagy as well?
6. Quality of data presentation should be improved. For example, the labels in some figures are either not shown up or appear blurry such as in Fig 1B, 1C, Fig 2a, Fig 3H, 3N, Fig 4B & Fig 5A.
7. There lacks conclusive evidence to demonstrate HOPX as a novel regulator for iPSC-CM maturation.
8. The authors should provide data to conclude that HADHA is required for fatty acid beta-oxidation in their iPSC-CMs. The therapeutic effect of SS-31 should be tested as well.

Reviewer #2 (Remarks to the Author):

NCOMMS-18-29739-T analysis

Mitochondria trifunctional protein (MTFP) is a heteroctomer; two genes encode the alpha and beta subunits of MTFP, respectively, hydratase subunit A (HADHA) and hydratase subunit B (HADHB). MTFP deficiency results of impaired fatty acid oxidation (FAO) due to mutations in hydroxyacyl-CoA dehydrogenase/3-ketoacyl-CoA thiolase/enoyl-CoA hydratase subunit A (HADHA/LCHAD) or subunit B (HADHB). MTFP-deficient newborns developed arrhythmias and sudden infant death syndrome (SIDS) when they begin nursing on lipid-rich breast milk. The main aim of the present MS is to investigate the molecular and cellular basis of this disease. To this end, the authors generated both HADHA-deficient and HADHA-mutated hiPSC that were differentiated to cardiomyocytes with a novel, engineered miR maturation cocktail. The exposure to fatty acid (FA) displayed pro-arrhythmic state characterized by alterations in calcium handling, repolarization and beating that were essentially associated with a mis-regulation of cardiolipin homeostasis. This alteration results in change of mitochondrial create structure and function. They concluded that HADHA has a key role in FA oxidation and as an acyl-transferase in cardiolipin remodeling for cardiac homeostasis.

My analysis of this work is:

1. Important biomedical problem in children. The exact cause of SIDS remains unknown. SIDS was the third leading cause of death in children less than one year old in the United States in 2011.

2. Optimal state of the art. The authors presented an updated, deep and critical revision of the state of the art.

3. High conceptual novel hypothesis. A Pubmed search shows that this is the first report describing that alterations in cardiolipin homeostasis has a key role in the pathogenesis of MTFP deficiency.

4. High quality of the experimental design and results. The authors used a broad and important methodologies and approaches to solve the proposed problem. The data are clear and relevant.

5. Conclusions are adequate and based on the results.

6. Important new translational information. This work provides potential relevant information for the future treatment of MTFP-deficient newborns.

The authors should address the following:

A. Major points:

1. Essentially in vitro work. Although the work provides new and interesting new insights. The main question is how these in vitro studies mimics the real clinical situation. Some most key findings should be tested in human samples.

2. Mitochondrial calcium. This ion plays a critical role in mitochondrial energy metabolism due to some key enzyme activities are Ca²⁺-dependent. In my opinion it is important to know if cytosolic Ca²⁺ disturbances described in the MS are also accompanied of alteration in mitochondrial Ca²⁺ and also explained impaired mitochondrial energy metabolism

3. MS presentation is not optimal. The main focus is search and findings are the molecular basis of the disease. Therefore, some secondary and distracting information more oriented to the development of the tools should be moved as Supplementary information (i.e. Fig 2: Cardiomyocyte maturation microRNA screen. Please select the essential information shown in Figure 1 (Generation of HADHA mutant and knockout stem cell derived cardiomyocytes) and Figure 3 (MiMaC accelerates

hiPSC-CM maturation) and fused in only one main figure. Move irrelevant results to Supplementary information.

B. Minor points:

1. The unit of Mr is kDa. Replace K by kDa in all Western blots.
2. The references must uniform to the style of the journal

Reviewer #3 (Remarks to the Author):

In this study, the authors sought to probe the cellular derangements driven by mitochondrial trifunctional protein (TFP) deficiency using iPSC-generated cardiac myocytes (iPSC-CM). In order to unveil the metabolic and functional phenotypic manifestations in a more mature cellular state, a mixture of microRNAs identified by this group previously to be involved in cardiac myocyte maturation was used. In addition, fatty acid loading was used to provoke cellular phenotypic manifestations. Under these latter conditions, there was evidence for altered calcium handling and markers of repolarization derangements in TFP mutant cells. Single cell RNA sequencing analyses revealed distinct subsets in the mutant cells. Lastly, there was evidence for altered mitochondrial structure and function associated with evidence of defective cardiolipin remodeling consistent with previous observations that components of the TFP complex have cardiolipin acyltransferase activity. The new information here includes strategies to promote iPSC-CM maturation and the possible role of cardiolipin disturbances in the cardiac manifestations of mitochondrial TFP deficiency. However, as outlined below, there are concerns regarding some methodological approaches and, more importantly, the evidence supporting a causal link between the observed cardiolipin abnormalities and the phenotype of TFP deficiency (in vivo relevance).

Specific Critique:

1. The assessment of the metabolic maturity of the hiPSC-CM following manipulation with the miRNA cocktail is rather incomplete. Specifically, it is known that Seahorse-based assays to assess

fatty acid oxidation (FAO) lack sensitivity and specificity. It would be important to assess the results of more traditional labeled (^{14}C or ^3H) palmitate or oleate oxidation assays. In addition, it would be important to know whether the expression of genes involved in FAO, an adult metabolic signature, were significantly induced to the level where TFP deficiency would be manifest. These signatures are important given that the goal was to assess a defect in the beta-oxidation pathway.

2. The fatty acid loading experiments aimed at unveiling a cellular phenotype in the mutant cells is rather poorly described. More details should be provided. It would also be important to assess the level of neutral lipid accumulation using triglyceride assays or lipid staining.

3. The defects in mitochondrial structure and cardiac myocyte action potential are of interest but the mechanistic basis is unclear. It seems likely that these effects could be related to general effects of cellular "lipotoxicity". The authors should comment on this and put in the context of the conclusion that much of the phenotype is due to abnormalities in cardiolipin structure.

4. One of the potentially novel findings of this work relates to the role of cardiolipin abnormalities in the TFP-deficient phenotype. In its current form, the evidence provides an association rather than a cause-effect relationship between alterations in cardiolipin and the myriad of abnormalities reported in the mutant cells loaded with fatty acid. It would be important to provide additional data regarding the potential role of altered cardiolipin in the cellular manifestations. Unfortunately, this would likely require in vivo data in mouse models or human tissue with TFP deficiency. At the very least, demonstrating that such cardiolipin abnormalities exist in vivo would be important. In addition, consideration could be given to some type of "rescue" studies in which acyltransferase function is replenished in cells. Short of such results, it is difficult to conclude that the observed abnormality in cardiolipin drives all or any of the derangements in the lipid-loaded cardiac myocytes.

5. The impact of the MiMaC on cardiac myocyte maturation is interesting. By what mechanism do the microRNAs and their targets actually drive this maturation process?

Reviewer #4 (Remarks to the Author):

Critiques:

The authors are not the first one to tackle the mechanism of MTP-deficiency and an MTP-null mouse model has been previously published showing neonatal sudden death and recapitulating the full presentation of the human condition (PubMed: 11390422). However, the novelty is represented by the detailed molecular dissection performed using the engineered MiMaC-treated hPSC-CM models.

The authors stated that they checked for all possible predicted off-target positions. However, CRISPR/Cas9 is known to produce also random off-target modifications. How did the authors deal with potential off-target in non-predicted sites? Or, at least did the authors back-mutate the engineered cell lines using CRISPR to prove it was not the effect of unpredicted off-target variants?

The authors tested hPSC-CM engineered for radical HADHA variants (either KO the first exon or frameshift variants), thus all leading to LOF. What about using pathogenic missense variants such as the common variant HADHA: NM_000182.4:c.1528G>C (p.Glu510Gln)? The p.Glu510Gln variant has been identified in patients presenting with hypoglycemia, cardiomyopathy, muscle hypotonia, and hepatomegaly during the first 2 years of life (PubMed: 9003853). Therefore, this variant would have provided further insights into the mechanisms of cardiac dysfunction.

The authors showed the presence of an alternative and apparently shorter protein product from the HADHAMut clones. Did the authors investigate potential cryptic alternative promoters/initiation codons as the results of the mutation events?

Did the authors study the glycogen metabolism in mature HADHAMut or HADHAKO CM, provided that it represents the secondary fuel system for cardiomyocytes?

Did the authors study what happens if after 6-12 days treatment with FA media, they allow HADHAMut or HADHAKO CM to grow in non-FA culture media again? Does normalization occur? Are the observed changes permanent or all or partially each of the abnormalities normalize?

Minor points:

Please define FPKM at line 192, because it is the first time the acronym appears

MiR-452 OE up-regulation of electrophysiology shows a trend, but does not appear as convincing and compelling as in MiR-208b. Is there any statistical analysis showing significance for the effect of OE of MiR-452 on electrophysiology?

On line 163 the authors refer to six miRNAs, but in Supplemental figure E, they tested five (miR-200a is missing). Why RNA sequencing was not performed on miR-200a? Please reconcile.

What was the molar ratio among miR of MIMAC?

At line 211 the authors specified the use of the MIMAC cocktail (Let7i OE, miR-452 OE, miR-122 KO, and miR-200a KO), but at line 249 they mentioned the chosen six miRs. Please reconcile.

Why measured calcium transients were performed on HADHAMut but not on HADHAKO?

Why was the HADHAKO not assessed for proton leak rescue by SS-31?

Please define ALCAT1

Reviewer #5 (Remarks to the Author):

In this manuscript, Miklas and colleagues present their evaluation of MTP disease mechanism as a result of HADHA mutation. They generate HADHA-deficient hiPSCs and improve their maturation by treating them with a combination of miRNA cocktail. They provide a detailed study on how their miRNA cocktail leads to maturation of hiPSC-cardiomyocytes through upregulation of HOPX. They show that this cocktail improves CM maturation and allows for FA metabolism through FAO as energy source, hence providing the appropriate tools to study how FAO may contribute to the pro-arrhythmia in HADHA-deficient CMs. They then show that FA challenged HADHA mutant CMs showed aberrant calcium handling, delayed repolarization and erratic beating suggesting a pro-arrhythmic state. They found that this pathologic finding was a result of mitochondrial dysfunction due to proton gradient loss and lack of normal cristae of the mitochondria. Very interestingly, they

provide convincing evidence that HADHA knockout leads to a mis-regulation of cardiolipin homeostasis due to the reduction of tetra[18:2] cardiolipin species.

First, the authors should be congratulated for undertaking a tour de force study using meticulous research methods to explore how HADHA is required for FAO and cardiolipin remodeling. The authors use rigorous experimental techniques and well designed studies to provide a complete study. The manuscript is well written (although it may include some data that are not directly related to the final message of the paper). Overall, I believe this paper contributes significantly to our current knowledge in two major ways: i) providing a valuable tool for maturation of PSC-derived cardiomyocytes that can be used in a variety of studies (although this is based on altered expression of miRs and may not be applicable in every system) and ii) providing an in vitro model of human MTP deficient CMs to explore disease mechanism. This is an exciting and near-complete story that may benefit from a few changes.

1. The authors perform 4 different characterization/functional assays to assess the effect of OE or KO of their candidate miRs. They show miR 208-b overexpression increases CM area, miR-452 OE increased the corrected field potential duration, miR-200a KO resulted in an increase in force of contraction and KO of miR-122 resulted in increased O₂ consumption. However, for their bioinformatic analysis of candidate miRs, they chose to perform RNA sequencing on miR-378e OE, -208b OE, -452 OE, -122 KO or -205 KO. The rationale behind this experiment is not entirely clear. One would expect that gene expression analysis would be performed based on the altered expression of miRs that resulted in phenotypic changes (i.e. altered expression of miR 208-b, 200a, 452 and 122). RNAseq on miR 200 OE (which resulted in increased force generation) was not tested here. But interestingly, miR200a OE is included in their final cocktail.
2. Single cell RNA-Sequencing analysis of miR treated CM is interesting and intriguing. The authors have performed diligent analysis of their data to identify clusters according to CM maturity and show that this cluster enriched for MiMac treated CMs. This analysis would have been significantly more convincing if this data was compared to fetal and adult cardiomyocytes (i.e. single cells RNA seq) to illustrate clustering of endogenous adult CMs with MiMac treated hiPSC-derived CMs.
3. It is not clear whether Ca⁺⁺ handling and EP experiments were performed in the presence or absence of MiMAC, since the text and figures (4D-L) don't specify that.
4. Why do authors perform their experiments using both HADHA mut and KO? In certain experimental models, it is not clear why they chose one over the other. Perhaps presenting data on one line may be a more cohesive story.
5. The authors did not observe an accumulation of MLCL in the HADHA Mut CMs (normally seen with TAZ mutations). Based on this observation, they conclude that CL remodeling is the result of first processing by TAZ and then by HADHA. It may be helpful for the authors to include in the discussion the possibility that HADHA remodeling of MLCL to mature cardiolipin may (or may not) be independent of TAZ function.

Minor points:

Figure 1A is not referenced in the paper.

Are the values for CM area in figure 2C significant? From the figure it doesn't look like it. Did the authors compare the values for miR208 overexpression to miR205KO to show significance? It should be compared to control (i.e. empty virus). Additionally, the authors don't show the corrected field potential duration results for KO miRs (they only show overexpression data).

What does EV stand for? I assume this is the control hiPSC line?

Does figure 3J refer to cardiomyocytes isolated from fetal and adult human tissues?

On page 7, line 260, the authors state: "These data indicate that Let7i OE maturation does not govern HOPX cardiac maturation pathways." This observation is solely based on a single OE of let7i that showed no changes in HOPX expression pattern. This is a strong statement and can be toned down, since OE experiments cannot recapitulate the stepwise regulatory mechanisms involving HOPX.

Reviewers' comments

Reviewer #1 (Remarks to the Author):

General concerns

In this study Miklas et al. reported a miRNA cocktail (MiMaC) promoting maturation of human iPSC-derived cardiomyocytes (CM), and defined its target HOPX essential for CM maturation. They generated 3 different hiPSC lines including wild type, HADHAMut, and HADHAKO using CRISPR/Cas9 system, and induced CM differentiation and maturation to explore the mechanism of a human disease sudden infant death syndrome with mitochondrial trifunctional protein deficiency. The authors showed the mutant iPSC-CM has defect in fatty acid beta-oxidation, reduced mitochondrial proton gradient, disrupted cristae structure and defective cardiolipin remodeling. Although several high-throughput screenings were used to determine the microRNA candidates and their targets for CM maturation, similar works have been reported by other groups and thus the novelty of the current study may be a concern. Furthermore, the quality of data presented should be improved, there lacks decisive evidence for concluding HOPX as a key regulator of iPSC-CM maturation, and there is no data to support peptide SS-31 as a treatment for stabilizing cardiolipin.

Specific comments:

1. The maturity of iPSC-CM should be further characterized especially the electrophysiology, given that infant sudden death is typically caused by arrhythmia. At least, whole cell patch clamp for calcium channels should be performed in the HADHA iPSC-CMs.

Response: We thank the reviewer for this question. We have now performed whole cell patch clamp on WT and HADHA Mut CMs that have been treated with glucose and fatty acids. We found there was a significant increase in action potential duration in the HADHA Mut CMs as compared to WT CMs (Average ADP90: WT 541 ms, Mut 1068ms). This elongation period is seen during the plateau phase of the action potential where calcium ions are opposing the change in voltage due to potassium ions. This extent of elongation is indicative of a pathological state and suggests calcium handling as a potential source of this abnormal action potential. Furthermore, the increased tau decay constant, an indication of how rapidly calcium is being sequestered into the sarco/endoplasmic reticulum, is also slower in the HADHA Mut CMs as compared to WT CMs. Since, elongations of action potentials have been shown to result in arrhythmic heart conditions, phase 2 reentry [1], our whole cell patch clamp data suggests the HADHA Mut CMs may be in a pro-arrhythmic state suggesting that indeed, HADHA Mut CMs when challenged with FAs could result in SIDS due to abnormal calcium handling.

We are also excited to note that from our two monolayer paced assays of calcium handling (calcium transients) and electrophysiology (fluovolt) the patch clamp data reinforces our

previous findings at the single cell level. We previously found, using calcium transients, abnormal calcium handling with a longer tau-decay constant in HADHA Mut CMs. This matches well with the longer tau-decay constant found with patch clamp. With fluovolt we found that, examine a monolayer of paced CMs, the HADHA Mut CMs displayed a longer time to full wave duration. This finding is corroborated and shown to be much more dramatic at the single cell level utilizing patch clamp where HADHA Mut CMs had a dramatically longer action potential duration. Utilizing these three orthogonal methods, we have found no abnormalities in the WT CMs treated with glucose and fatty acids and a striking defect in calcium handling resulting in elongated action potentials in HADHA Mut CMs treated with glucose and fatty acids.

The following text has now been added to the manuscript:

Since HADHA Mut CMs cultured in Glc+FA media exhibited abnormal calcium cycling, we assessed whether or not these CMs also exhibited abnormal electrophysiology. Using single cell whole cell patch clamp, we found there was a significant increase in action potential duration in the HADHA Mut CMs as compared to WT CMs (Average ADP90: WT 541 ms, Mut 1068ms) (Figure 4G,H). This elongation period is seen during the plateau phase of the action potential where calcium ions are opposing the change in voltage due to potassium ions. This extent of elongation is indicative of a pathological state and suggests calcium handling as a potential source of this abnormal action potential (Figure 4D-F). Furthermore, the increased tau decay constant is also slower in the HADHA Mut CMs as compared to WT CMs (Figure 4I). There was no difference in resting membrane potential (Supplemental Figure 6C). Since, elongations of action potentials have been shown to result in arrhythmic heart conditions, phase 2 re-entry [1], our whole cell patch clamp data suggests the HADHA Mut CMs may be in a pro-arrhythmic state suggesting that indeed, HADHA Mut CMs when challenged with FAs could result in SIDS due to abnormal calcium handling.

2. In screening candidate miRNAs for CM maturation (which has been reported by several groups), the authors stated that some candidates targeting the same seed region thus choosing miR-378e from the miR-378 family and miR-200a instead of miR-141. Why is that? A random selection? How did the authors determine which one from the microRNA families to be essential for CM maturation?

Response: We found that miRs-378e and -378g were two miRs up-regulated during cardiomyocyte maturation, second to the highest up-regulated miR family of Let7. We also found that miRs-200a and -141 were down-regulated during cardiomyocyte maturation. When we overexpressed miR-378e and -378g in hiPSC-CMs, miR-378e showed a greater OE. MiRs-378e and -378g have the same seed region, except 1 bp. Consequently, these two miRs probably have similar mRNA targets they are repressing.

For miR-141 and -200a, we chose to knockout the miR that was most repressed during cardiomyocyte maturation. miR-200a was more down-regulated in adult human left ventricular tissue than miR-141, consequently, we chose miR-200a to knockout in the microRNA maturation cocktail.

3. The authors stated that different miRNAs regulate different stages of CM maturation. Is it possible that treatment of different miRNAs at different time points may lead to different degrees of maturity? Also, why miR-208b was excluded from the MiMaC, given that miR-208b OE enhances iPSC-CM cell size and twitch force as shown in Figure 2? How about Let-7i?

Response: This is a very interesting question on transient miR expression during post-natal development, and their potential roles. There are certain miRs that are important for directing cardiomyocyte cell fate [2, 3]. These miRs are transiently turned on and off during cardiomyocyte differentiation. Since, some of these miRs are then down-regulated once the cardiomyocyte fate has been established, we argued that these miRs are most likely not necessary for cardiomyocyte maturation. We have found, from our human fetal and adult ventricular miR sequencing data that there is a distinct set of up- and down-regulated miRs during post cardiomyocyte fate commitment maturation. Once a stem cell has differentiated into a cardiomyocyte, it is now a committed somatic cell albeit stuck in an immature state due to *in vitro* culture conditions. It is possible that there are some transient miRs that could be turned on and then off, or vice a versa, during the fetal to adult maturation processes. However, there is no data currently to uncover that as we would need human cardiac tissue samples from different ages of individuals, which is rare to obtain. However, we would postulate that the miR signature found in the adult heart is most likely the dominate miR signature of a mature cardiomyocyte which, when expressed in an immature cardiomyocyte would bring about maturation. Future research is required to identify and analyze potential transient miRs during post-natal maturation.

We excluded miR-208b from MiMaC in order to create a cocktail of miRs with the least number of miRs possible. As the reviewer stated, we showed that miR-208b brought about a significant increase in cell area (Figure 2C) and a modest, but not significant increase, in twitch force (Figure 2E). Let7i, however, was part of the most up-regulated miR family found in the human adult ventricle as compared to the human fetal ventricle [4]. We previously published this finding and the role Let7i had in stem cell derived cardiomyocyte maturation in Kuppusamy *et al.* PNAS 2015 [4]. In this paper, we showed that Let7i was able to bring about an increase in cell area and twitch force. As a result, in order to keep MiMaC with as few redundant miRs as possible, we decided to use Let7i instead of miR-208b. However, we agree with the reviewer, miR-208b is a very interesting miR to continue to study in terms of its role in cardiomyocyte maturation.

4. The authors should carefully define the status of so-called “intermediate maturity of iPSC-CM” including the morphological, electrophysiological and functional changes. Also, how does the treatment of Let-7i and miR-452 OE lead to the intermediate status?

Response: We thank the reviewer for this question. Based on our scRNA-Seq data (Figure 3L-O), we identified cluster 1 as an intermediate state of cardiomyocyte maturation. Cluster 1 was enriched for hiPSC-CMs overexpressing miR-452 and -Let7i. Examining the differentially expressed genes found in cluster 1, we found GO terms for metabolism and mitochondria for the up-regulated genes and GO terms for cardiac muscle and structure for the down-regulated genes (Rebuttal Figure 1A and Rebuttal Tables 1-3). To gain further insight into the physiological relevance of these gene changes, we took the significantly changed genes in these GO categories

and asked if there was any overlap in these same gene expression changes between the human fetal and adult ventricular cardiomyocyte state. The overlap in genes changing in the same direction were: 14 genes heart contraction, 5 ion channel and 21 for OXPHOS and ATP (Rebuttal Tables 4-6). These data suggest that the intermediate state has aspects of human cardiomyocyte maturation, yet, is not as mature as the MiMaC treated CMs. This reinforces the need for multiple miR perturbations to be performed on hPSC-CMs in order to bring about a more robust maturation process.

As the reviewer states, we found that the intermediate cell type was enriched for CMs treated with Let-7i and miR-452 OE. The treatment of hiPSC-CMs with let-7i and miR-452 led to a significant change in 36 let-7i targets and 23 miR-452 targets (Rebuttal Tables 7 and 8). Similar to what we did previously, we took those targets and overlapped them with the genes changing between the human fetal and adult ventricular cardiomyocyte state. We found 11 Let7i and 4 miR-452 targets that were repressed in our hiPSC-CMs and that are down-regulated during the fetal to adult CM transition (Rebuttal Tables 9 and 10). These data suggest that the treatment of hiPSC-CMs with Let-7i and miR-452 OE does lead to predicted target gene repression and that these targets do have physiological significance as they are also repressed in the human adult heart. However, as previously stated, these two miRs alone are not sufficient to bring about robust cardiomyocyte maturation but, when combined with miR-200a and -122 KO (MiMaC) we see a more robust maturation response.

When characterizing the intermediate state within our WT and HADHA mutant scRNA-Seq data, we took this opportunity to ask, in an unbiased way, whether or not an intermediate state was present in the progression of healthy to diseased CM. Our first goal was to determine if we could use scRNA-Seq algorithms to identify a sensical kinetics plot of disease progression. Indeed, using CellTrails, we were able to do so.

The following text has been added to the manuscript:

Since we had identified two different disease states, replication/endoreplicating or not HADHA Mut CMs, along with our healthy WT CMs and a shared intermediate state, we wondered if we could find a trajectory upon which a healthy CM follows to a disease state in an unbiased manner. To do this, we used the CellTrails[5] method as it allows for branching. Based on CellTrails clustering, we found six different states with a clear distinction between mutant and WT CMs (Supplemental Figure 8A). Clusters 1 and 4 were identified as WT cells, while clusters 2, 3, 5 and 6 were identified as HADHA mutant CMs. When examining the kinetics of cell progression with a start point of mature WT CMs, the path starts at state 1 (S1) and then follows down a path through S4, S3 and S5 until the end where it splits into two final paths, S6 and S2, both of which are HADHA Mut CM enriched clusters (Figure 5K,L). What was exciting to see was that we were able to model a path from matured WT CMs down to the pathological mutant CMs and, there were three in between states, S4, S3 and S5. We next wanted to identify whether or not our intermediate state was found in these states.

Utilizing two hallmark maturation genes, one that goes up with ventricular maturation (MYH7) and one that goes down with ventricular maturation (MYL7) we examined the maturity of the three in between states (Supplemental Figure 8B). S4, enriched for WT CMs, had low

expression of the maturation gene MYH7 and a high expression of the immature gene MYL7. Moving down the kinetic pathway we found S3, enriched for HADHA Mut CMs, also had low MYH7 and high MYL7. Finally S5, also enriched for HADHA Mut CMs, had a scattered expression of both genes suggesting further progression to a pathological state. These data suggest that S4 and S3 are less mature CMs, representing the previously identified intermediate CM while S5 is on its way to a disease state.

Finally, our kinetics model identified two branches as end states of the pathological HADHA Mut CMs, S6 and S2 (Supplemental Figure 8C). What was exciting about this finding was that these two branches identified the two different HADHA Mut states from our scRNA-Seq analysis, non-replicating (S2) and replicating/endocycling (S6) HADHA Mut CMs. These kinetics data, in an unbiased manner, identified WT CMs as one point to start this pathway for a CM to follow. The pathway then led through an intermediate CM state that had not been able to mature to a matured WT CM. Finally, the HADHA Mut CMs progress further down the pathway to end in two separate states, the pathological non-replicating and replicating/endocycling HADHA Mut CMs.

5. The authors showed changes of mitochondrial structure and function in HADHA mutant cells. Did the authors check mitochondrial fusion, fission and mitophagy as well?

Response: Thank you for the question. We have now taken an unbiased approach and examined genes associated with mitochondrial fusion, fission and mitophagy in our scRNA-Seq data set. We found two genes (BAX and MFN2) that were highly expressed in cluster three, MiMaC Cluster, but downregulated in cells defective for HADHA, cluster 0.

The following text has been added to the manuscript:

Finally, we found two genes (BAX and MFN2) that were highly expressed in cluster three, MiMaC Cluster, but downregulated in cells defective for HADHA, cluster 0 (Supplemental Figure 7J). These findings support a recent study showing that HADHA mutants have defects in mitochondrial fission and fusion machinery, specifically, they also found the gene MFN2 to be mis-regulated leading to punctate malfunctioning mitochondria [6].

6. Quality of data presentation should be improved. For example, the labels in some figures are either not shown up or appear blurry such as in Fig 1B, 1C, Fig 2a, Fig 3H, 3N, Fig 4B & Fig 5A.

Response: Thank you for bringing this to our attention. We have now provided higher quality images.

7. There lacks conclusive evidence to demonstrate HOPX as a novel regulator for iPSC-CM maturation.

Response: Thank you for inquiring about HOPX. In our manuscript, we concluded the HOPX section by stating HOPX and its role in cardiomyocyte maturation had been further explored in Friedman *et al.* [7]. This paper, of which we provided the HOPX OE functional data, utilized scRNA-Seq to examine cardiomyocyte differentiation and some aspects of HOPX in cardiomyocyte maturation [7]. We have now analyzed further the role HOPX has in repressing serum response factor genes and how this repression may lead to a reduction of cell cycle which, facilitates the cell to move into a hypertrophic growth phase.

The following text has been added to the manuscript:

HOPX is a novel regulator of CM maturation

To better understand the molecular mechanisms that are critical for cardiac maturation, the overlapping predicted targets of the screened six miRs were determined. We had previously studied the predicted targets of Let7, insulin receptor-pathway and PRC2 function, and their role in cardiomyocyte maturation[4], however, we had not studied the targets of the down-regulated miRs. We found that all four down-regulated miRs during maturation had five common predicted targets. One of these predicted targets, HOPX (Figure 3H), is important for cardiomyoblast specification[8], yet an understanding of how this transcriptional regulator regulates human cardiomyocyte maturation is still not well understood. HOPX expression was up-regulated *in vitro* (Figure 3I), *in vivo* (Figure 3J) and in MiMaC treated hiPSC-CMs (Figure 3K). To analyze how our MiMaC miRs might individually regulate HOPX expression during maturation, we analyzed HOPX levels in miR-122 KO and Let7i OE hiPSC-CMs. We found HOPX was up-regulated 6.8 fold in D30 miR-122 KO hiPSC-CMs while Let7i OE matured hiPSC-CMs had no effect on HOPX expression (Supplemental Figure 3E). These data suggest that Let7i OE maturation does not govern HOPX cardiac maturation pathways. This highlights the necessity of combining multiple miRs together to generate a robust maturation effect in hPSC-CMs and that HOPX seems to be a strong candidate for post-committed cardiomyocyte maturation.

We previously assessed the role of HOPX OE in cardiomyocyte maturation and found that HOPX OE led to an increase in CM size[7]. To do this, we generated an iPSC line that conditionally overexpressed HOPX fused with a nuclear localization signal (NLS) and enhanced green fluorescent protein (eGFP) termed HOPX OE. The overexpression of HOPX was controlled by a tetracycline promoter. A corresponding control iPSC line of NLS-eGFP under the control of a tetracycline promoter was also generated termed negative control (NC)[7]. To better understand the transcriptional changes HOPX OE had on CMs, we had previously performed RNA-Sequencing on CMs that had 2 weeks of HOPX OE or just eGFP OE (NC) and determined the differentially expressed genes between these two groups. GO term analysis showed that many of the up-regulated processes were related to cell growth and maturation while many of the down-regulated processes were associated with cell cycle[7]. However, it was still unclear how these genes were related to one another and why HOPX OE led to cell cycle repression.

Using STRING analysis, we showed that the differentially expressed genes associated with cell division in the HOPX OE group generated a highly-interconnected network with key cell cycle genes highly down-regulated (Supplemental Figure 4A) which, also re-capitulated the

cell cycle repression finding we found with MiMaC treated hiPSC-CMs. We then generated four clusters using Kmeans clustering: regulation of mitotic cell cycle, cell division, inhibition of cilia and ubiquitin protein. Representative cell cycle genes, BUB1, MKI67 and CENPE, were down-regulated while the inhibitor of many G1 cyclin/cdk complexes, CDKN1C, was significantly up-regulated in the HOPX OE condition (Supplemental Figure 4B). These data suggest that HOPX OE is a driver of exit from cell cycle and cardiomyocyte hypertrophy.

HOPX regulates cell cycle via SRF genes

HOPX is a homeodomain protein that does not bind DNA but rather is recruited to locations in the genome by serum response factor (SRF). HOPX in turn recruits histone deacetylase (HDAC) and removes acetylation marks resulting in the silencing of genes (Supplemental Figure 4C). HOPX OE led to a significant down-regulation of 294 SRF targets (hypergeometric test p-value is 1.31×10^{-5}) while the group of SRF targets that were up-regulated was not significant (hypergeometric test p-value is 0.99) (Supplemental Figure 4D). Clearly, HOPX OE had successfully led to the repression of many SRF target genes.

We validated using qPCR a known SRF target gene that should be repressed during cardiomyocyte maturation, natriuretic peptide precursor A (NPPA). After two weeks of HOPX OE, NPPA was significantly repressed, while cardiac troponin C, a non-SRF cardiac gene was unaffected by HOPX OE. The ventricular isoform of myosin light chain, MYL2, which is up-regulated as ventricular cardiomyocytes mature, increased 1.87-fold in expression after two weeks of HOPX OE (Supplemental Figure 4E).

To better understand the role HOPX may have in repressing SRF target genes during normal cardiomyocyte maturation we determined the SRF target genes in common between HOPX OE vs. NC hiPSC-CMs and the human adult vs. fetal myocardium (ventricular myocardium) transitions. There were 294 SRF targets lower in HOPX OE vs. NC hiPSC-CMs and 564 SRF targets lower in adult vs. fetal myocardium. 76 SRF targets were common between the two groups and formed a significant group of genes (hypergeometric test p-value is 5.44×10^{-24}) (Supplemental Figure 4F). Assessing these 76 genes via GO term analysis showed a strong association with repression of cell cycle (Supplemental Figure 4G). Using STRING analysis, we determined the network of connected genes out of the 76 genes in common and ran Kmeans clustering to generate 4 clusters (Supplemental Figure 4H). We found the majority of the genes fell into two cell cycle clusters (mitotic cell cycle and cell cycle), the third cluster was associated with DNA repair and the fourth cluster associated with muscle development. Finally, we wanted to determine the network of SRF regulated cell cycle genes that was in common between the HOPX OE line and adult cardiomyocytes (Supplemental Figure 4I). This network identified, at least in part, the way in which HOPX directly regulates SRF cell cycle genes to mature a cardiomyocyte through its exit from cell cycle. Using MCL clustering, one cluster was found showing that all genes were associated with cell cycle with 7 of the 10 genes associated with the spindle machinery. These data indicate that, at least partly MiMaC acts through HOPX, repressing SRF cell cycle targets.

8. The authors should provide data to conclude that HADHA is required for fatty acid beta-oxidation in their iPSC-CMs. The therapeutic effect of SS-31 should be tested as well.

Response: We thank the reviewer for this chance to clarify our data on how fatty acid oxidation is disrupted in the HADHA mutant and knockout cells. In Figure 4C, we showed that HADHA Mut and KO CMs were unable to process the long-chain fatty acid palmitate. In this assay, we measured background oxidation rates using the Seahorse platform. During the assay, we injected bovine serum albumin (BSA) conjugated to palmitate. During these injections, the only substrate present that could lead to an increase in oxygen consumption rate was the long-chain fatty acid palmitate. Figure 4C showed that the matured WT CMs were able to oxidize palmitate while, the matured HADHA Mut and KO CMs were not capable of oxidizing palmitate.

We have now also added new mass spec analysis showing that HADHA defective cells cannot perform FAO in mitochondria (Figure 7B-G). Short explanation follows: A mouse model of an HADHA KO mouse was studied by Ibdah *et al.* and found that newborn HADHA KO mice died of sudden death 6-36 hours after birth [9]. In this paper, they examined the serum of the HADHA KO mice in the 6-36 hours window before the pups died. They showed that there was an increase in long chain serum fatty acids, long chain serum acylcarnitines and long chain serum hydroxylated acylcarnitines. The KO of HADHA led to the specific disruption of long-chain fatty oxidation, as medium chain fatty oxidation utilizes different enzymes, and results in specific accumulation of long chain fatty acids and their intermediates, including the unique accumulation of hydroxylated long chain fatty acids seen in MTP deficient patients. We showed in Figure 7B the accumulation of long-chain acyl carnitines and we showed in Figures 7C-E the accumulation of fatty acid intermediates in HADHA KO CMs generated right after the first step of long-chain fatty acid oxidation that would accumulate if no HADHA was present. These were the same findings from the mouse model in Ibdah *et al.* We have now also added Figure 7G and Supplemental Figure 10G which show the specific accumulation of hydroxylated long-chain fatty acids, a hallmark of the clinical manifestation of MTP deficiency [9, 10]. Multiple hydroxylated long-chain fatty acid species have an over 400-fold accumulation as compared to the WT CMs. These data have been added to Table 1 in the manuscript. Finally, we have also now examined the total abundance of triglycerides in the HADHA Mut and KO CMs and found those to be significantly higher than those in the WT CMs (Figure 7F). We believe these data combined strongly indicate there is a halt in specifically, long-chain fatty acid oxidation.

We have now quantified mitotracker staining in WT and HADHA Mut 12D FA and SS-31 treated CMs. We found that there is no significant difference between the WT and HADHA Mut SS-31 treated CMs (Rebuttal Figure 2A). As a result, we believe the main therapeutic effect of SS-31 is to help maintain mitochondrial proton gradient as shown by less proton leak and more polarized mitochondria via mitotracker staining. We postulate that SS-31 is interacting with cardiolipin due to previous publications that have identified SS-31 as interacting with cardiolipin [11, 12].

Reviewer #2 (Remarks to the Author):

NCOMMS-18-29739-T analysis

Mitochondria trifunctional protein (MTFP) is a heterooctamer; two genes encode the alpha and beta subunits of MTFP, respectively, hydratase subunit A (HADHA) and hydratase subunit B (HADHB). MTFP deficiency results of impaired fatty acid oxidation (FAO) due to mutations in hydroxyacyl-CoA dehydrogenase/3-ketoacyl-CoA thiolase/enoyl-CoA hydratase subunit A (HADHA/LCHAD) or subunit B (HADHB). MTFP-deficient newborns developed arrhythmias and sudden infant death syndrome (SIDS) when they begin nursing on lipid-rich breast milk. The main aim of the present MS is to investigate the molecular and cellular basis of this disease. To this end, the authors generated both HADHA-deficient and HADHA-mutated hiPSC that were differentiated to cardiomyocytes with a novel, engineered miR maturation cocktail. The exposure to fatty acid (FA) displayed pro-arrhythmic state characterized by alterations in calcium handling, repolarization and beating that were essentially associated with a mis-regulation of cardiolipin homeostasis. This alteration results in change of mitochondrial create structure and function. They concluded that HADHA has a key role in FA oxidation and as an acyl-transferase in cardiolipin remodeling for cardiac homeostasis.

My analysis of this work is:

1. Important biomedical problem in children. The exact cause of SIDS remains unknown. SIDS was the third leading cause of death in children less than one year old in the United States in 2011.
2. Optimal state of the art. The authors presented an updated, deep and critical revision of the state of the art.
3. High conceptual novel hypothesis. A Pubmed search shows that this is the first report describing that alterations in cardiolipin homeostasis has a key role in the pathogenesis of MTFP deficiency.
4. High quality of the experimental design and results. The authors used a broad and important methodologies and approaches to solve the proposed problem. The data are clear and relevant.
5. Conclusions are adequate and based on the results.
6. Important new translational information. This work provides potential relevant information for the future treatment of MTFP-deficient newborns.

The authors should address the following:

A. Major points:

1. Essentially in vitro work. Although the work provides new and interesting new insights. The

main question is how these *in vitro* studies mimics the real clinical situation. Some most key findings should be tested in human samples.

Response: We thank the reviewer for this great question on how our *in vitro* platform mimics the human clinical situation. We now have iPSC lines with the founder mutation in *HADHA* c.1528G>C from a collaborator in Finland, Prof. Wartiovaara-Suomalainen. We differentiated these iPSCs to cardiomyocytes and characterized their phenotype when matured and cultured in the presence of long-chain fatty acids. We are very excited to share that the *HADHA* c.1528G>C CMs pheno-copy our CRISPR generated lines. We now show that *HADHA* c.1528G>C CMs cultured in glucose have well-formed myofibrils and sarcomeres as seen by phalloidin and α -actinin staining (Figure 6K). Furthermore, when *HADHA* c.1528G>C CMs are cultured in glucose with fatty acids for 12-days, we see the same sarcomere dissolution and loss of mitochondrial potential gradient, as shown via mitotracker, as we previously had shown with our *HADHA* Mut, but not control CM line (Figure 6K,L).

Finally, we performed targeted GC-MS to examine cardiolipin species in the fatty acid treated *HADHA* c.1528G>C CMs. We previously found that *HADHA* KO CMs were unable to remodel cardiolipin to generate a more mature cardiolipin profile. The *HADHA* c.1528G>C CMs displayed a similar inability to remodel cardiolipin, specifically they had low abundance of [18:1][18:2][18:2][20:2] and [18:2]₃[20:2] were not remodeled while [18:1]₃[18:2] was only partially remodeled (Figure 7I). These data reinforce the cardiolipin abnormalities found in our CRISPR generated *HADHA* KO CMs correspond well with a human patient derived *HADHA* c.1528G>C founder mutation CM.

These data, for the first time, show a human patient background with the founder mutation, c.1528G>C, in cardiomyocytes in a cell culture model. As a brief reminder, culture of adult cardiomyocytes *ex vivo* (from mouse) is not a trivial task. On top of this, since the heart is a non-regenerative organ, clinical biopsies of patient cardiomyocytes is very rare and certainly not enough cells would be obtained to perform cell culture assays. Our data with the patient *HADHA* c.1528G>C CMs recapitulates our *HADHA* Mut and KO CM results and reinforces our findings, with another independent line, that the disease pathology we have observed is due to mutations in *HADHA*.

Human clinical samples of cardiomyocytes from MTP deficiency patients have not previously been studied and would be extremely difficult to obtain and be in such small numbers as to not be able to perform experiments with these cells. However, previous groups have studied fibroblasts from these patients along with lipids found in these patients' serum. Previously, as an alternative, a mouse model of an *HADHA* KO mouse was studied by Ibdah *et al.* which found newborn *HADHA* KO mice died of sudden death 6-36 hours after birth [9]. In this paper, they examined the serum of the *HADHA* KO mice in the 6-36 hours window before the pups died. They showed that there was an increase in long-chain serum fatty acids, long-chain serum acylcarnitines and long-chain serum hydroxylated acylcarnitines. The KO of *HADHA* led to the specific disruption of long-chain fatty oxidation, as medium chain fatty oxidation utilizes different enzymes, and results in specific accumulation of long chain fatty acids and their intermediates, including the unique accumulation of hydroxylated long chain fatty acids seen in MTP deficient patients.

Previously, we showed in the manuscript there are a number of markers, fatty acid species, that have abnormal abundances in cases of fatty acid oxidation. We showed the accumulation of long-chain acyl carnitines (Figure 7B) and we showed the accumulation of fatty acid intermediates that would be expected to accumulate right after the first step of long-chain fatty acid oxidation (Figures 7C-E). These results are the same as what has been previously shown *in vivo* from the HADHA mouse model in Ibdah *et al.*

As the reviewer points out, a connection to the human clinical setting would strengthen our *in vitro* platform. One of the hallmarks for testing whether a patient has MTP deficiency is to examine if there are increased levels of hydroxylated long-chain fatty acids in blood serum [13, 14]. In the latest clinical papers looking at MTP deficiency and LCHAD children, they only are able to examine physical features of these children (birth weight, height), perform sequencing, obtain blood samples to look at fatty acid levels [15]. Typically, fibroblasts from patient skin biopsies have been cultured and been shown to secrete increased levels of hydroxylated fatty acids secreted [10]. The reason for what seems like a superficial examination of these patients is due to severity of asking for a heart biopsy from a patient, whether they are healthy or sick. In general, heart biopsies are only performed on aborted fetuses and human adults that are about to undergo a heart transplant (unhealthy heart tissue) or are unfortunately comatose from a car accident or some other trauma and when they are taken off life support, if they consent to donate their organs, you can get heart samples. These heart samples are typically in the form of RNA you can buy from a company or tissue sections. People do not typically preserve cardiac tissue in a proper way or immediately extract from these biopsies for GC-MS and LC-MS to perform metabolomics and lipidomics.

However, we are excited to say that we have now found the same extreme increase in the abundance of hydroxylated long-chain fatty acids in our HADHA KO CMs. We now show in Figure 7G the significant increase in the sum of all hydroxylated long-chain fatty acids in the HADHA KO CMs as compared to the WT CMs. Furthermore, we show in Table 1, the fold increase of all the identified hydroxylated FA species. These data match well with the *in vivo* mouse data from Ibdah *et al.* and we have also identified new hydroxylated species that change which have not been previously identified [9].

Finally, one of the key findings we present is the abnormal cardiolipin remodeling that occurs due to mutations in HADHA. We presented in the manuscript a decrease in tetra[18:2]-CL and an increase in CL intermediates as a result of the inability to remodel nascent cardiolipin to the mature tetra[18:2]-CL (Figure 7H-J). Due to the sudden death of HADHA KO mice and the immature profile of cardiolipin in a newborn neonate, these *in vivo* experiments have not been performed. However, in the heterozygous HADHA mouse heart, which presents with a much milder phenotype, due to still having one wild type copy of HADHA, Mejia *et al.* showed that there is a reduction in the critical tetra[18:2]-CL [16]. These data show that a mutation in HADHA leads to cardiolipin defects in an *in vivo* context. These data reinforce our *in vitro* model and we have now added this discussion to the paper. We hope that these new findings and comparisons to both *in vivo* mouse model, the current available human clinical data and our HADHA c.1528G>C founder mutation CM data demonstrate the robustness and fidelity of our *in vitro* model of human cardiac MTP deficiency.

2. Mitochondrial calcium. This ion plays a critical role in mitochondrial energy metabolism due to some key enzyme activities are Ca^{2+} -dependent. In my opinion it is important to know if cytosolic Ca^{2+} disturbances described in the MS are also accompanied of alteration in mitochondrial Ca^{2+} and also explained impaired mitochondrial energy metabolism.

Response: We thank the reviewer for this excellent question. We have now used a fluorescent calcium indicator that is sequestered in the mitochondria. Using this dye, we were able to examine the relative fluorescence intensity in the mitochondria of WT and HADHA Mut CMs to assess disturbances in calcium concentration within the mitochondria. We found that there was a significant decrease in co-localization and intensity of the calcium indicator dye in the HADHA Mut CMs treated for 12-days of glucose and fatty acid medium as compared to the WT CMs.

The following text has been added to the manuscript:

Since the mitochondrial membrane potential was disrupted and we had previously seen abnormal calcium handling (Figure 4D-F), we wanted to determine the calcium levels in the mitochondria to see if this was perturbed as it could be potentially contributing to the mitochondrial potential gradient disruption. Using a mitochondrial calcium sequestering dye, Rhod2, we were able to examine the relative fluorescence intensity in the mitochondria of WT and HADHA Mut CMs to assess disturbances in calcium concentration within the mitochondria. We found that there was a significant decrease in co-localization and intensity of the calcium indicator dye in the HADHA Mut CMs treated for 12-days of glucose and fatty acid medium as compared to the WT CMs (Figure 6C and Supplemental Figure 9D).

3. MS presentation is not optimal. The main focus is search and findings are the molecular basis of the disease. Therefore, some secondary and distracting information more oriented to the development of the tools should be moved as Supplementary information (i.e. Fig 2: Cardiomyocyte maturation microRNA screen. Please select the essential information shown in Figure 1 (Generation of HADHA mutant and knockout stem cell derived cardiomyocytes) and Figure 3 (MiMaC accelerates hiPSC-CM maturation) and fused in only one main figure. Move irrelevant results to Supplementary information.

Response: We thank the reviewer for their comment. We believe we are in a bit of a “catch 22” with this comment. On the one hand, we agree with the reviewer, the main focus of the paper is the disease model. It would be more streamlined to push all the tool building to the supplement. However, the disease modeling was possible due to the maturation strategies developed and employed in this paper. These strategies offer a novel cocktail of easily transducible miRs to mature hPSC-CMs to a point where they can now utilize long-chain fatty acids which results in an easy means to study lipid diseases *in vitro*. As a result of the importance of these tools, we thought it would be important to show how they were developed. The “catch 22” comes from the fact that we have actually streamlined the paper already, putting mechanistic action of HOPX data into a separate manuscript [7], however, some of the reviewers wanted to see that data more clearly in this manuscript (this question is answered in reviewer 1’s section). They would like even more data for the justification of the tools used in this paper. So, we are stuck in a “catch 22” where some people would like to see less tools but when we remove some data other people

then want to see more about the tools. We hope the reviewer can appreciate the situation we are in and that the figure sets we have presented are as streamlined as we believe they should be.

B. Minor points:

1. The unit of Mr is kDa. Replace K by kDa in all Western blots.
This has now been corrected.

2. The references must uniform to the style of the journal
This has now been corrected.

Reviewer #3 (Remarks to the Author):

In this study, the authors sought to probe the cellular derangements driven by mitochondrial trifunctional protein (TFP) deficiency using iPSC-generated cardiac myocytes (iPSC-CM). In order to unveil the metabolic and functional phenotypic manifestations in a more mature cellular state, a mixture of microRNAs identified by this group previously to be involved in cardiac myocyte maturation was used. In addition, fatty acid loading was used to provoke cellular phenotypic manifestations. Under these latter conditions, there was evidence for altered calcium handling and markers of repolarization derangements in TFP mutant cells. Single cell RNA sequencing analyses revealed distinct subsets in the mutant cells. Lastly, there was evidence for altered mitochondrial structure and function associated with evidence of defective cardiolipin remodeling consistent with previous observations that components of the TFP complex have cardiolipin acyltransferase activity. The new information here includes strategies to promote iPSC-CM maturation and the possible role of cardiolipin disturbances in the cardiac manifestations of mitochondrial TFP deficiency. However, as outlined below, there are concerns regarding some methodological approaches and, more importantly, the evidence supporting a causal link between the observed cardiolipin abnormalities and the phenotype of TFP deficiency (in vivo relevance).

Specific Critique:

1. The assessment of the metabolic maturity of the hiPSC-CM following manipulation with the miRNA cocktail is rather incomplete. Specifically, it is known that Seahorse-based assays to assess fatty acid oxidation (FAO) lack sensitivity and specificity. It would be important to assess the results of more traditional labeled (14C or 3H) palmitate or oleate oxidation assays. In addition, it would be important to know whether the expression of genes involved in FAO, an adult metabolic signature, were significantly induced to the level where TFP deficiency would be manifest. These signatures are important given that the goal was to assess a defect in the beta-oxidation pathway.

Response: We thank the reviewer for their comment. Please refer to Reviewer #1 question #8 for our response to how the cells are utilizing fatty acids.

We have now assessed our scRNA-Seq data for significantly changed fatty acid oxidation genes. We found two significantly changed genes, ALDH3A2 and EC11, that were up-regulated in the matured MiMaC treated CMs and these data have now been added to the manuscript (Supplemental Figure 5E). These data suggest that the microRNA maturation cocktail does up-regulate the machinery used to process fatty acids.

2. The fatty acid loading experiments aimed at unveiling a cellular phenotype in the mutant cells is rather poorly described. More details should be provided. It would also be important to assess the level of neutral lipid accumulation using triglyceride assays or lipid staining.

Response: We have now provided these further details in the methods section.

The following text has been added to the manuscript:

Glucose and Fatty Acid Media: The base media which we are calling Glucose Media, is RPMI supplemented with B27 with insulin. The fatty acid media is the glucose media with oleic acid conjugated to BSA (Sigma O3008): 12.7 μ g/mL, linoleic acid conjugated to BSA (Sigma L9530): 7.05 μ g/mL, sodium palmitate (Sigma P9767) conjugated to BSA (Sigma A8806): 52.5 μ M and L-carnitine: 125 μ M. Fatty acid (FA) experiments used this B27 + insulin + the three FAs (oleic acid, linoleic acid and palmitic acid), in RPMI media. This media was changed every other day during the 6-days or 12-days of treatment.

We have now examined the total abundance of triglycerides in the HADHA Mut and KO CMs and found those to be significantly more abundant than those in the WT CMs (Figure 7F). We thank the reviewer for helping us better describe the lipid profile in our HADHA Mutant and KO CMs.

3. The defects in mitochondrial structure and cardiac myocyte action potential are of interest but the mechanistic basis is unclear. It seems likely that these effects could be related to general effects of cellular “lipotoxicity”. The authors should comment on this and put in the context of the conclusion that much of the phenotype is due to abnormalities in cardiolipin structure.

Response: We recognize the point the reviewer has raised, lipotoxicity has been a major concern in the field when culturing cardiomyocytes *in vitro*. However, we along with others, have now shown that this problem can be avoided, at least for human stem cell derived cardiomyocytes [17-19] by utilizing a carefully chosen and prepared FA cocktail for these analyses. See below for more details:

In the text we have stated, “Finally, we assessed the addition of fatty acids with MiMaC to increase cardiomyocyte maturation. Three long chain fatty acids, palmitate, linoleic and oleic acid were added to the basal cardiac media used. We found the MiMaC + FA cells were enriched in cluster 2. While some studies have shown lipotoxicity with particular FAs, the analysis of our carefully optimized FA-treatment procedure showed no increase in transcripts indicative of apoptosis, indicating minimal lipotoxicity in this assay (Supplemental Table S5) [20]. These data indicate that MiMaC was essential to bring about a robust transcriptional maturation of our hiPSC-CMs and that it was necessary to incorporate all four microRNAs together to bring about this robust maturation response.”

We also stated, “While challenging CMs with FAs can lead to lipotoxicity, we have carefully developed a concentration and combination of three fatty acids that do not result in lipotoxicity (Figure 3L and Supplemental Figure 6A) [21, 22]. Moreover, other groups in the field of hPSC-CM maturation have also found that carefully chosen and fully conjugated FAs stimulate aspects of CM maturation [17].”

The following text has been added to the manuscript:

It was also important to rule out lipotoxicity as a potentiator of the pro-arrhythmic events found in the HADHA Mut CMs. We were able to rule this out as we saw no up-regulation of cell death

pathways in the WT CMs treated with FAs and no abnormalities in calcium handling or electrophysiology in the WT CMs treated with FAs.

4. One of the potentially novel findings of this work relates to the role of cardiolipin abnormalities in the TFP-deficient phenotype. In its current form, the evidence provides an association rather than a cause-effect relationship between alterations in cardiolipin and the myriad of abnormalities reported in the mutant cells loaded with fatty acid. It would be important to provide additional data regarding the potential role of altered cardiolipin in the cellular manifestations. Unfortunately, this would likely require in vivo data in mouse models or human tissue with TFP deficiency. At the very least, demonstrating that such cardiolipin abnormalities exist in vivo would be important. In addition, consideration could be given to some type of “rescue” studies in which acyltransferase function is replenished in cells. Short of such results, it is difficult to conclude that the observed abnormality in cardiolipin drives all or any of the derangements in the lipid-loaded cardiac myocytes.

Response: We thank the reviewer for this question. For the response, please refer to Reviewer #2 question #1.

5. The impact of the MiMaC on cardiac myocyte maturation is interesting. By what mechanism do the microRNAs and their targets actually drive this maturation process?

Response: We thank the reviewer for this question. Please see reviewer #1 question #7 for our answer.

Reviewer #4 (Remarks to the Author):

Critiques:

The authors are not the first one to tackle the mechanism of MTP-deficiency and an MTP-null mouse model has been previously published showing neonatal sudden death and recapitulating the full presentation of the human condition (PubMed: 11390422). However, the novelty is represented by the detailed molecular dissection performed using the engineered MiMaC-treated hPSC-CM models.

The authors stated that they checked for all possible predicted off-target positions. However, CRISPR/Cas9 is known to produce also random off-target modifications. How did the authors deal with potential off-targets in non-predicted sites? Or, at least did the authors back-mutate the engineered cell lines using CRISPR to prove it was not the effect of unpredicted off-target variants?

Response: We thank the reviewer for this question. We have now obtained a new human induced pluripotent stem cell line from a patient with the founder mutation found in mitochondria protein deficiency *HADHA* c.1528G>C. We then differentiated this hiPSC line into cardiomyocytes and assessed the sarcomere structure, mitochondrial potential gradient and cardiolipin profile. We found that this third, independent line, with a different background from our CRISPR modified lines and only a single point mutation in the gene *HADHA*, recapitulates the disease phenotype found in our *HADHA* Mut and KO CMs (Figure 6J-L and Figure 7I). Consequently, since three independent lines showed the same disease phenotype, we do not believe there was any off-targets that would have generated our observed findings but rather our findings are *HADHA* dependent. Further details on these new findings can be found in Reviewer #2 question #1.

The authors tested hPSC-CM engineered for radical *HADHA* variants (either KO the first exon or frameshift variants), thus all leading to LOF. What about using pathogenic missense variants such as the common variant *HADHA*: NM_000182.4:c.1528G>C (p.Glu510Gln)? The p.Glu510Gln variant has been identified in patients presenting with hypoglycemia, cardiomyopathy, muscle hypotonia, and hepatomegaly during the first 2 years of life (PubMed: 9003853). Therefore, this variant would have provided further insights into the mechanisms of cardiac dysfunction.

Response: We thank the reviewer for this excellent question. We have now obtained a human iPSC line from a patient with the founder mutation in *HADHA* c.1528G>C from a collaborator in Finland. This patient line recapitulated our disease phenotype findings we previously observed in the *HADHA* Mut and KO CMs we generated via CRISPR. Please refer to Reviewer #2 question #1 for further details on these new findings.

The authors showed the presence of an alternative and apparently shorter protein product from the *HADHA*mut clones. Did the authors investigate potential cryptic alternative promoters/initiation codons as the results of the mutation events?

Response: We thank the reviewer for this question. We have now looked into where enhancers may be located in the HADHA gene on the genome. We used H3K27 acetylation marks as a mark of where enhancers may be located. We only found H3K27ac marks around the transcription start site of HADHA and no peaks within gene body (Rebuttal Figure 3A). In contrast, Pyruvate Kinase M (PKM), a gene known to have multiple transcript isoforms, has multiple H3K27ac peaks within the gene body (Rebuttal Figure 3B).

Did the authors studied the glycogen metabolism in mature HADHAMut or HADHAKO CM, provided that it represents the secondary fuel system for cardiomyocytes?

Response: We thank the reviewer for their question. As glycogen can be converted into various forms of glucose, we performed Seahorse analysis to determine the glycolytic capacity of the WT, HADHA Mut and HADHA KO CMs (Supplemental Figure 9H). We actually saw a depression of the glycolytic capacity in the HADHA Mut and KO CMs. This may suggest that glycogen stores are not being abnormally pulled on in this state. Furthermore, based on our scRNA-Seq data (Figure 5H-J) we did not see an up-regulation of glycogen pathways or glucose pathways in the HADHA Mut CMs. We did see however that the HADHA Mut CMs that are replicative have an increase in folate, methionine and cysteine metabolism.

Did the authors studied what happens if after 6-12 days treatment with FA media, they allow HADHAMut or HADHAKO CM to grow in non-FA culture media again? Does normalization occur? Are the observed changes permanent or all or partially each of the abnormalities normalize?

Response: We thank the reviewer for their question. We have now preformed this experiment where we have treated the HADHA Mut CMs for 6-days with Glc+FA and then removed the FA and treated the cells for another 6-days with just Glc. Our Seahorse Analysis showed that the 6D Glc+FA + 6D Glc cells still had depressed maximum OCR, ATP production and increased proton leak just like the HADHA Mut CMs 12D Glc+FA as compared to HADHA Mut CMs that had only been cultured in Glc media for 12D (Rebuttal Figure 4A-C).

Minor points:

Please define FPKM at line 192, because it is the first time the acronym appears
FPKM has now been defined.

MiR-452 OE up-regulation of electrophysiology shows a trend, but does not appear as convincing and compelling as in MiR-208b. Is there any statistical analysis showing significance for the effect of OE of MiR-452 on electrophysiology?

miR-452 was not statistically significant in its ability to increase cFPD. Please see reviewer #1 question #2 for our comments on miR-208b.

On line 163 the authors refer to six miRNAs, but in Supplemental figure E, they tested five (miR 200a is missing). Why RNA sequencing was not performed on miR-200a? Please reconcile. MiR-200a was examined a bit after those initial transcriptomics data were generated and we functionally found that miR-200a had a great increase on twitch force. As a result, we added

miR-200a into the microRNA maturation cocktail.

What was the molar ratio among miR of MIMAC?

We are utilizing a lentivirus for each OE or KO of a miR in MiMaC. Since the KO miRs are done with a gRNA that is targeted near the seed region of the miR, these should be non-targeting miRs after insertion and deletion errors are introduced. The OE miRs are made with the same backbone, same lentivirus protocol and used at the same ratio of concentrated virus.

At line 211 the authors specified the use of the MIMAC cocktail (Let7i OE, miR-452 OE, miR-122 KO, and miR-200a KO), but at line 249 the mentioned the chosen six miRs. Please reconcile Thank you for this clarification. We have now changed the wording to say screened instead of chosen, as we were looking into the predicted targets of all six miRs screened in the paper.

Why measured calcium transients were performed on HADHAMut but not on HADHAKO?
In trying to find a link between why in the clinic MTP deficient children sometimes present with sudden infant death syndrome, we thought that the precipitating cause could be due to cardiac arrhythmias that are somehow precipitating due to the mutations in HADHA. There are no clinical indications of a child born with a full KO of HADHA. In mice, a full KO of HADHA results in death of the animal 6-36 hours after birth due to unknown reasons [9]. As a result, the most clinically relevant line to study a potential pro-arrhythmic state in was the HADHA Mut CMs, which showed an intermediate pathological phenotype.

Why was the HADHAKO not assessed for proton leak rescue by SS-31?

Since the HADHA KO has no HADHA, the correct forms of cardiolipin could not be generated for SS-31 to bind to and help mitigate further potential decay of the mitochondria.

Please define ALCAT1

Acyl-CoA:lysocardiolipin acyltransferase-1 (ALCAT1) has now been defined in the text.

Reviewer #5 (Remarks to the Author):

In this manuscript, Miklas and colleagues present their evaluation of MTP disease mechanism as a result of HADHA mutation. They generate HADHA-deficient hiPSCs and improve their maturation by treating them with a combination of miRNA cocktail. They provide a detailed study on how their miRNA cocktail leads to maturation of hiPSC-cardiomyocytes through upregulation of HOPX. They show that this cocktail improves CM maturation and allows for FA metabolism through FAO as energy source, hence providing the appropriate tools to study how FAO may contribute to the pro-arrhythmia in HADHA-deficient CMs. They then show that FA challenged HADHA mutant CMs showed aberrant calcium handling, delayed repolarization and erratic beating suggesting a pro-arrhythmic state. They found that this pathologic finding was a result of mitochondrial dysfunction due to proton gradient loss and lack of normal cristae of the mitochondria. Very interestingly, they provide convincing evidence that HADHA knockout leads to a mis-regulation of cardiolipin homeostasis due to the reduction of tetra[18:2] cardiolipin species.

First, the authors should be congratulated for undertaking a tour de force study using meticulous research methods to explore how HADHA is required for FAO and cardiolipin remodeling. The authors use rigorous experimental techniques and well designed studies to provide a complete study. The manuscript is well written (although it may include some data that are not directly related to the final message of the paper). Overall, I believe this paper contributes significantly to our current knowledge in two major ways: i) providing a valuable tool for maturation of PSC-derived cardiomyocytes that can be used in a variety of studies (although this is based on altered expression of miRs and may not be applicable in every system) and ii) providing an in vitro model of human MTP deficient CMs to explore disease mechanism. This is an exciting and near-complete story that may benefit from a few changes.

1. The authors perform 4 different characterization/functional assays to assess the effect of OE or KO of their candidate miRs. They show miR 208-b overexpression increases CM area, miR-452 OE increased the corrected field potential duration, miR-200a KO resulted in an increase in force of contraction and KO of miR-122 resulted in increased O₂ consumption. However, for their bioinformatic analysis of candidate miRs, they chose to perform RNA sequencing on miR-378e OE, -208b OE, -452 OE, -122 KO or -205 KO. The rationale behind this experiment is not entirely clear. One would expect that gene expression analysis would be performed based on the altered expression of miRs that resulted in phenotypic changes (i.e. altered expression of miR 208-b, 200a, 452 and 122). RNAseq on miR 200 OE (which resulted in increased force generation) was not tested here. But interestingly, mir200a OE is included in their final cocktail.

Response: MiR-200a was examined a bit after those initial transcriptomics data were generated and we functionally found that miR-200a had a great increase on twitch force. As a result, we added miR-200a into the microRNA maturation cocktail.

2. Single cell RNA-Sequencing analysis of miR treated CM is interesting and intriguing. The authors have performed diligent analysis of their data to identify clusters according to CM maturity and show that this cluster enriched for MiMac treated CMs. This analysis would have

been significantly more convincing if this data was compared to fetal and adult cardiomyocytes (i.e. single cells RNA seq) to illustrate clustering of endogenous adult CMs with MiMac treated hiPSC-derived CMs.

Response: We thank the reviewer for this question. We found scRNA-Seq data of human fetal left ventricle [23] as a comparison to our scRNA-Seq hiPSC-CMs. There is currently no published human adult CM data. We did principal component analysis (PCA) of the human fetal ventricle single cell data. We used genes expressed in both our scRNA-seq data and the human fetal ventricle scRNA-seq data. In the PCA, PC2 (x-axis) represents the progression of fetal CM maturation. We then projected our single cell data to the PCA and found that our hiPSC-CMs clustered well with human fetal ventricle CMs at later time points among all CMs aged 5-24 weeks of age via principle component analysis (Rebuttal Figure 5A). When utilizing a density plot all single cells along PC2 maturation axis, our hiPSC-CMs were around a 6-week-old fetal CM (Rebuttal Figure 5B).

3. It is not clear whether Ca⁺⁺ handling and EP experiments were performed in the presence or absence of MiMAC, since the text and figures (4D-L) don't specify that.

Response: Thank you for this clarification point. When CMs were treated with MiMac, we explicitly state it in the graph legend as in Figure 4A. For the calcium transient and fluovolt experiments, we did not utilize MiMaC.

4. Why do authors perform their experiments using both HADHA mut and KO? In certain experimental models, it is not clear why they chose one over the other. Perhaps presenting data on one line may be a more cohesive story.

Response: In trying to find a link between why in the clinic MTP deficient children sometimes present with sudden infant death syndrome, we thought that the precipitating cause could be due to cardiac arrhythmias that are somehow precipitating due to the mutations in HADHA. There are no clinical indications of a child born with a full KO of HADHA. In mice, a full KO of HADHA results in death of the animal 6-36 hours after birth due to unknown reasons [9]. As a result, the most clinically relevant line to study a potential pro-arrhythmic state in was the HADHA Mut CMs, which showed an intermediate pathological phenotype. However, when trying to elucidate the novel function of HADHA as an acyl-transferase, the HADHA KO lines were the most insightful. We believe that the two lines help uncover various aspects of the clinical manifestation of MTP deficiency and also uncover novel basic biological questions situated around the protein HADHA.

5. The authors did not observe an accumulation of MLCL in the HADHA Mut CMs (normally seen with TAZ mutations). Based on this observation, they conclude that CL remodeling is the result of first processing by TAZ and then by HADHA. It may be helpful for the authors to include in the discussion the possibility that HADHA remodeling of MLCL to mature cardiolipin may (or may not) be independent of TAZ function.

Response: We agree with the reviewer and have now added a sentence to describe this.

The following text has been added to the manuscript:

These data suggest that CL remodeling is the result of first processing by TAZ and then by HADHA to generate tetra[18:2]-CL in human cardiomyocytes, however, it is still possible they are acting in parallel (Figure 7M).

Minor points:

Figure 1A is not referenced in the paper.

We thank the reviewer for finding this oversight. We have now added a reference to Figure 1A.

Are the values for CM area in figure 2C significant? From the figure it doesn't look like it. Did the authors compare the values for miR208 overexpression to miR205KO to show significance? It should be compared to control (i.e. empty virus). Additionally, the authors don't show the corrected field potential duration results for KO miRs (they only show overexpression data).

The over-expression miRs were based on the backbone of pLKO (pLKO-EV). The knockout miRs were based on the backbone of LentiCRISPRv2 (CRISPR/Cas9-EV). We compared each OE miR to pLKO-EV and each KO miR to CRISPR/Cas9-EV. Yes, miR-208b OE for cell area was compared to its control pLKO-EV and is significant. There is a good amount of overlap however, the average area for pLKO-EV is $2890\mu\text{m}^2$ while for miR-208b OE is $5801\mu\text{m}^2$. For the KO miRs, we found some great functional data pertaining to twitch force and metabolism for miRs-200a and -122 KO. The micro-electrode-array (MEA) plates are expensive and technically challenging to maintain monolayers of hPSC-CMs upon. As a result, we decided to leave those experiments are just the OE miRs.

What does EV stand for? I assume this is the control hiPSC line?

EV stands for empty vector (or empty virus as the reviewer noted in the above question). EV is our control virus that we use. Our control hiPSC line, that serves as an isogenic control to the HADHA modified lines, is the unmodified WTC #11 hiPSC line.

Does figure 3J refer to cardiomyocytes isolated from fetal and adult human tissues?

That is correct. This was commercially purchased human fetal and adult RNA from human fetal and adult ventricular tissue. We sent these samples for RNA-Sequencing in a previous study published by Kuppusamy *et al.* in PNAS [4]. We revisited that RNA-Seq data to examine the role of HOPX during human cardiomyocyte maturation.

On page 7, line 260, the authors state: "These data indicate that Let7i OE maturation does not govern HOPX cardiac maturation pathways." This observation is solely based on a single OE of let7i that showed no changes in HOPX expression pattern. This is a strong statement and can be toned down, since OE experiments cannot recapitulate the stepwise regulatory mechanisms involving HOPX.

We thank the reviewer for this point of clarification. We have now changed the text to state that, “These data suggest...” instead of saying indicate.

Rebuttal Letter Figures Captions

Rebuttal Figure 1: GO Terms for Intermediate Enriched Cluster 1. A) Intermediate GO terms.

Rebuttal Figure 2: Mitotracker Quantification of SS-31 treated cells. A) Mitotracker colocalization was not significantly different between the WT and HADHA 12D FA treated CMs when also treated with 12D of SS-31.

Rebuttal Figure 3: HADHA Enhancers. A) Plot showing enhancers around the transcription start site of HADHA. B) Plot showing enhancers along the gene of PKM.

Rebuttal Figure 4: Seahorse Analysis. 6D of FA treated followed by 6D of glucose treatment does not rescue the HADHA Mut depressed metabolic state. A) Maximum oxygen consumption rate. B) ATP Production. C) Proton Leak.

Rebuttal Figure 5: MiMaC Benchmarking against Human Fetal Ventricular CMs. A) PCA of our scRNA-Seq CMs (all conditions) with different weeks of human fetal ventricular CMs. B) Density plot showing the relative age of our hiPSC-CMs (dotted line) as compared to the human fetal ventricular samples.

References

1. Tse, G., *Mechanisms of cardiac arrhythmias*. J Arrhythm, 2016. **32**(2): p. 75-81.
2. Gama-Carvalho, M., J. Andrade, and L. Bras-Rosario, *Regulation of Cardiac Cell Fate by microRNAs: Implications for Heart Regeneration*. Cells, 2014. **3**(4): p. 996-1026.
3. Malizia, A.P. and D.Z. Wang, *MicroRNAs in cardiomyocyte development*. Wiley interdisciplinary reviews. Systems biology and medicine, 2011. **3**(2): p. 183-90.
4. Kuppusamy, K.T., et al., *Let-7 family of microRNA is required for maturation and adult-like metabolism in stem cell-derived cardiomyocytes*. Proc Natl Acad Sci U S A, 2015.
5. Ellwanger, D.C., et al., *Transcriptional Dynamics of Hair-Bundle Morphogenesis Revealed with CellTrails*. Cell Rep, 2018. **23**(10): p. 2901-2914 e14.
6. Hagenbuchner, J., et al., *Very long-/ and long Chain-3-Hydroxy Acyl CoA Dehydrogenase Deficiency correlates with deregulation of the mitochondrial fusion/fission machinery*. Sci Rep, 2018. **8**(1): p. 3254.
7. Friedman, C.E., et al., *Single-Cell Transcriptomic Analysis of Cardiac Differentiation from Human PSCs Reveals HOPX-Dependent Cardiomyocyte Maturation*. Cell Stem Cell, 2018. **23**(4): p. 586-598 e8.
8. Jain, R., et al., *HEART DEVELOPMENT. Integration of Bmp and Wnt signaling by Hopx specifies commitment of cardiomyoblasts*. Science, 2015. **348**(6242): p. aaa6071.
9. Ibdah, J.A., et al., *Lack of mitochondrial trifunctional protein in mice causes neonatal hypoglycemia and sudden death*. J Clin Invest, 2001. **107**(11): p. 1403-9.
10. Jones, P.M., et al., *Accumulation of free 3-hydroxy fatty acids in the culture media of fibroblasts from patients deficient in long-chain l-3-hydroxyacyl-CoA dehydrogenase: a useful diagnostic aid*. Clin Chem, 2001. **47**(7): p. 1190-4.
11. Szeto, H.H. and A.V. Birk, *Serendipity and the discovery of novel compounds that restore mitochondrial plasticity*. Clin Pharmacol Ther, 2014. **96**(6): p. 672-83.
12. Birk, A.V., et al., *The mitochondrial-targeted compound SS-31 re-energizes ischemic mitochondria by interacting with cardiolipin*. J Am Soc Nephrol, 2013. **24**(8): p. 1250-61.
13. Wajner, M. and A.U. Amaral, *Mitochondrial dysfunction in fatty acid oxidation disorders: insights from human and animal studies*. Biosci Rep, 2015. **36**(1): p. e00281.
14. Martinez, G., et al., *Plasma free fatty acids in mitochondrial fatty acid oxidation defects*. Clin Chim Acta, 1997. **267**(2): p. 143-54.
15. De Biase, I., et al., *Diagnosis, Treatment, and Clinical Outcome of Patients with Mitochondrial Trifunctional Protein/Long-Chain 3-Hydroxy Acyl-CoA Dehydrogenase Deficiency*. JIMD Rep, 2017. **31**: p. 63-71.
16. Mejia, E.M., et al., *Differential reduction in cardiac and liver monolysocardiolipin acyltransferase-1 and reduction in cardiac and liver tetralinoleoyl-cardiolipin in the alpha-subunit of trifunctional protein heterozygous knockout mice*. Biochem J, 2015. **471**(1): p. 123-9.
17. Mills, R.J., et al., *Functional screening in human cardiac organoids reveals a metabolic mechanism for cardiomyocyte cell cycle arrest*. Proc Natl Acad Sci U S A, 2017. **114**(40): p. E8372-E8381.
18. Ramachandra, C.J.A., et al., *Fatty acid metabolism driven mitochondrial bioenergetics promotes advanced developmental phenotypes in human induced pluripotent stem cell derived cardiomyocytes*. Int J Cardiol, 2018. **272**: p. 288-297.

19. Correia, C., et al., *Distinct carbon sources affect structural and functional maturation of cardiomyocytes derived from human pluripotent stem cells*. *Sci Rep*, 2017. **7**(1): p. 8590.
20. Listenberger, L.L., et al., *Triglyceride accumulation protects against fatty acid-induced lipotoxicity*. *Proc Natl Acad Sci U S A*, 2003. **100**(6): p. 3077-82.
21. Miller, T.A., et al., *Oleate prevents palmitate-induced cytotoxic stress in cardiac myocytes*. *Biochem Biophys Res Commun*, 2005. **336**(1): p. 309-15.
22. Tumova, J., et al., *Protective Effect of Unsaturated Fatty Acids on Palmitic Acid-Induced Toxicity in Skeletal Muscle Cells is not Mediated by PPARdelta Activation*. *Lipids*, 2015. **50**(10): p. 955-64.
23. Cui, Y., et al., *Single-Cell Transcriptome Analysis Maps the Developmental Track of the Human Heart*. *Cell Rep*, 2019. **26**(7): p. 1934-1950 e5.

A

Intermediate GO Terms

A

Mitotracker Quantification

■ HADHA 12DFA SS31

■ WT 12DFA SS31

A**B**
A Maximum OCR

- Mut 12D Glc
- Mut 12D FA
- Mut 6D Glc+FA
- Mut 6D Glc+FA + 6D Glc

B ATP Production

- Mut 12D Glc
- Mut 12D FA
- Mut 6D Glc+FA
- Mut 6D Glc+FA + 6D Glc

C Proton Leak

- Mut 12D Glc
- Mut 12D FA
- Mut 6D Glc+FA
- Mut 6D Glc+FA + 6D Glc

A**B**
Differentially Expressed Genes - Lower in Cluster 1 - Heart Contraction

genes	names	p_val	avg_logFC	percent.ex	percent.ex	p_val_adj
SLC8A1	solute carri	5.33E-75	-1.20134	0.65	0.982	1.8E-70
ATP2A2	ATPase sar	7.76E-72	-1.17391	0.774	0.991	2.62E-67
TTN	titin	3.38E-69	-1.15747	0.965	1	1.14E-64
MYBPC3	myosin bin	1.59E-61	-0.84871	0.862	0.994	5.35E-57
RYR2	ryanodine r	2.74E-60	-0.76605	0.318	0.939	9.22E-56
CACNA1C	calcium vol	1.78E-57	-0.7903	0.339	0.914	6.01E-53
MIR133A1	microRNA :	4.07E-53	-0.64425	0.138	0.768	1.37E-48
BVES	blood vesse	8.42E-53	-0.83238	0.491	0.951	2.84E-48
GJA1	gap junctio	1.33E-52	-0.85367	0.502	0.948	4.47E-48
CORIN	corin, serin	1.53E-50	-0.71965	0.477	0.966	5.14E-46
CACNB2	calcium vol	2.19E-49	-0.59322	0.307	0.853	7.37E-45
PKP2	plakophilin	6.06E-48	-0.63403	0.686	0.979	2.04E-43
EHD3	EH domain	4.06E-45	-0.63232	0.389	0.884	1.37E-40
ATP1B1	ATPase Na-	2.37E-44	-0.70401	0.498	0.963	7.98E-40
KCNH2	potassium	1.83E-40	-0.54772	0.399	0.902	6.17E-36
POPDC2	popeye doi	3.46E-39	-0.55975	0.83	0.988	1.17E-34
GJA5	gap junctio	4.21E-39	-0.65116	0.134	0.645	1.42E-34
RNF207	ring finger	4.9E-39	-0.54161	0.258	0.755	1.65E-34
MEF2A	myocyte er	1.45E-38	-0.56198	0.625	0.966	4.87E-34
ATP1A1	ATPase Na-	1.61E-38	-0.54023	0.42	0.939	5.43E-34
DSP	desmoplak	2.24E-38	-0.49494	0.435	0.942	7.54E-34
SNTA1	syntrophin	5.05E-38	-0.54873	0.565	0.942	1.7E-33
CAMK2D	calcium/cal	6.63E-37	-0.53677	0.622	0.972	2.23E-32
PRKACA	protein kin.	1.3E-36	-0.489	0.428	0.878	4.39E-32
AGT	angiotensir	7.22E-36	-0.38864	0.145	0.664	2.43E-31
MYH7	myosin, he	3.82E-35	-0.69804	0.986	0.997	1.29E-30
CACNA1G	calcium vol	4.59E-35	-0.34782	0.078	0.55	1.55E-30
TMEM65	transmemt	2.98E-34	-0.40538	0.279	0.777	1E-29
CHRM2	cholinergic	4.23E-33	-0.35941	0.247	0.758	1.43E-28
ANK2	ankyrin 2, r	1.09E-32	-0.36044	0.283	0.789	3.66E-28
JUP	junction pl:	1.15E-31	-0.41304	0.456	0.911	3.86E-27
GATA4	GATA bindi	3.65E-31	-0.40898	0.403	0.862	1.23E-26
ATP2B4	ATPase pla	6.79E-31	-0.38552	0.329	0.838	2.29E-26
KCNJ5	potassium	1.09E-30	-0.27878	0.078	0.517	3.66E-26
CXADR	coxsackie v	3.28E-30	-0.39845	0.389	0.914	1.11E-25
DMD	dystrophin	9.73E-30	-0.41698	0.375	0.865	3.28E-25
AKAP9	A-kinase ar	4.6E-28	-0.34464	0.304	0.774	1.55E-23
HRC	histidine ric	2.4E-26	-0.28287	0.226	0.688	8.08E-22
CELF2	CUGBP, Ela	2.89E-26	-0.42507	0.512	0.954	9.75E-22
KCNJ3	potassium	9.65E-24	-0.25238	0.159	0.56	3.25E-19
VEGFB	vascular en	2.06E-23	-0.31166	0.431	0.862	6.93E-19
FGF12	fibroblast g	4.41E-22	-0.38319	0.445	0.85	1.49E-17
MYH6	myosin, he	1.42E-21	-0.6383	0.721	0.963	4.77E-17
SCN5A	sodium vol	2.86E-21	-0.26147	0.311	0.746	9.62E-17
HEY2	hes related	5.68E-21	-0.32654	0.148	0.52	1.91E-16
HSPB7	heat shock	3.04E-20	-0.3602	0.986	1	1.03E-15

Differentially Expressed Genes - Lower in Cluster 1 - Ion Channels

genes	names	p_val	avg_logFC	percent.ex	percent.ex	p_val_adj
RYR2	ryanodine r	2.74E-60	-0.76605	0.318	0.939	9.22E-56
CACNA1C	calcium vol	1.78E-57	-0.7903	0.339	0.914	6.01E-53
CACNB2	calcium vol	2.19E-49	-0.59322	0.307	0.853	7.37E-45
KCNH2	potassium i	1.83E-40	-0.54772	0.399	0.902	6.17E-36
FXVD6	FXVD doma	2.67E-35	-0.539	0.587	0.954	9E-31
CACNA1G	calcium vol	4.59E-35	-0.34782	0.078	0.55	1.55E-30
ITGAV	integrin sul	2.94E-32	-0.43543	0.466	0.942	9.91E-28
TMEM63B	transmemt	6.71E-32	-0.35501	0.24	0.725	2.26E-27
KCNJ5	potassium i	1.09E-30	-0.27878	0.078	0.517	3.66E-26
CACHD1	cache dom	1.48E-29	-0.31438	0.166	0.618	4.98E-25
SLC1A4	solute carri	7.56E-27	-0.31912	0.127	0.526	2.55E-22
AQP1	aquaporin i	8.01E-26	-0.36063	0.29	0.722	2.7E-21
JPH2	junctionphi	7.36E-25	-0.37086	0.481	0.875	2.48E-20
KCNJ3	potassium i	9.65E-24	-0.25238	0.159	0.56	3.25E-19
ANXA6	annexin A6	1.12E-23	-0.38187	0.572	0.945	3.78E-19
ANO6	anoctamin	1.36E-21	-0.25462	0.261	0.664	4.6E-17
SCN5A	sodium vol	2.86E-21	-0.26147	0.311	0.746	9.62E-17

Differentially Expressed Genes - Higher in Cluster 1 - OXPHOS ATP

genes	names	p_val	avg_logFC	percent.ex	percent.ex	p_val_adj
NDUFS5	NADH:ubiq	1.31E-41	0.711229	0.989	0.991	4.43E-37
ATP5E	ATP syntha	1.02E-39	0.531332	1	1	3.42E-35
ATP5J2	ATP syntha	2.63E-38	0.832612	1	0.994	8.87E-34
COX7A2	cytochrom	2.92E-38	0.668872	0.993	0.985	9.85E-34
UQCRCQ	ubiquinol-c	2.34E-37	0.609566	1	0.997	7.9E-33
ATP5L	ATP syntha	1.09E-36	0.634258	1	1	3.66E-32
TMSB4X	thymosin b	6.22E-36	1.685419	0.982	0.991	2.1E-31
UQCR10	ubiquinol-c	3.85E-34	0.635831	1	0.994	1.3E-29
COX5B	cytochrom	7.85E-33	0.620659	0.996	0.997	2.64E-28
NDUFA12	NADH:ubiq	1.38E-28	0.705236	0.89	0.905	4.65E-24
ATP5H	ATP syntha	2.78E-28	0.765748	0.929	0.927	9.38E-24
ATPIF1	ATPase inh	1.59E-27	0.676763	0.975	0.997	5.37E-23
COX6A1	cytochrom	3.16E-27	0.529963	0.958	0.979	1.07E-22
NDUFA4	NDUFA4, r	4.37E-27	0.556733	1	1	1.47E-22
NDUFA1	NADH:ubiq	2.83E-26	0.557075	0.993	0.994	9.55E-22
NDUFA2	NADH:ubiq	3.98E-26	0.500396	0.936	0.969	1.34E-21
NDUFB3	NADH:ubiq	2.03E-25	0.632211	0.943	0.954	6.86E-21
NDUFAB1	NADH:ubiq	2.64E-25	0.56778	0.933	0.939	8.9E-21
NDUFB4	NADH:ubiq	1.07E-24	0.693149	0.883	0.924	3.62E-20
NDUFA6	NADH:ubiq	2.87E-23	0.653197	0.954	0.933	9.69E-19
GAPDH	glyceraldeh	3.17E-23	0.398062	1	1	1.07E-18
ATP5J	ATP syntha	4.89E-23	0.53576	0.993	0.991	1.65E-18
NDUFA7	NADH:ubiq	3.17E-22	0.544542	0.866	0.823	1.07E-17
COX7A2L	cytochrom	1.52E-21	0.529573	0.859	0.917	5.12E-17
ATP5G1	ATP syntha	1.54E-21	0.606061	0.979	0.985	5.2E-17
PARK7	Parkinsonis	8.36E-21	0.478177	0.901	0.936	2.82E-16
CYCS	cytochrom	9.03E-21	0.584284	0.982	0.994	3.04E-16
ATP5G3	ATP syntha	1.13E-20	0.611176	0.989	1	3.82E-16
ATP5G2	ATP syntha	4.61E-20	0.48762	0.989	0.994	1.55E-15
ATP5I	ATP syntha	7.6E-20	0.398827	0.996	1	2.56E-15
NDUFV2	NADH:ubiq	8.29E-20	0.570915	0.873	0.896	2.79E-15
ATP5L2	ATP syntha	2.24E-19	0.684577	0.77	0.722	7.56E-15
COX7C	cytochrom	1.37E-18	0.45738	1	1	4.61E-14
UQCRB	ubiquinol-c	1.98E-18	0.46815	0.996	0.997	6.68E-14
NDUFC1	NADH:ubiq	2.08E-18	0.586883	0.954	0.954	7.02E-14
ATP5O	ATP syntha	2.27E-18	0.614778	0.961	0.982	7.65E-14
NDUFB2	NADH:ubiq	3.11E-18	0.428957	0.982	1	1.05E-13
NDUFB1	NADH:ubiq	2.18E-15	0.445062	0.989	0.997	7.36E-11
GUK1	guanylate k	6.83E-14	0.345846	0.958	0.985	2.3E-09
COX5A	cytochrom	7.76E-14	0.478355	0.993	0.988	2.61E-09
ATP5C1	ATP syntha	1.29E-13	0.489841	0.859	0.936	4.34E-09
NDUFB7	NADH:ubiq	3.97E-13	0.420199	0.929	0.969	1.34E-08
NDUFB10	NADH:ubiq	8.27E-13	0.389434	0.898	0.924	2.79E-08
ATP5D	ATP syntha	2.91E-12	0.383528	0.972	0.994	9.8E-08
NDUFB8	NADH:ubiq	3.95E-12	0.44444	0.855	0.924	1.33E-07

COX4I1	cytochrome	4.36E-12	0.322652	0.993	1	1.47E-07
NDUFA8	NADH:ubiq	1.35E-11	0.501256	0.816	0.89	4.54E-07
NDUFS4	NADH:ubiq	1.95E-11	0.516747	0.703	0.722	6.56E-07
ALDOA	aldolase, fr	2.95E-11	0.416655	0.986	0.994	9.94E-07
NDUFS8	NADH:ubiq	5E-11	0.36547	0.89	0.96	1.68E-06
NDUFA5	NADH:ubiq	7.19E-11	0.466109	0.862	0.92	2.42E-06
TPI1	triosephosj	8.41E-11	0.39451	0.972	0.985	2.83E-06
NDUFS6	NADH:ubiq	2.2E-10	0.355972	0.986	0.994	7.43E-06
UQCC2	ubiquinol-c	4.97E-10	0.3928	0.852	0.896	1.67E-05
NDUFS7	NADH:ubiq	1.02E-09	0.423658	0.845	0.917	3.44E-05
NDUFC2	NADH:ubiq	1.66E-09	0.365474	0.781	0.832	5.59E-05
NDUFB5	NADH:ubiq	2.43E-09	0.398454	0.88	0.954	8.18E-05
NDUFB6	NADH:ubiq	4.73E-09	0.398092	0.784	0.81	0.000159
UQCRH	ubiquinol-c	1.02E-08	0.261468	0.993	1	0.000343
CHCHD10	coiled-coil-	1.91E-07	0.411741	0.996	0.997	0.006423
UQCRHL	ubiquinol-c	2.99E-07	0.386767	0.827	0.859	0.01008
UQCRFS1	ubiquinol-c	3.76E-07	0.336903	0.912	0.979	0.01268

Differentially Expressed Genes - Lower in Cluster 1 Overlapped with Fetal Vs. Adult Heart - Heart Contraction

genes	names	p_val	avg_logFC	percent.ex	percent.ex	p_val_adj	id	gene.symb	gene.name	Fetal	Adult	foldChange	log2FoldCh	pval	padj
SLC8A1	solute carri	5.33E-75	-1.20134	0.65	0.982	1.8E-70	ENSG00000100000	SLC8A1	solute carri	18629.56	3418.825	0.18356	-2.44568	1.45E-06	3.62E-05
MIR133A1	microRNA	4.07E-53	-0.64425	0.138	0.768	1.37E-48	ENSG00000100000	MIR133A1	microRNA	721.3507	76.18485	0.106852	-3.22631	1.85E-14	1.83E-12
GJA1	gap junctio	1.33E-52	-0.85367	0.502	0.948	4.47E-48	ENSG00000100000	GJA1	gap junctio	7364.852	2672.704	0.362986	-1.46201	0.000857	0.00799
CORIN	corin, serin	1.53E-50	-0.71965	0.477	0.966	5.14E-46	ENSG00000100000	CORIN	corin, serin	4506.19	854.3647	0.189778	-2.39762	1.37E-06	3.44E-05
CACNB2	calcium vol	2.19E-49	-0.59322	0.307	0.853	7.37E-45	ENSG00000100000	CACNB2	calcium vol	1254.973	436.2998	0.348176	-1.52211	0.000305	0.003532
RNF207	ring finger	4.9E-39	-0.54161	0.258	0.755	1.65E-34	ENSG00000100000	RNF207	ring finger	4600.896	1342.509	0.291947	-1.77622	1.17E-05	0.000227
MEF2A	myocyte er	1.45E-38	-0.56198	0.625	0.966	4.87E-34	ENSG00000100000	MEF2A	myocyte er	5168.119	1765.645	0.341769	-1.54891	0.004373	0.027074
CACNA1G	calcium vol	4.59E-35	-0.34782	0.078	0.55	1.55E-30	ENSG00000100000	CACNA1G	calcium vol	502.5989	12.67304	0.027151	-5.20287	0.001271	0.010877
KCNJ5	potassium	1.09E-30	-0.27878	0.078	0.517	3.66E-26	ENSG00000100000	KCNJ5	potassium	823.3023	157.9689	0.192853	-2.37443	9.41E-07	2.49E-05
HRC	histidine ric	2.4E-26	-0.28287	0.226	0.688	8.08E-22	ENSG00000100000	HRC	histidine ric	2706.624	11557.37	4.268822	2.093838	0.000553	0.005642
FGF12	fibroblast g	4.41E-22	-0.38319	0.445	0.85	1.49E-17	ENSG00000100000	FGF12	fibroblast g	6221.831	1907.375	0.306673	-1.70523	0.002467	0.018005
MYH6	myosin, he	1.42E-21	-0.6383	0.721	0.963	4.77E-17	ENSG00000100000	MYH6	myosin, he	33336.8	188727.1	5.661085	2.501079	1.37E-05	0.000259
HEY2	hes related	5.68E-21	-0.32654	0.148	0.52	1.91E-16	ENSG00000100000	HEY2	hes related	1312.33	530.8064	0.40493	-1.30426	0.001993	0.015355
HSPB7	heat shock	3.04E-20	-0.3602	0.986	1	1.03E-15	ENSG00000100000	HSPB7	heat shock	25734.65	112482.1	4.370712	2.127868	0.000847	0.007916

Differentially Expressed Genes - Lower in Cluster 1 Overlapped with Fetal Vs. Adult Heart - Ion Channels

genes	names	p_val	avg_logFC	percent.ex	percent.ex	p_val_adj	id	gene.symb	gene.name	Fetal	Adult	foldChange	log2FoldCh	pval	padj
CACNB2	calcium vol	2.19E-49	-0.59322	0.307	0.853	7.37E-45	ENSG00000108000	CACNB2	calcium vol	1254.973	436.2998	0.348176	-1.52211	0.000305	0.003532
FXYP6	FXYP dom	2.67E-35	-0.539	0.587	0.954	9E-31	ENSG00000108000	FXYP6	FXYP dom	1063.827	166.2955	0.15711	-2.67015	1.8E-05	0.000329
CACNA1G	calcium vol	4.59E-35	-0.34782	0.078	0.55	1.55E-30	ENSG00000108000	CACNA1G	calcium vol	502.5989	12.67304	0.027151	-5.20287	0.001271	0.010877
KCNJ5	potassium	1.09E-30	-0.27878	0.078	0.517	3.66E-26	ENSG00000108000	KCNJ5	potassium	823.3023	157.9689	0.192853	-2.37443	9.41E-07	2.49E-05
CACHD1	cache dom	1.48E-29	-0.31438	0.166	0.618	4.98E-25	ENSG00000108000	CACHD1	cache dom	230.6388	84.73474	0.370123	-1.43392	0.002974	0.020679
AQP1	aquaporin	8.01E-26	-0.36063	0.29	0.722	2.7E-21	ENSG00000108000	AQP1	aquaporin	2204.601	5157.646	2.338884	1.225821	0.009515	0.047596

Differentially Expressed Genes - Higher in Cluster 1 Overlapped with Fetal Vs. Adult Heart - OXPHOS ATP

genes	names	p_val	avg_logFC	percent.ex	percent.ex	p_val_adj	id	gene.symb	gene.name	Fetal	Adult	foldChange	log2FoldCh	pval	padj
COX7A2	cytochrom	2.92E-38	0.668872	0.993	0.985	9.85E-34	ENSG00000100000	COX7A2	cytochrom	723.7494	3014.088	4.160181	2.056646	0.001507	0.012405
UQCRQ	ubiquinol-c	2.34E-37	0.609566	1	0.997	7.9E-33	ENSG00000100000	UQCRQ	ubiquinol-c	506.5544	1851.315	3.64949	1.867695	0.001981	0.015273
UQCR10	ubiquinol-c	3.85E-34	0.635831	1	0.994	1.3E-29	ENSG00000100000	UQCR10	ubiquinol-c	823.6615	4234.651	5.13623	2.36071	0.000627	0.00621
COX5B	cytochrom	7.85E-33	0.620659	0.996	0.997	2.64E-28	ENSG00000100000	COX5B	cytochrom	1968.686	10956.92	5.563282	2.475936	0.000488	0.005097
NDUFA12	NADH:ubiq	1.38E-28	0.705236	0.89	0.905	4.65E-24	ENSG00000100000	NDUFA12	NADH:ubiq	649.1606	2808.175	4.32074	2.111278	0.001423	0.011865
NDUFB3	NADH:ubiq	2.03E-25	0.632211	0.943	0.954	6.86E-21	ENSG00000100000	NDUFB3	NADH:ubiq	274.2414	1227.912	4.464851	2.158612	0.000213	0.002619
NDUFAB1	NADH:ubiq	2.64E-25	0.56778	0.933	0.939	8.9E-21	ENSG00000100000	NDUFAB1	NADH:ubiq	1002.774	2374.007	2.366077	1.242497	0.005476	0.031895
NDUFB4	NADH:ubiq	1.07E-24	0.693149	0.883	0.924	3.62E-20	ENSG00000100000	NDUFB4	NADH:ubiq	584.2673	2083.309	3.561295	1.832402	0.002071	0.015797
ATP5G1	ATP syntha	1.54E-21	0.606061	0.979	0.985	5.2E-17	ENSG00000100000	ATP5G1	ATP syntha	806.2469	3423.783	4.242548	2.084931	0.000753	0.007197
CYCS	cytochrom	9.03E-21	0.584284	0.982	0.994	3.04E-16	ENSG00000100000	CYCS	cytochrom	3305.486	9934.506	3.004854	1.587295	0.005885	0.033633
COX5A	cytochrom	7.76E-14	0.478355	0.993	0.988	2.61E-09	ENSG00000100000	COX5A	cytochrom	1873.359	6600.358	3.521928	1.816365	0.00167	0.013386
ATP5C1	ATP syntha	1.29E-13	0.489841	0.859	0.936	4.34E-09	ENSG00000100000	ATP5C1	ATP syntha	2748.182	8464.374	3.079234	1.622572	0.005296	0.031138
NDUFB7	NADH:ubiq	3.97E-13	0.420199	0.929	0.969	1.34E-08	ENSG00000100000	NDUFB7	NADH:ubiq	592.3335	3340.283	5.631374	2.493487	9.58E-05	0.001348
NDUFB10	NADH:ubiq	8.27E-13	0.389434	0.898	0.924	2.79E-08	ENSG00000100000	NDUFB10	NADH:ubiq	854.959	4083.068	4.771335	2.254393	0.000339	0.003834
ATP5D	ATP syntha	2.91E-12	0.383528	0.972	0.994	9.8E-08	ENSG00000100000	ATP5D	ATP syntha	944.7875	4740.245	5.013013	2.325678	0.000332	0.003777
COX4I1	cytochrom	4.36E-12	0.322652	0.993	1	1.47E-07	ENSG00000100000	COX4I1	cytochrom	3480.465	15060.95	4.326325	2.113142	0.001137	0.009978
NDUFA8	NADH:ubiq	1.35E-11	0.501256	0.816	0.89	4.54E-07	ENSG00000100000	NDUFA8	NADH:ubiq	884.4018	3176.541	3.588812	1.843506	0.004776	0.028874
NDUFS8	NADH:ubiq	5E-11	0.36547	0.89	0.96	1.68E-06	ENSG00000100000	NDUFS8	NADH:ubiq	595.6282	3135.951	5.257799	2.394459	0.000113	0.001556
UQCRH	ubiquinol-c	1.02E-08	0.261468	0.993	1	0.000343	ENSG00000100000	UQCRH	ubiquinol-c	968.3037	2849.215	2.940477	1.55605	0.007261	0.039063
UQCRHL	ubiquinol-c	2.99E-07	0.386767	0.827	0.859	0.01008	ENSG00000100000	UQCRHL	ubiquinol-c	269.6626	1111.47	4.110175	2.0392	0.000608	0.006079
UQCRRF1	ubiquinol-c	3.76E-07	0.336903	0.912	0.979	0.01268	ENSG00000100000	UQCRRF1	ubiquinol-c	1042.274	2917.051	2.797014	1.483888	0.000817	0.007666

Significantly Changed Let7i Targets in Cluster 1 vs 0

genes	names	p_val	avg_logFC	percent.ex	percent.ex	p_val_adj
MYOCD	myocardin	4.74E-53	-0.6836	0.399	0.951	1.6E-48
EMILIN2	elastin mic	1.93E-49	-0.5764	0.265	0.856	6.49E-45
PABPN1	poly(A) bin	1.09E-43	-0.57053	0.452	0.933	3.67E-39
MXRA7	matrix rem	1.07E-40	-0.56152	0.555	0.966	3.61E-36
FIGN	fidgetin, mi	7.16E-39	-0.41535	0.184	0.688	2.41E-34
STRN	striatin	8.84E-39	-0.44051	0.251	0.771	2.98E-34
CELF1	CUGBP, Ela	1.17E-37	-0.48965	0.357	0.887	3.93E-33
ADIPOR2	adiponectin	1.28E-35	-0.36054	0.276	0.786	4.32E-31
ATG9A	autophagy	1.06E-31	-0.37916	0.279	0.765	3.57E-27
MSI2	musashi RN	6.47E-31	-0.37568	0.364	0.853	2.18E-26
ATXN2	ataxin 2	2.89E-30	-0.3103	0.251	0.743	9.73E-26
COX6B1	cytochrome	3.12E-28	0.606993	1	1	1.05E-23
IPO9	importin 9	3.13E-26	-0.32376	0.332	0.752	1.05E-21
SUMO1	small ubiqu	3.65E-26	0.629178	0.845	0.884	1.23E-21
C19orf53	chromosom	1.43E-25	0.524641	0.915	0.969	4.82E-21
VCL	vinculin	1.82E-25	-0.37864	0.452	0.924	6.13E-21
NSD1	nuclear rec	4.61E-24	-0.30219	0.332	0.765	1.55E-19
MDM4	MDM4, p53	6.42E-24	-0.26955	0.272	0.673	2.16E-19
ATP6V1G1	ATPase H+	1.56E-23	0.535994	0.929	0.963	5.26E-19
WASL	Wiskott-Alk	1.85E-23	-0.27207	0.371	0.817	6.25E-19
IGF2BP1	insulin like	1.25E-22	-0.25927	0.226	0.627	4.23E-18
ZCCHC3	zinc finger	2.19E-22	-0.26502	0.254	0.642	7.39E-18
RRAD	RRAD, Ras	5.3E-22	-0.29221	0.237	0.654	1.79E-17
TSC22D2	TSC22 dom	7.11E-22	-0.29769	0.336	0.761	2.4E-17
GNG5	G protein s	1.97E-21	0.426402	0.993	1	6.63E-17
IGF1R	insulin like	4.24E-21	-0.2679	0.297	0.709	1.43E-16
KDM5B	lysine dem	5.6E-21	-0.29515	0.339	0.746	1.89E-16
ATP6V1F	ATPase H+	1.66E-20	0.505813	0.912	0.942	5.59E-16
PEG10	paternally	8.87E-20	-0.45867	0.36	0.731	2.99E-15
EPHA4	EPH recept	3.61E-19	-0.26137	0.191	0.541	1.22E-14
GOLGA4	golgin A4	3.09E-18	-0.25668	0.378	0.807	1.04E-13
DVL3	dishevelled	6.54E-18	-0.26383	0.413	0.798	2.2E-13
SOD2	superoxide	1.38E-16	-0.27081	0.643	0.966	4.65E-12
RDX	radixin	1.88E-16	-0.27844	0.604	0.942	6.32E-12
RWDD1	RWD doma	7.49E-08	0.331717	0.742	0.817	0.002523

Significantly Changed miR-452 Targets in Cluster 1 vs 0

genes	names	p_val	avg_logFC	percent.ex	percent.ex	p_val_adj
NAV1	neuron nav	1.78E-77	-1.1612	0.636	0.979	6.01E-73
WNK1	WNK lysine	1.63E-46	-0.61463	0.452	0.957	5.49E-42
LAMC1	laminin suk	1.9E-38	-0.49132	0.36	0.924	6.4E-34
EPM2AIP1	EPM2A inte	6.24E-37	-0.43696	0.364	0.887	2.1E-32
ITGA9	integrin sul	9.46E-35	-0.35858	0.131	0.633	3.19E-30
ATP2B4	ATPase pla	6.79E-31	-0.38552	0.329	0.838	2.29E-26
SUB1	SUB1 homc	2.12E-29	0.581599	0.954	0.976	7.13E-25
CDC14B	cell divisor	5.52E-28	-0.29425	0.198	0.627	1.86E-23
BMPR2	bone morp	5.41E-27	-0.3566	0.357	0.844	1.82E-22
CDK6	cyclin-depe	6.09E-27	-0.40536	0.473	0.936	2.05E-22
MIDN	midnolin	6.07E-26	-0.29762	0.375	0.85	2.04E-21
KIAA1715	KIAA1715	1.06E-24	-0.29692	0.293	0.737	3.56E-20
CLASP1	cytoplasmic	5.51E-24	-0.26356	0.191	0.575	1.86E-19
DSTN	destrin, act	1.17E-23	0.518252	0.996	0.997	3.95E-19
TCF4	transcriptic	1.66E-22	-0.28978	0.233	0.63	5.59E-18
HSBP1	heat shock	7.78E-22	0.524365	0.951	0.972	2.62E-17
PTPRM	protein tyr	8.99E-22	-0.28226	0.24	0.612	3.03E-17
ATP5G3	ATP syntha	1.13E-20	0.611176	0.989	1	3.82E-16
RPS16	ribosomal p	2.43E-20	0.352894	1	0.994	8.2E-16
RPL7	ribosomal p	1.53E-18	0.42334	1	1	5.17E-14
CAPRIN1	cell cycle a	1.03E-15	-0.27295	0.481	0.869	3.47E-11
EIF3E	eukaryotic	9.7E-11	0.502635	0.845	0.927	3.27E-06

Significantly Changed Let7i Targets in Cluster 1 vs 0 Overlapped with Fetal vs Adult

genes	names	p_val	avg_logFC	percent.ex	percent.ex	p_val_adj	id	gene.symb	gene.name	Fetal	Adult	foldChange	log2FoldCh	pval	padj
MYOCD	myocardin	4.74E-53	-0.6836	0.399	0.951	1.6E-48	ENSG00000108000	MYOCD	myocardin	1937.508	641.7602	0.331575	-1.59259	7.83E-05	0.001136
EMILIN2	elastin mic	1.93E-49	-0.5764	0.265	0.856	6.49E-45	ENSG00000108000	EMILIN2	elastin mic	2721.661	885.5974	0.325636	-1.61867	0.000737	0.007088
FIGN	fidgetin, m	7.16E-39	-0.41535	0.184	0.688	2.41E-34	ENSG00000108000	FIGN	fidgetin, m	957.4971	333.7283	0.349222	-1.51778	1.1E-05	0.000215
ATG9A	autophagy	1.06E-31	-0.37916	0.279	0.765	3.57E-27	ENSG00000108000	ATG9A	autophagy	970.3816	2737.42	2.819098	1.495234	0.003015	0.020863
COX6B1	cytochrom	3.12E-28	0.606993	1	1	1.05E-23	ENSG00000108000	COX6B1	cytochrom	1668.503	6544.658	3.920723	1.97112	0.004759	0.028797
C19orf53	chromosom	1.43E-25	0.524641	0.915	0.969	4.82E-21	ENSG00000108000	C19orf53	chromosor	377.2217	1545.248	4.088205	2.031468	0.001509	0.012419
NSD1	nuclear rec	4.61E-24	-0.30219	0.332	0.765	1.55E-19	ENSG00000108000	NSD1	nuclear rec	2243.686	997.2958	0.444737	-1.16897	0.005025	0.029988
IGF2BP1	insulin like	1.25E-22	-0.25927	0.226	0.627	4.23E-18	ENSG00000108000	IGF2BP1	insulin like	1061.005	0.652696	0.001556	-9.32775	4.95E-95	1.61E-90
TSC22D2	TSC22 dom	7.11E-22	-0.29769	0.336	0.761	2.4E-17	ENSG00000108000	TSC22D2	TSC22 dom	1606.776	685.0098	0.426682	-1.22877	0.00817	0.042666
IGF1R	insulin like	4.24E-21	-0.2679	0.297	0.709	1.43E-16	ENSG00000108000	IGF1R	insulin like	2195.459	907.549	0.413643	-1.27354	0.007068	0.038319
KDM5B	lysine dem	5.6E-21	-0.29515	0.339	0.746	1.89E-16	ENSG00000108000	KDM5B	lysine dem	1553.809	267.9864	0.173003	-2.53113	1.55E-08	6.3E-07
ATP6V1F	ATPase H+	1.66E-20	0.505813	0.912	0.942	5.59E-16	ENSG00000108000	ATP6V1F	ATPase H+	166.9243	605.2684	3.610366	1.852145	0.003237	0.02195
PEG10	paternally	8.87E-20	-0.45867	0.36	0.731	2.99E-15	ENSG00000108000	PEG10	paternally	2118.177	53.27242	0.02561	-5.28714	5.65E-33	3.17E-30

Significantly Changed miR-452 Targets in Cluster 1 vs 0 Overlapped with Fetal vs Adult

genes	names	p_val	avg_logFC	percent.ex	percent.ex	p_val_adj	id	gene.symb	gene.name	Fetal	Adult	foldChange	log2FoldCh	pval	padj
NAV1	neuron nav	1.78E-77	-1.1612	0.636	0.979	6.01E-73	ENSG00000108080	NAV1	neuron nav	12450.89	2522.497	0.20266	-2.30287	4.56E-08	1.66E-06
LAMC1	laminin sut	1.9E-38	-0.49132	0.36	0.924	6.4E-34	ENSG00000108080	LAMC1	laminin sut	13117.99	4648.228	0.354389	-1.49659	0.003214	0.021834
EPM2AIP1	EPM2A intr	6.24E-37	-0.43696	0.364	0.887	2.1E-32	ENSG00000108080	EPM2AIP1	EPM2A intr	1887.539	755.8982	0.400785	-1.3191	0.001067	0.009494
CDK6	cyclin-depe	6.09E-27	-0.40536	0.473	0.936	2.05E-22	ENSG00000108080	CDK6	cyclin-depe	3088.466	419.6851	0.136168	-2.87654	3.88E-10	2.1E-08

REVIEWERS' COMMENTS:

Reviewer #1 (Remarks to the Author):

The authors have addressed all my questions.

Reviewer #2 (Remarks to the Author):

In my previous analysis I stated this work investigated an important biomedical problem in children because the cause of SIDS still remains unknown. SIDS is the third leading cause of death in children less than one year old. This study provides potential relevant and novel information for the potential treatment of MTFP-deficient newborns.

In general, the article is clear and well presented. The authors presented an updated, deep and critical revision of the state of the art and the working hypothesis is conceptual novel. The quality of the experimental design is high and broad methodologies and approaches were used to solve the proposed problem. The results are clear and relevant and the conclusions are adequate and based on the results.

I found two major criticisms: a) although the work provides new and interesting insights, the main question is how these in vitro studies mimics the real clinical situation. Some key findings should be tested in human samples. b) Calcium plays a critical role in mitochondrial energy metabolism due to some key enzyme activities are Ca²⁺-dependent. In my opinion it is important to know if cytosolic Ca²⁺ disturbances described in the MS are also accompanied of alteration in mitochondrial Ca²⁺ and also explained impaired mitochondrial energy metabolism.

In this article, the authors made an important effort to completely address these two major points, providing new data and discussion. The present article has significantly improved. I do not have any additional comments.

Reviewer #3 (Remarks to the Author):

None

Reviewer #4 (Remarks to the Author):

The authors have adequately answered to the reviewers' comments

Reviewer #5 (Remarks to the Author):

The authors have addressed most of the issues raised by this reviewer. The addition of the new iPSC line with founder HADHA mutation and characterization of CMs derived from this line significantly strengthens the manuscript.

However, the first issue in my review is not adequately addressed. The explanation provided is brief and not convincing. If there is any reason for how this approach was pursued, it should be clearly described in the manuscript and the reason for including certain miRs in the cocktail should be discussed (regardless of whether or not they were discovered after the initial transcriptomic data).

Reviewer #5 (Remarks to the Author):

The authors have addressed most of the issues raised by this reviewer. The addition of the new iPSC line with founder HADHA mutation and characterization of CMs derived from this line significantly strengthens the manuscript.

However, the first issue in my review is not adequately addressed. The explanation provided is brief and not convincing. If there is any reason for how this approach was pursued, it should be clearly described in the manuscript and the reason for including certain miRs in the cocktail should be discussed (regardless of whether or not they were discovered after the initial transcriptomic data).

Reviewer's first issue was:

1. The authors perform 4 different characterization/functional assays to assess the effect of OE or KO of their candidate miRs. They show miR 208-b overexpression increases CM area, miR-452 OE increased the corrected field potential duration, miR-200a KO resulted in an increase in force of contraction and KO of miR-122 resulted in increased O₂ consumption. However, for their bioinformatic analysis of candidate miRs, they chose to perform RNA sequencing on miR-378e OE, -208b OE, -452 OE, -122 KO or -205 KO. The rationale behind this experiment is not entirely clear. One would expect that gene expression analysis would be performed based on the altered expression of miRs that resulted in phenotypic changes (i.e. altered expression of miR 208-b, 200a, 452 and 122). RNAseq on miR 200 OE (which resulted in increased force generation) was not tested here. But interestingly, mir200a OE is included in their final cocktail. **Response:** MiR-200a was examined a bit after those initial transcriptomics data were generated and we functionally found that miR-200a had a great increase on twitch force. As a result, we added miR-200a into the microRNA maturation cocktail.

We thank the reviewer for their comment and apologize for not adequately answering their concern with our initial answer. From the RNA-Seq, the main take away was that the miRs we were testing were perturbing cardiac specific pathways. This helped give us confidence that our initial microRNA sequencing screen, from a heterogeneous tissue (human ventricular myocardium) was yielding microRNAs that were most likely found within the myocytes and not the non-myocytes when examining the highest up- and down-regulated miRs. Moreover, the RNA-Seq provided justification for the reason to pursue utilizing and the need to utilize multiple microRNAs. We showed that each microRNA we tested and sent for sequencing yielded a different effect on the transcriptome, effecting different cardiac pathways. This suggested that combining multiple microRNAs together would yield a more robust and well-rounded maturation response. Ideally, we would have sent miR-200a KO sample for sequencing, but we did not do this. The way in which we detail which miRs were chosen was by stating, "From each of the functional assays, we chose a miR that brought a significant increase in maturation to generate a cocktail that consisted of the smallest number of miRs." As we state, the choice for the full set of miRs was based on functional data as the primary factor. This resulted in the choice of miR-200a and the other miRs chosen. The other miRs not chosen were ruled out by poor performance in the functional assays, poor transcriptomic changes or redundancy, which

was the case for miR-208b which, while able to increase cell size was functionally redundant to Let7i.